# TGFβ-induced long non-coding RNA *LINC00313* activates Wnt signaling and promotes cholangiocarcinoma

Panagiotis Papoutsoglou [1,2], Raphaël Pineau[1], Raffaële Leroux[1], Corentin Louis[1], Anaïs L'Haridon[1], Dominika Foretek[2], Antonin Morillon [2], Jesus M Banales [3,4], David Gilot [1,5], Marc Aubry [1] & Cédric Coulouarn [1]✉

## Abstract

**Cholangiocarcinoma is a devastating liver cancer characterized by high aggressiveness and therapy resistance, resulting in poor prognosis. Long non-coding RNAs and signals imposed by oncogenic pathways, such as transforming growth factor β (TGFβ), frequently contribute to cholangiocarcinogenesis. Here, we explore novel effectors of TGFβ signalling in cholangiocarcinoma. *LINC00313* is identified as a novel TGFβ target gene. Gene expression and genome-wide chromatin accessibility profiling reveal that nuclear *LINC00313* transcriptionally regulates genes involved in Wnt signalling, such as the transcriptional activator *TCF7*. *LINC00313* gain-of-function enhances TCF/LEF-dependent transcription, promotes colony formation in vitro and accelerates tumour growth in vivo. Genes affected by *LINC00313* over-expression in CCA tumours are associated with *KRAS* and *TP53* mutations and reduce overall patient survival. Mechanistically, ACTL6A and BRG1, subunits of the SWI/SNF chromatin remodel-ling complex, interact with *LINC00313* and affect *TCF7* and *SULF2* transcription. We propose a model whereby TGFβ induces *LINC00313* in order to regulate the expression of hallmark Wnt pathway genes, in co-operation with SWI/SNF. By modulating key genes of the Wnt pathway, *LINC00313* fine-tunes Wnt/TCF/LEF-dependent transcriptional responses and promotes cholangiocarcinogenesis.**

**Keywords** Cholangiocarcinoma; Chromatin Remodelling; lncRNA; TGFβ Signalling; Wnt Pathway
**Subject Categories** Cancer; RNA Biology; Signal Transduction

## Introduction

Cholangiocarcinoma (CCA) is an aggressive cancer arising from the biliary tree characterized by resistance to chemotherapies and poor prognosis. CCAs are categorized into intrahepatic (iCCA) perihilar (pCCA) and distal (dCCA) CCAs (Banales et al, 2020). CCA subtypes exhibit high tumour cell heterogeneity, as well as specific genetic and epigenetic alterations, including frequent *KRAS*, *TP53* and *IDH1/2* mutation and *FGFR2* gene fusions with enhanced tumourigenic activity (Kendall et al, 2019). Accordingly, signals imposed by oncogenic pathways, such as fibroblast growth factor (FGF), phosphatidylinositol-3-kinase (PI3K)-AKT and transforming growth factor β (TGFβ) frequently contribute to CCA development and/or progression (Fouassier et al, 2019).

TGFβ signals through type I and type II TGFβ receptors (TβRI and TβRII) that activate the transcription factors SMAD2, SMAD3 and SMAD4, which modulate gene expression (Tzavlaki and Moustakas, 2020). TGFβ-responsive targets include protein-coding and non-coding genes, which participate in a wide range of processes, such as liver fibrosis, tumour progression and metastasis (Papoutsoglou and Moustakas, 2020). TGFβ exerts both anti- and pro-tumourigenic functions. In normal epithelium, TGFβ restricts tumour initiation by promoting cell cycle arrest and apoptosis. However, in advanced malignancies, cancer cells become unresponsive to its tumour-suppressive actions, but still obey its tumour-promoting functions, such as epithelial-to-mesenchymal transition (EMT) (Tu et al, 2019). In addition, TGFβ in the stroma facilitates cancer cell invasion and tumour immune evasion (Batlle and Massagué, 2019). In CCA, *SMAD4* is subjected to mutations, leading to gene inactivation. Loss of *SMAD4* is capable of nullifying the tumour suppressive axis of TGFβ. In addition, TGFβ is able to induce EMT and favour a fibrotic tumour microenvironment, thereby promoting metastasis (Papoutsoglou et al, 2019a). Notably, enhanced secretion of TGFβ2 in the stroma of iCCAs correlates with poor patient's prognosis (Sulpice et al, 2013).

[1]Inserm, Univ Rennes, OSS (Oncogenesis, Stress, Signaling) laboratory, UMR_S 1242, Centre de Lutte contre le Cancer Eugène Marquis, F-35042 Rennes, France. [2]ncRNA, Epigenetic and Genome Fluidity, CNRS UMR3244, Sorbonne University, PSL University, Institut Curie, Centre de Recherche, Paris, France. [3]Department of Liver and Gastrointestinal Diseases, Biogipuzkoa Health Research Institute, Donostia University Hospital, CIBERehd, Ikerbasque, San Sebastian, Spain. [4]Department of Biochemistry and Genetics, School of Sciences, University of Navarra, Pamplona, Spain. [5]Present address: Mechanistic & Structural Biology, Discovery Sciences, R&D, AstraZeneca, SE-48183 Mölndal, Sweden. ✉E-mail: cedric.coulouarn@inserm.fr

Long non-coding RNAs (lncRNAs) are transcripts longer than 200 nucleotides that lack protein-coding capacities. Long intergenic non-coding RNAs (lincRNAs) do not overlap with protein-coding genes. They form autonomous transcriptional units under the control of their own promoters. Although they structurally resemble messenger RNAs (mRNAs), they are not translated into proteins. However, short open reading frames (ORFs) are inherent to some lncRNAs, rendering them capable of encoding short functional peptides (de Andres-Pablo et al, 2017). LncRNAs play crucial roles in physiological and pathological processes, including cancer. They do so by regulating molecular processes, such as gene transcription, splicing and translation, via interactions with DNA, proteins, mRNAs or microRNAs (miRNAs) (Yao et al, 2019). Several lncRNAs promote the proliferative, migratory and invasive properties of CCA cells (Jiang and Ling, 2019). Notably, lncRNAs with oncogenic functions, such as *lncRNA-ATB* (Lin et al, 2019), *CCAT1* (Zhang et al, 2017) and *TLINC* (Merdrignac et al, 2018) are over-expressed in CCA. Here, we focus on novel effectors of the TGFβ pathway and describe the biological role and clinical relevance of the previously unknown *long intergenic non-protein coding RNA 313* (*LINC00313*) in CCA.

## Results

### *LINC00313* is a novel TGFβ target in CCA

By gene expression profiling, we identified 103 non-redundant genes differentially expressed by TGFβ in both HuCCT1 and Huh28 human CCA cell lines (Fig. 1A), including known (*SERPINE1*) and novel coding and non-coding TGFβ targets (Dataset EV1). Notably, *LINC00313* was greatly induced by TGFβ in CCA human cells (HuCCT1, Huh28) and normal human cholangiocytes (NHC), but not in the other CCA cell lines tested (Fig. 1A,B; Appendix Fig S1A). *LINC00313* was upregulated by TGFβ in a dose-dependent manner (Fig. 1C) and appeared to be an intermediate-to-late and early responsive gene in HuCCT1 and Huh28 cells, respectively (Fig. 1D). Using exon-specific primers, we further confirmed the induction of *LINC00313* isoforms by TGFβ in CCA cells (Appendix Fig. S1B, S1C). *LINC00313* was also highly expressed in hepatocellular carcinoma (HCC) cell lines HepG2 and Hep3B but TGFβ had no impact on its expression (Appendix Fig. S1D). Finally, subcellular fractionation demonstrated that *LINC00313* was mainly a nuclear RNA, similar to *MALAT1* and *RNU48*, although a significant part (20%) was also observed in the cytosol (Fig. 1E).

The gene encoding *LINC00313* is located at the chromosome 21 (21q22.3), contains four exons and three introns and encodes a 579-nt-long RNA (NR_026863.1) (Fig. EV1A). *LINC00313* shows tissue-specific expression patterns, with higher expression levels observed in liver, adrenal glands, gallbladder, testis and endometrium (Fig. EV1B). In addition, a high expression of *LINC00313* was observed in primary liver cancers (HCC and CCA) from the TCGA dataset (Fig. EV1C). Evaluation of its protein-coding potential using PhyloCSF score and CPAT coding probability revealed that it is likely to be a non-coding transcript (PhyloCSF score: −35.8813, CPAT coding probability: 8.59%). Moreover, the *LINC00313* genomic locus does not seem to be conserved (lncipedia.org/db/transcript/LINC00313:6). Several lncRNAs contain suboptimal

ORFs, from which small peptides may be produced. Thus, we utilized ORF Finder, and queried the *LINC00313* sequence for predicted ORFs. Interestingly, the NR_026863.1 transcript was predicted to carry seven ORFs within its sequence, although only one of them (ORF3) could potentially encode a 77-amino acid, putative and uncharacterized protein (UniProtKB/Swiss-Prot: P59037.1), according to protein Blast (BLASTP) (Fig. EV1D). The rest of the ORFs were predicted to encode peptides with no obvious similarity to known proteins. Querying the Ensembl database we observed that *LINC00313* has nine transcript variants, suggesting that it may undergo alternative splicing (Fig. EV1E).

### TGFβ-induced *LINC00313* expression requires TβRI/Smad-dependent pathways

TGFβ signalling is mediated through Smad-dependent (canonical) or Smad-independent (non-canonical) pathways (Tzavlaki and Moustakas, 2020). TβRI inhibitor LY2157299 abolished TGFβ-induced *LINC00313* expression in TGFβ-responsive cells (Fig. 2A). Similarly, blocking SMAD3 activation using SIS3 inhibitor completely abolished the TGFβ-mediated upregulation of *LINC00313* and *SERPINE1* in CCA cell lines (HuCCT1 and Huh28) (Fig. 2B). In HuCCT1, SMAD4 silencing reduced baseline and TGFβ-induced *LINC00313* expression similar to the combined SMAD2/3/4 silencing (Fig. 2C; Appendix Fig. S2A). In Huh28, individual SMAD2 or SMAD4 or combined SMAD2/3/4 silencing also prevented TGFβ-induced *LINC00313* expression (Fig. 2D; Appendix Fig. S2A). Silencing of SMADs was also verified at the protein level (Appendix Fig. S2B).

As regard to the non-canonical TGFβ pathway, p38 inhibition decreased, while MEK inhibition increased *LINC00313* expression (Fig. EV2A). Inhibition of c-Jun N-terminal kinase (JNK) induced *LINC00313* in the presence of TGFβ in HuCCT1 cells (Fig. EV2A). The efficiency of MAPKs inhibition was verified (Fig. EV2B). In Huh28 cells, *LINC00313* remained unchanged after blocking MAPKs (Fig. EV2A). Since p38 activity can be exerted by different p38 isoforms (p38α, p38β, p38γ and p38δ) we silenced each one of them and evaluated whether *LINC00313* expression is regulated by specific p38 isoforms (Fig. EV2C). We observed that experimental silencing of individual p38 isoforms (p38β encoded by *MAPK11*, p38γ encoded by *MAPK12*, p38δ encoded by *MAPK13* and p38α endoded by *MAPK14*) did not cause any significant change in TGFβ-induced *LINC00313* expression (Fig. EV2D). Thus, mainly the canonical TGFβ pathway is required for the TGFβ-mediated upregulation of *LINC00313*.

### *LINC00313* modulates the expression of genes involved in the Wnt pathway

Based on its predominant nuclear localization, we hypothesized that *LINC00313* acts at the transcriptional level to regulate gene expression (Fig. 3A). Gain-of-function tools were developed in HuCCT1 cells (Fig. EV3A). An efficient *LINC00313* over-expression (Fig. EV3B) and a nuclear abundance similarly to parental HuCCT1 cells were confirmed (Fig. EV3C). In an independent experiment using single molecule inexpensive fluorescent *in situ* hybridization (smiFISH), we observed mainly nuclear *LINC00313* RNA signal (Fig. EV3D), confirming our previous observation, using subcellular fractionation. RNA-seq analysis

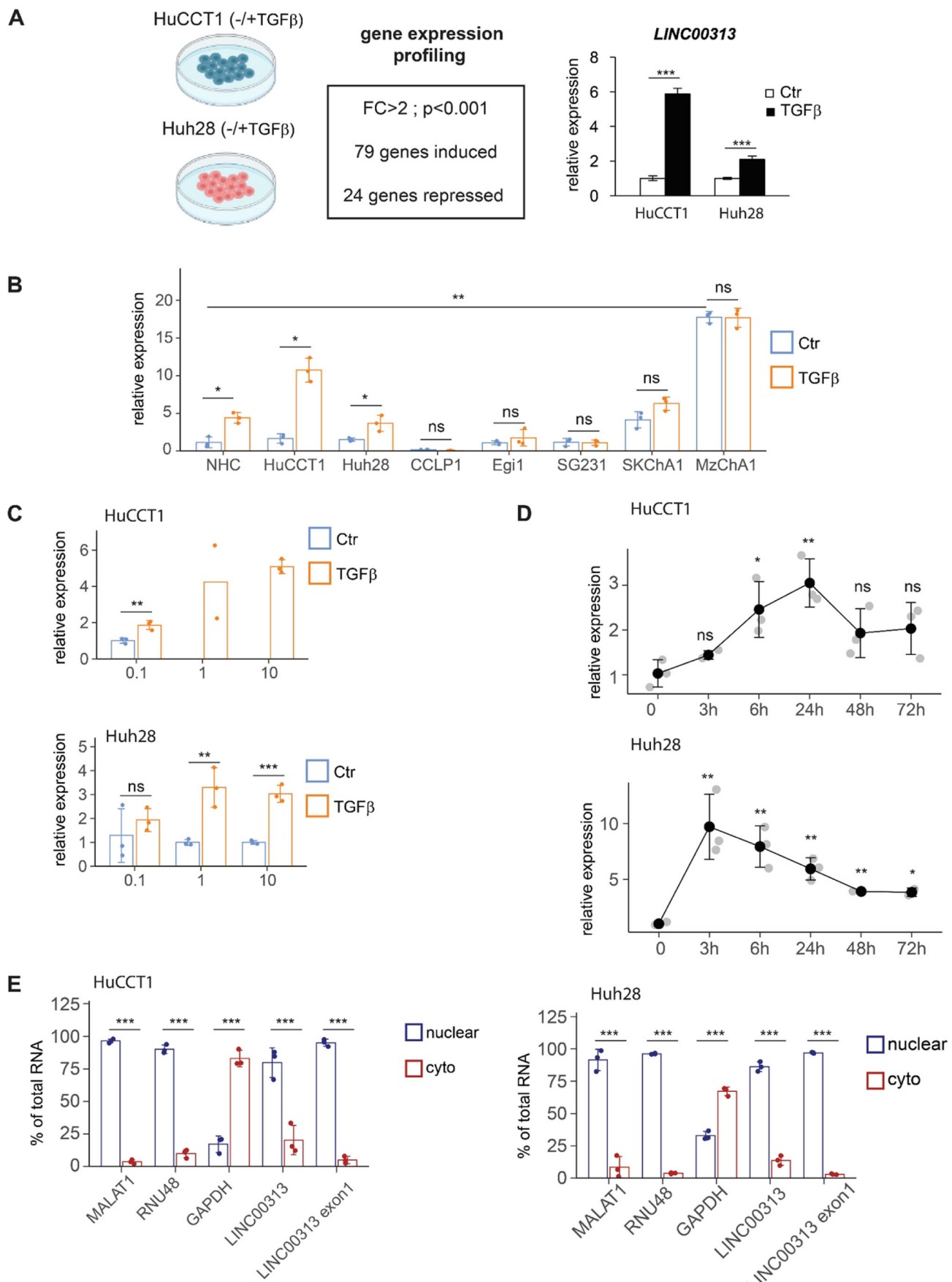

◄ **Figure 1.  *LINC00313* is a TGFβ target gene.**

(A) Experimental design to dissect TGFβ-regulated genes in CCA cells and *LINC00313* expression in HuCCT1 and Huh28 in response to TGFβ1, as identified by gene expression profiling. (B) *LINC00313* expression in response to TGFβ1 stimulation for 16 h, in normal human cholangiocytes (NHC) and CCA cell lines. (C) *LINC00313* expression in response to the indicated doses of TGFβ1 in HuCCT1 and Huh28 cell lines. (D) *LINC00313* expression in response to TGFβ1 for the indicated time periods in HuCCT1 and Huh28 cell lines. (E) *MALAT1*, *RNU48*, *GAPDH* and *LINC00313* RNA levels in nuclear and cytoplasmic fractions of HuCCT1 and Huh28 cell lines. Data information: Data in graphs are presented as mean ± SD ($n = 3$ biological replicates). *$P ≤ 0.05$, **$P ≤ 0.01$, ***$P ≤ 0.001$ (Student's *t*-test), n.s. not significant. Source data are available online for this figure.

identified 334 up and 316 downregulated genes after *LINC00313* gain-of-function (Figs. 3B, EV3E and Dataset EV2). Upregulated genes were related to signalling pathways regulating pluripotency of stem cells (e.g. WNT5A, *TCF7*, *AXIN2*, *FZD2* and *ID1*) (Fig. 3C). Gene Set Enrichment Analysis (GSEA) further highlighted signatures of Hippo, Wnt and TGFβ pathways in the gene expression profile of *LINC00313* overexpressing cells (Figs. 3D and EV3F). Induction of *TCF7*, *WNT5A*, *AXIN2* and *SULF2* following *LINC00313* gain-of-function was also validated (Figs. 3E and EV3G).

Next, gene expression profiling after *LINC00313* silencing was performed in the presence or absence of TGFβ (Appendix Fig. S3A and Dataset EV3). GO analysis confirmed that TGFβ-induced genes participated in cell migration, extracellular matrix organization and cell differentiation, as expected (Appendix Fig. S3B). KEGG pathway analysis revealed that many genes repressed after *LINC00313* silencing are associated with cancer signalling pathways regulating pluripotency of stem cells, including the Wnt signalling (Appendix Fig. S3C). To better clarify the relationship between TGFβ stimulation and Wnt pathway activation we performed GSEA using four well-characterized Wnt signalling pathway signatures (from MSigDB, UC San Diego and Broad Institute, www.gsea-msigdb.org) and the gene expression profiles of HuCCT1 cells incubated with TGFβ versus non-exposed cells. Supporting our initial hypothesis of a crosstalk between TGFβ and Wnt signalling, GSEA demonstrated that all four tested Wnt signalling pathway signatures are positively and significantly enriched ($P < 0.01$) in the gene expression profiles of HuCCT1 cells stimulated with TGFβ (Appendix Fig. S3D). In addition, several key genes from the Wnt signalling pathway (e.g. *TCF7*, *WNT11*, *WNT5B*, *WNT7A*) are present in the core enrichment when performing GSEA, as exemplified below for the GOBP_CANONICAL_WNT_SIGNALING_PATHWAY (Appendix Fig. S3E).

In agreement with *LINC00313* gain-of-function experiments, *WNT5A*, *AXIN2* and *SULF2* expression levels were downregulated upon *LINC00313* silencing in HuCCT1 cells (Fig. 3F). Expression of *TCF7*, *SULF2* and *WNT5A* was also diminished in *LINC00313*-silenced Huh28 and TFK1 cells (Appendix Fig. S4A, S4B). We excluded the possibility that our siRNA non-specifically targets the aforementioned mRNAs, by performing nucleotide BLAST (Appendix Fig. S4C). We interestingly observed that a part of LINC00313 sequence is highly similar to a lncRNA annotated as *LINC01669* (or *LOC102724354*). Our siRNA targets exon 3 of *LINC00313* and, therefore, also *LINC01669* (Appendix Fig. S4D). However, according to NCBI and Ensembl databases this annotation is considered as a false duplication and therefore likely redundant with *LINC00313*. Nevertheless, the observations above were reproduced in Huh28 and TFK1 cells using a second individual siRNA (siLINC#2), that targets exon 1 (nucleotides 33

to 51), in order to exclude possible siRNA off-target effects (Appendix Fig. S4E, S4F).

## Nuclear *LINC00313* alters chromatin accessibility at genomic loci of Wnt-related genes

The predominant nuclear localization of *LINC00313* prompted us to hypothesize that it may regulate gene expression by modifying chromatin conformation. To explore this possibility, we investigated genome-wide alterations in chromatin accessibility in HuCCT1 cells over-expressing *LINC00313* using an assay for transposase-accessible chromatin followed by high-throughput sequencing (ATAC-seq). ATAC-seq identified genome-wide alterations in chromatin accessibility, including 23,628 unique peaks in *LINC00313* over-expressing cells (pcLINC00313) (Fig. 4A; Appendix Fig. S5A, S5B). No global alteration of the location of peaks, relative to genomic annotations was observed (Appendix Fig. S5C). Differential region analysis revealed 1657 genomic regions with increased accessibility and 2090 regions with decreased accessibility in pcLINC00313 cells (Fig. 4B and Dataset EV4). HOMER motif enrichment analysis highlighted Fos-related antigen 1 (Fra1), an AP-1 transcription factor subunit, as the most significant enriched motif, although additional motifs of known transcription factors were identified (Appendix Fig. S5D).

We then integrated ATAC-seq and RNA-seq data and pinpointed the common genes. We found 44 genes that show both altered chromatin accessibility and altered expression after *LINC00313* over-expression (Fig. 4C). Interestingly, many of these genes were associated with the Wnt signalling pathway (Fig. 4D), known to play a key role in CCA progression (Boulter et al, 2015). Thus, we decided to investigate the regulation of selected genes of the pathway, such as the master transcription factor *TCF7*. Notably, we observed increased peak signal around the transcription start site and in the gene body of *TCF7* locus (Fig. 4E), as well as elevated *TCF7* mRNA and protein levels in *LINC00313* over-expressing cells (Fig. 4F). Interestingly, TGFβ induced *TCF7* expression whereas silencing *LINC00313* reduced *TCF7* mRNA and protein levels (Fig. 4G) in both control and TGFβ stimulated cells, suggesting that *LINC0313* may promote chromatin opening and enhanced transcriptional activity at the *TCF7* locus.

## *LINC00313* potentiates TCF/LEF transcriptional responses

Then, we investigated whether *LINC00313* could overall modulate Wnt/β-catenin-dependent transcription. We first validated luciferase reporter assays in HuCCT1 cells treated with CHIR99021 (CHIR), a glycogen synthase kinase 3 (GSK3) inhibitor that activates Wnt signalling (Fig. EV4A). Treatment with CHIR induced TCF/LEF luciferase activity in HuCCT1 cells expressing

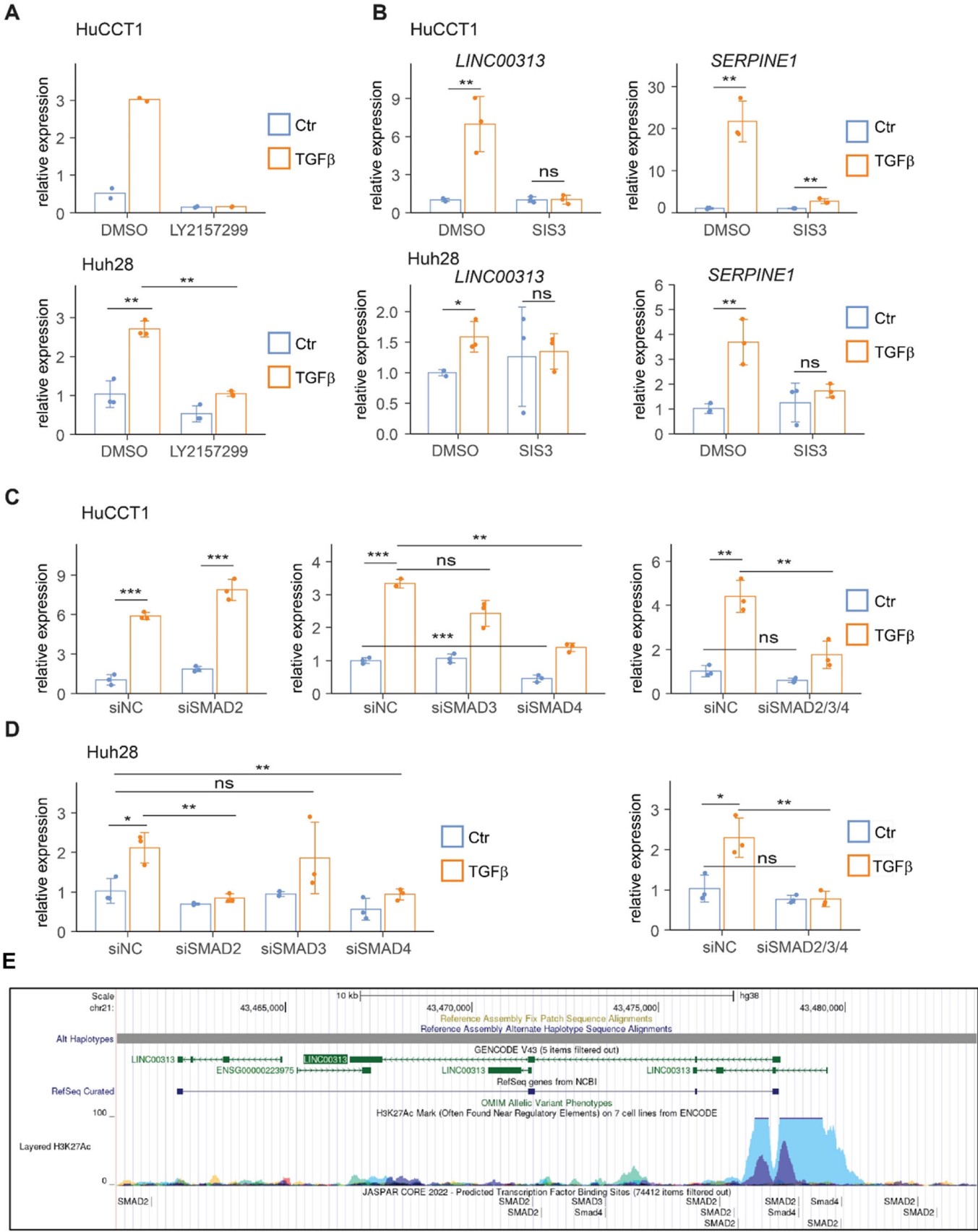

**Figure 2. TGFβ induces *LINC00313* through TβRI/Smad-dependent pathways.**

(A) *LINC00313* expression in HuCCT1 and Huh28 cells treated with LY2157299 or DMSO, and stimulated with TGFβ1 or BSA/HCl for 16 h. (B) *LINC00313* and *SERPINE1* expression in HuCCT1 and Huh28 treated with SIS3 or DMSO and stimulated with TGFβ1 or BSA/HCl for 16 h. (C) *LINC00313* expression in HuCCT1 transiently transfected with siRNA targeting *SMAD2*, *SMAD3*, or *SMAD4*, alone or in combination, with or without TGFβ1. (D) *LINC00313* expression in Huh28 transiently transfected with siRNA targeting *SMAD2*, *SMAD3*, or *SMAD4*, alone or in combination, with or without TGFβ1. (E) Snapshot of the UCSC genome browser showing predicted SMAD-binding regions in *LINC00313* gene locus, using the JASPAR CORE 2022 collection. Active histone mark (H3K27ac) is also shown around LINC00313 TSS. Data information: Data are presented as mean ± SD ($n = 3$ biological replicates). *$P \le 0.05$, **$P \le 0.01$, ***$P \le 0.001$ (Student's *t*-test), n.s.: not significant. In panel (A) the experiment in HuCCT1 cells was performed twice ($n = 2$ biological replicates) and data are presented as single data points. Source data are available online for this figure.

a wild-type TCF/LEF reporter (Top-Flash), but not a mutant reporter (FOP-Flash) (Fig. EV4B). Also, in HuCCT1 cells stably expressing a TCF/LEF reporter, CHIR increased reporter activity (Fig. EV4C). Silencing *LINC00313* repressed CHIR-induced TCF/LEF reporter activity (Fig. 5A) and the associated target genes (Fig. 5B). *LINC00313* over-expression further enhanced the baseline and CHIR-induced TCF/LEF-luciferase activity (Fig. 5C). XAV939, a Wnt pathway inhibitor that targets tankyrases, did not influence TCF/LEF responses in control cells, but reduced it in pcLINC00313 cells, which exhibit enhanced Wnt activity (Fig. 5D).

The nuclear import of β-catenin is the major event that drives Wnt transcriptional responses, upon Wnt activation. CHIR treatment promoted a partial nuclear translocation of β-catenin in control cells, but *LINC00313* over-expression did not have any further effect (Fig. EV4D, EV4E). Collectively, the data support a model whereby *LINC00313* acts as a positive regulator of TCF/LEF-mediated signalling, but is dispensable for the initial steps of Wnt activation.

## *LINC00313* promotes CCA colony-forming capacities in vitro and tumour growth in vivo

Wnt signalling regulates cancer stem cell maintenance, cancer cell proliferation and migration (Rim et al, 2022). In this context, we showed that *LINC00313* boosted the ability of single cells to form colonies in vitro (Fig. 5E). Moreover, *LINC00313* over-expression enhanced HuCCT1 cell proliferation (Appendix Fig. S6A) and reduced *CDKN1A*, while potentiated *ID1* and *ID3* gene expression (Appendix Fig. S6B), thereby counteracting TGFβ-target genes, involved in cell cycle progression. Furthermore, *LINC00313* promoted cell migration and invasion (Appendix Fig. S6C) while silencing *LINC00313* slightly decreased the gap closure in a wound-healing assay (Appendix Fig. S6D).

Administration of CHIR increased the viability of control HuCCT1 cells, an effect that was more pronounced in cells over-expressing *LINC00313* (Fig. EV4F). Moreover, CHIR boosted colony formation of HuCCT1 control cells and to a larger magnitude of *LINC00313* over-expressing cells (Fig. EV4G). Interestingly, XAV939 reduced colony formation (Fig. EV4G). We confirmed the efficiency of the inhibitors by measuring *AXIN2* expression, as a typical Wnt-target gene. Indeed, CHIR strongly induced *AXIN2*, an effect that was enhanced in *LINC00313* over-expressing cells, whereas XAV939 slightly reduced *AXIN2*. Also, CHIR potentiated *TCF7* but not *LINC00313*, while XAV939 had no effect on their expression (Fig. EV4H).

Interestingly, *LINC00313* accelerated tumour growth in vivo when cells were xenografted in nude mice (Fig. 5F; Appendix Fig S7A). Expression analysis revealed a positive correlation between *LINC00313* and *TCF7*, *SULF2* and *AXIN2* mRNA levels in resected

tumours (Fig. 5G). Moreover, we measured increased *SULF2* and *AXIN2* expression in *LINC00313* xenograft tumours, compared to control (Fig. 5H). Although *TCF7* mRNA levels were unchanged (Fig. 5H), we observed increased TCF7 protein expression in *LINC00313* tumours (Appendix Fig. S7B). *LINC00313* had no impact on the development of spontaneous metastases in mice lung or liver (Appendix Fig. S7C).

## The SWI/SNF complex subunit ACTL6A interacts with LINC00313 transcripts

Nuclear lncRNAs frequently regulate gene expression by facilitating chromatin remodelling or nucleosome repositioning, via interactions with proteins (Han and Chang, 2015). In order to decipher the molecular mechanisms underlying *LINC00313* biological functions, we performed an RNA pull-down assay followed by an unbiased proteomic analysis, using in vitro transcribed RNA (Fig. 6A), aiming at identifying molecular partners that act in cooperation with *LINC00313*. In total, 1538 proteins were identified to interact with *LINC00313* and 1840 proteins with the negative control *firefly luciferase* (*f-luc*) mRNA (Fig. 6B, Dataset EV5). From the above, 121 proteins were specifically bound to *LINC00313* (Fig. 6B). In order to single out interesting candidates, we set three sorting layers (Fig. 6C). First, we focused on nuclear proteins (51/121), considering the nuclear localisation of *LINC00313*. Protein network analysis revealed multiple physical and functional associations and clustered them in four groups. The largest cluster contained proteins involved in cell cycle regulation and the rest members of the transcription factor TFIID complex, the integrator complex and proteins with RNA helicase activity (Fig. 6D). Second, we predicted the binding of *LINC00313* to 51 nuclear proteins using catRAPID (Agostini et al, 2013). Thirty-two proteins were predicted to interact with *LINC00313* (interaction propensity ≥ 75, discriminative power ≥ 98%) (Dataset EV5). Third, GO analysis for molecular function of the 32 proteins revealed several transcription factors or chromatin modifiers (Fig. 6E), including actin-like 6 A (ACTL6A) (Fig. 6F), a subunit of the SWI/SNF chromatin remodelling complex (Chang et al, 2021). Given that *LINC00313* modulates chromatin state of certain loci, we investigated the physical and functional association between *LINC00313* and ACTL6A deeper. Initially, we verified the nuclear localization of ACTL6A. TGFβ stimulation did not alter the nuclear abundance of ACTL6A (Fig. 6G). A specific interaction between in vitro transcribed *LINC00313* lncRNA and endogenous ACTL6A was demonstrated in HuCCT1, especially in the presence of TGFβ (Fig. 6H). Similarly, in HA-tagged ACTL6A over-expressing HEK293T cells, an even stronger interaction between in vitro transcribed *LINC00313* and HA-ACTL6A was observed (Fig. 6I). We also confirmed this interaction by RNA immunoprecipitation

**A**

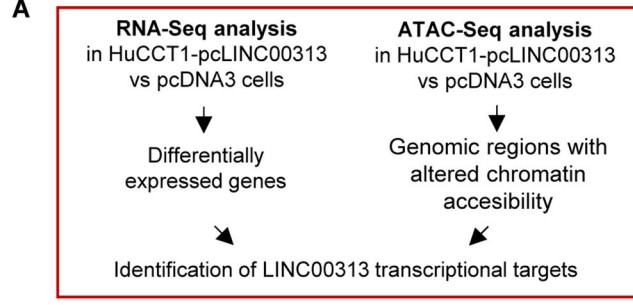

**B**

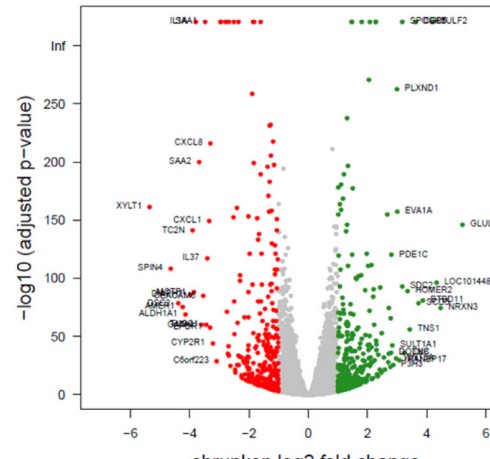

| RNA-seq (shrunken log2FC>1, adj. p-value<0.1) | | | |
|---|---|---|---|
| Ctr group | Exp. group | Genes Up | Genes Down |
| pcDNA3 | pcLINC00313 | n=334 | n=316 |

**C**

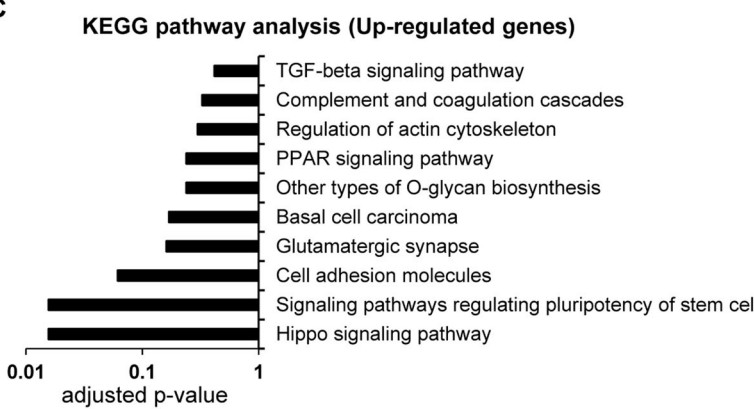

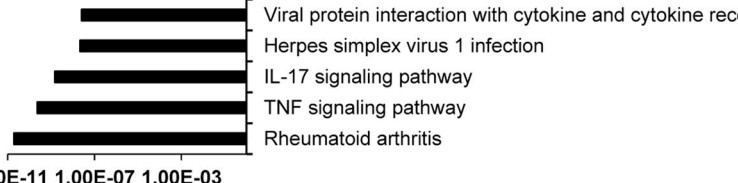

**D**

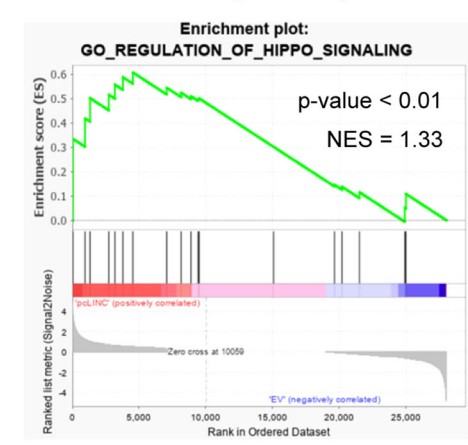

**E**

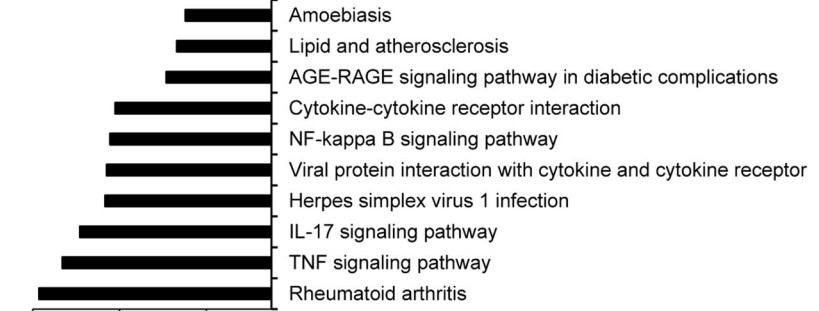

**F**

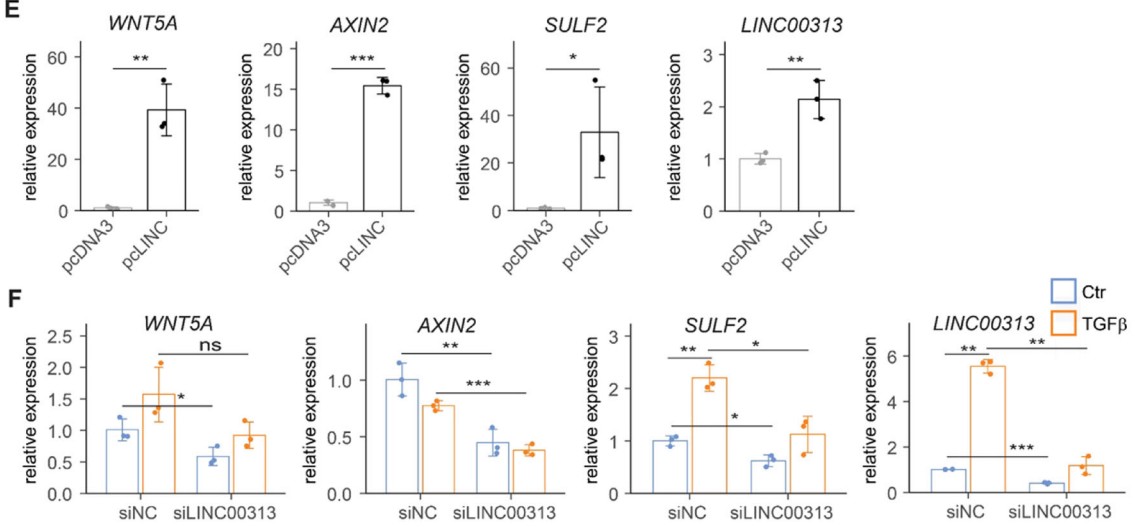

**Figure 3. *LINC00313* regulates genes involved in the Wnt pathway.**

(A) Experimental approach to identify *LINC00313* transcriptional targets. (B) Number of differentially expressed genes in pcLINC00313 *versus* pcDNA3 HuCCT1 cells (shrunken log2FC > 1, adjusted *p*-value < 0.1). Volcano plot shows differentially expressed genes (green colour: upregulated genes, red colour: downregulated genes). (C) KEGG pathway analysis of up or downregulated genes upon LINC00313 over-expression. (D) Example of GSEA of differentially expressed genes in response to LINC00313 over-expression. Shown is the enrichment of a Hippo signalling signature in the gene expression of cells overexpressing LINC00313. (E) *WNT5A, AXIN2* and *SULF2* mRNA levels in HuCCT1 expressing *LINC00313* (pcLINC) or empty vector (pcDNA3). (F) *WNT5A, AXIN2* and *SULF2* expression in HuCCT1 transfected with siLINC00313 and stimulated with TGFβ1 or BSA/HCl for 16 h. Data information: In panel (B) statistically significant differential expressed genes were identified using DEseq2. In panel (C) adjusted *p*-value was calculated using the Benjamini-Hockberg method for correction for multiple hypotheses testing. In panel (D) GSEA results are displayed as a normalized enrichment score (NES) and presented as an enrichment plot. The Kolmogorov-Smirnov test was used for statistical analysis of GSEA. In (E,F) data are presented as mean ± SD (*n* = 3 biological replicates). *$P \le 0.05$, **$P \le 0.01$, ***$P \le 0.001$ (Student's *t*-test). Source data are available online for this figure.

(RIP) assays. Interestingly, TGFβ stimulation increased the enrichment of endogenous *LINC00313* to immunoprecipitated ACTL6A (Fig. 6J). Overall, *LINC00313* forms ribonucleoprotein complexes with several nuclear proteins that belong to transcriptional regulatory networks.

## ACTL6A silencing or pharmacological inhibition of the SWI/SNF complex diminishes TCF/LEF-mediated gene expression

Next, we evaluated the effects of ACTL6A on TCF/LEF-dependent transcription. Silencing ACTL6A attenuated *TCF7* mRNA and protein levels but did not affect TGFβ-induced *LINC00313* (Fig. 7A,B). ACTL6A silencing also resulted in decreased basal and CHIR-induced TCF/LEF luciferase activity (Fig. 7C) and a drop in *TCF7* and *AXIN2* mRNA levels (Fig. 7D–F). Rescue experiments demonstrated that ACTL6A silencing dampened the *LINC00313*-mediated upregulation of *SULF2* mRNA (Fig. 7G) and *TCF7* mRNA and protein levels (Fig. 7H–J). Conversely, simultaneous silencing of *LINC00313* and *ACTL6A* had an additive inhibitory effect towards *TCF7* gene expression, compared to single knock-down (Fig. 7K).

Since ACTL6A is an accessory subunit of SWI/SNF, we hypothesized that blocking the catalytic activity of the complex may yield effects similar to ACTL6A silencing. Thus, we utilized two inhibitors, targeting different domains of the core catalytic subunits SMARCA2 (BRM)/SMARCA4 (BRG1). Notably, the bromodomain inhibitor PFI-3 did not affect *SULF2* or *TCF7* mRNA expression. In contrast, administration of BRM/BRG1 ATPase inhibitor significantly reduced *SULF2* mRNA (Appendix Fig. S8A) and *TCF7* expression both at the mRNA (Appendix Fig. S8B) and protein levels (Appendix Fig. S8C), in *LINC00313* over-expressing cells, implying that these two genes are indeed targets of SWI/SNF. Moreover, silencing *LINC00313* diminished *TCF7* expression an effect that was more pronounced after BRM/BRG1 ATPase inhibitor treatment (Appendix Fig. S8D). Consistent with the effects on the individual genes, BRM/BRG1 ATPase inhibitor, but not PFI-3 treatment resulted in a modest, but significant decrease of CHIR-induced TCF/LEF-luciferase reporter activity (Appendix Fig. S8E).

## BRG1 binds to *LINC00313* and is enriched in chromatin regions whose accessibility is enhanced by *LINC00313*

Then we hypothesized that *LINC00313* may interact with additional subunits of the SWI/SNF complex, such as the core subunits. Indeed, we confirmed an endogenous ribonucleoprotein

complex between *LINC00313* and the catalytic subunit BRG1 by RIP-qPCR and TGFβ stimulation further enhanced this interaction (Fig. 7L). We also evaluated the binding of BRG1 to the chromatin regions of *TCF7* and *SULF2* that are in an "open" conformation upon *LINC00313* over-expression, as revealed by ATAC-seq. Interestingly, BRG1 binding to these regions was increased in *LINC00313* over-expressing cells (Appendix Fig. S8F), suggesting that *LINC00313* could facilitate the loading of BRG1 to *TCF7* and *SULF2* gene loci, thereby enhancing their transcription.

Overall, we suggest a mechanism, whereby increased *LINC00313* levels promote chromatin accessibility in an ACTL6A/SWI/SNF-dependent manner and facilitate transcription of *TCF7* and *SULF2*, resulting in enhanced Wnt activation (Fig. 7M).

## A *LINC00313* signature predicts poor prognosis in patients with CCA

From the TCGA dataset, *LINC00313* was correlated with overall survival in patients with CCA (Fig. EV5A). Although we did not detect *LINC00313* over-expression in CCA tissues, the expression levels of *ACTL6A, TCF7, WNT5A, AXIN2* and *SULF2* were all increased in CCA human tumours (Fig. EV5B). Interestingly, expression correlation analysis revealed a positive correlation between *LINC00313* and two of the identified in vitro target genes (*TCF7* and *AXIN2*) in a set of 39 iCCA human tissues (Fig. EV5C).

Next, we hypothesized that the activity of *LINC00313* could better reflect its clinical relevance. Thus, we performed RNA-seq analysis in control and *LINC00313* over-expressing xenograft tumours to establish a signature reflecting *LINC00313* activity. In total, 347 genes were upregulated and 327 genes were downregulated in *LINC00313*-overexpressing tumours (Fig. 8A). By performing a GSEA using gene signatures established in vitro and the gene expression profiles of mouse tumours, we confirmed that the gene expression profiles in xenografted mice reflect the in vitro analysis. Indeed, GSEA demonstrated that the differentially regulated genes identified in the in vitro experiments were enriched in the gene expression profiles of the in vivo experiments, as expected (Fig. 8B). Then, we integrated the in vivo signature with the gene expression profiles of 255 cases of clinically annotated human CCA. Integrative transcriptomics using principal component analysis (PCA) showed that the four control samples clustered together and were distinct to the four *LINC00313* over-expressing samples (Fig. 8C). Dimension 2 of the PCA was related to *LINC00313* activity and identified two main clusters of human CCA (Fig. 8C). Interestingly, the cluster associated with *LINC00313* activity correlated with lower overall survival (Fig. 8D). Furthermore, *IDH1* and *IDH2* mutations were more prevalent in control

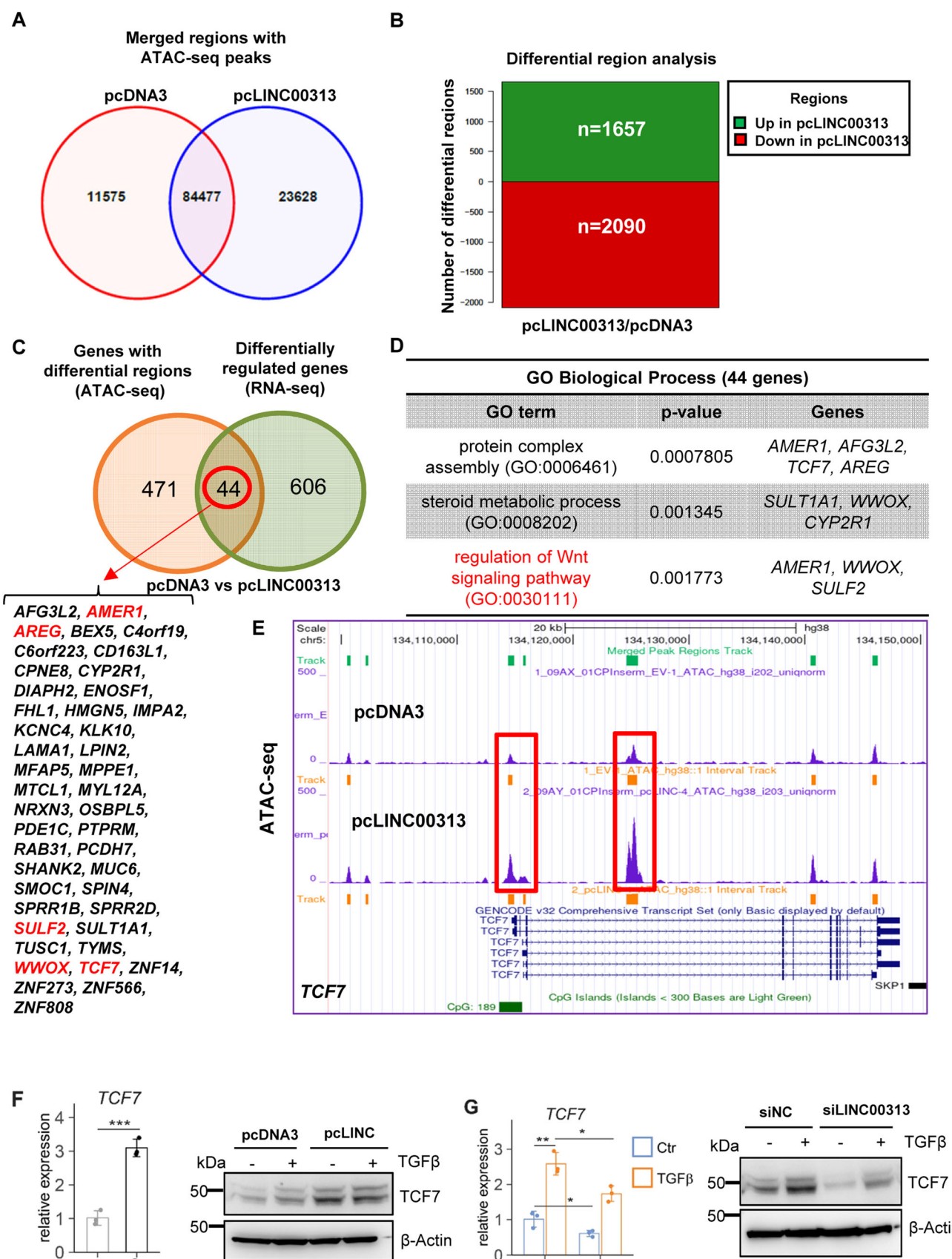

◄

**Figure 4. *TCF7* is transcriptionally induced by *LINC00313*.**

(A) Venn diagram showing merged regions with overlapping and unique ATAC-seq peaks between pcDNA3 and pcLINC00313 expressing HuCCT1 cells. (B) Bar plot illustrating the top differential regions. (C) Venn diagram highlighting the 44 genes, whose chromatin accessibility and expression are altered. (D) GO analysis for biological process of the 44 genes highlighted in panel (C) using Enrichr. (E) Snapshot of the UCSC genome browser with ATAC-seq peaks at human *TCF7* gene locus, in pcDNA3 *versus* pcLINC00313 over-expressing HuCCT1 cells. (F) *TCF7* mRNA and protein levels in HuCCT1 cells over-expressing LINC00313 (pcLINC) or pcDNA3 control. (G) *TCF7* RNA and protein levels in HuCCT1 cells, transfected with siLINC00313 and stimulated with TGFβ1 for 16 h. Data information: In panel (D) *p*-value was calculated by Enrichr using Fisher's exact test or hypergeometric test. In (F,G) qPCR data are presented as mean ± SD ($n = 3$ biological replicates). *$P ≤ 0.05$, **$P ≤ 0.01$, ***$P ≤ 0.001$ (Student's *t*-test). Source data are available online for this figure.

cluster, while *KRAS* and *TP53* mutations were significantly enriched in *LINC00313* cluster (Fig. 8E). We also observed increased levels of carbohydrate antigen 19-9 (CA19-9) and γ-glutamyltransferase (γ-GT) and decreased levels of albumin, markers of hepatobiliary disease and of unfavourable prognosis in CCA, in the *LINC00313* cluster (Fig. 8F). In addition, the *LINC00313* cluster was characterized by higher perineural invasion and regional lymph node metastasis (Fig. 8F). Collectively, we conclude that although *LINC00313* expression per se is not a strong prognostic factor in CCA, the in vivo gene signature that reflects its activity exhibits a strong prognostic value in terms of survival outcome.

## Discussion

By exploring the TGFβ-regulated transcriptome in human CCA cell lines, we identified *LINC00313* as a TGFβ-responsive gene, a finding that expands the growing list of TGFβ-regulated lncRNAs (Papoutsoglou and Moustakas, 2020). An interesting observation is that TGFβ is able to induce *LINC00313* only in a subset of CCA cell lines, as well as in normal human cholangiocytes. A possible explanation is that some CCA cell lines are not responsive to TGFβ stimulation. Since most of the CCA cell lines have not yet been fully characterized, concerning the responsiveness to TGFβ, a deeper investigation on TGFβ family members is required. On the other hand, TGFβ did not affect *LINC00313* expression in HepG2 and Hep3B, two HCC cell lines known to respond to TGFβ signalling, suggesting that TGFβ may selectively regulate this gene in certain cancer types or normal cell lines.

*LINC00313* is a lincRNA that possesses oncogenic properties and is a marker of poor prognosis in papillary thyroid cancer (Wu et al, 2018; Yan et al, 2019). Also, a pro-tumorigenic role of *LINC00313* has been described in osteosarcoma (Xing et al, 2022) and cervical carcinoma (Zhai et al, 2021). Moreover, *LINC00313* is upregulated in glioma and promotes tumorigenesis, via enhancing cell proliferation, migration and invasion (Shao et al, 2019). The functional role of *LINC00313* in CCA remained unknown but our data support a pro-oncogenic role. We demonstrated that *LINC00313* favours CCA tumour growth in vivo and colony formation in vitro. Although we did not observe upregulation of *LINC00313* in human CCA, we showed that in vivo gene signature associated with *LINC00313* gain-of-function correlated with poor overall survival. We speculate that *LINC00313* activity, rather than its expression, is of paramount importance for the stratification of CCA patients based on survival outcomes.

Integration of ATAC-seq and RNA-seq data allowed us to select genes involved in Wnt signalling. The Wnt pathway plays crucial roles in CCA progression and is often activated in CCA tissues.

CCA tumours are usually characterized by enhanced expression of several Wnt ligands and increased nuclear accumulation of β-catenin (Selvaggi et al, 2022). We focused on *TCF7* because it encodes one of the main transcription factors of Wnt/β-catenin pathway (Doumpas et al, 2019). Furthermore, *TCF7* reinforces CCA progression by positively regulating MYC and FOSL1 (Liu et al, 2019) and induces CCA proliferation and drug resistance, via regulating SOX9/FGF7/FGFR2 axis (Liu et al, 2022b). *SULF2* is the second interesting transcriptional target of *LINC00313*. It encodes a heparan sulfate-editing enzyme that activates Wnt signalling, by facilitating the bioavailability of Wnt ligands to their receptors (Rosen and Lemjabbar-Alaoui, 2010). *SULF2* is upregulated in human CCA and is associated with enhanced PDGFRβ-YAP signalling, tumour progression and chemoresistance. Importantly, targeting SULF2 protein with a monoclonal antibody abolishes tumour growth in a mouse CCA xenograft model (Luo et al, 2021). We also identified additional genes induced by *LINC00313*, such as *ID1* and *ID3*, which inhibit cell differentiation. Thus, we suggest that *LINC00313* positively regulates genes related to cell proliferation and stemness that could explain its effects on colony formation and CCA xenograft tumour growth.

Our proteomic analysis revealed numerous nuclear proteins that associate with *LINC00313*. We focused on ACTL6A, which is frequently over-expressed in cancer. For instance, ACTL6A is upregulated in HCC and promotes migration and invasion in vitro, as well as tumour growth and metastasis in vivo, via activating Notch signalling (Xiao et al, 2016). The role of ACTL6A in CCA is still unknown. However, the SWI/SNF catalytic subunit BRG1 is over-expressed and predicts a poor prognosis in iCCA. More importantly, it has been demonstrated that BRG1 activates the Wnt pathway through transcriptional regulation of Wnt receptor and target genes, but also via binding to β-catenin/TCF4 transcription complex, in HuCCT1 (Zhou et al, 2021). These observations are in line with our data indicating decreased activation of Wnt transcriptional responses upon *ACTL6A* silencing or BRG1 inhibition and increased BRG1 binding to *TCF7* and *SULF2* chromatin regions upon *LINC00313* ectopic expression.

At the molecular level, TGFβ promotes ribonucleoprotein complexes between *LINC00313* and ACTL6A without changing ACTL6A expression. TGFβ also enhances the interaction between the core subunit BRG1 and LINC00313 lncRNA. These findings open up the possibility that TGFβ may engage *LINC00313* to bring the chromatin remodelling machinery, in close proximity to SMADs, at the regulatory regions of selected targets. A direct interaction between SMADs and the core components of the SWI/SNF complex has been previously established indicating the importance of this complex in eliciting TGFβ-mediated transcriptional activation programs (Xi et al, 2008). Although ACTL6A is not a classical RNA-binding protein, it was previously reported to

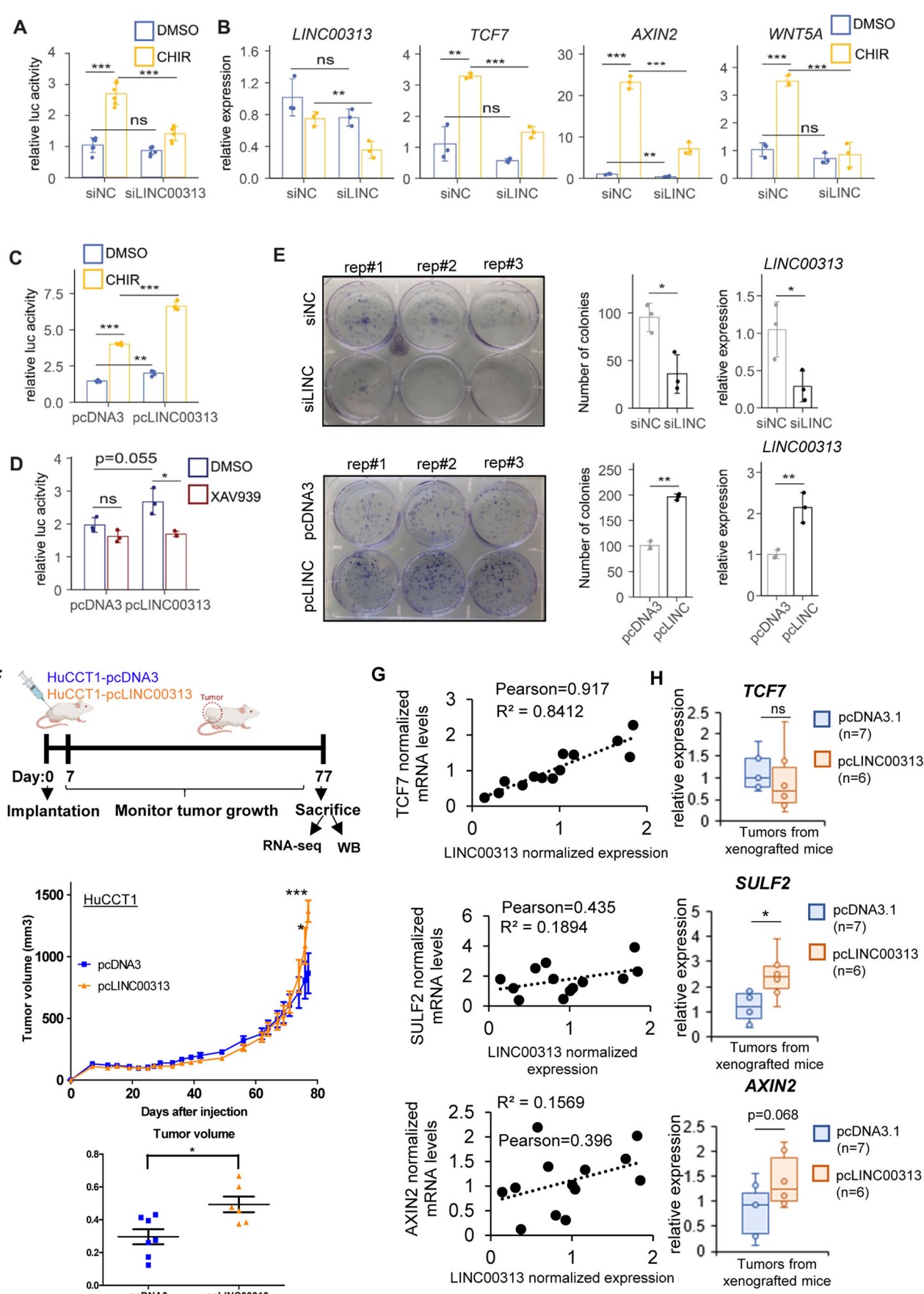

**Figure 5. *LINC00313* modulates TCF/LEF-dependent transcriptional responses and enhances colony formation.**

(A,B) TCF/LEF luciferase reporter assay (A) and qPCR analysis of *LINC00313*, *TCF7*, *AXIN2* and *WNT5A* expression (B) in HuCCT1 transiently transfected with siLINC00313 or siNC and treated with CHIR99021 or DMSO for 24 h. (C,D) TCF/LEF luciferase reporter assay in HuCCT1 stably over-expressing *LINC00313* or pcDNA3 and treated with CHIR99021 (C), XAV939 (D) or DMSO for 24 h. (E) Colony formation assay in HuCCT1 cells silenced or over-expressing *LINC00313*. Gene expression analysis to evaluate *LINC00313* knock-down or over-expression efficiency is also shown. (F) In vivo mouse xenograft experiment, tumour growth curves and dot plot depicting the tumour volume in pcDNA3 ($n = 7$) and pcLINC00313 ($n = 6$) xenografts. (G) Scatter plots depicting the correlation between *LINC00313* expression and each one of *TCF7*, *SULF2* and *AXIN2* mRNAs in RNA samples extracted from pcDNA3 and pcLINC00313 xenograft resected tumours. (H) Boxplots of *TCF7*, *SULF2* and *AXIN2* mRNA expression (*$P \leq 0.05$). Data information: Data from luciferase assays (A, C, D) data are presented as mean ± SD ($n = 3$ or $n = 6$ biological replicates). *$P \leq 0.05$, **$P \leq 0.01$, ***$P \leq 0.001$, n.s.: not significant (Student's *t*-test). In panel (E), data are presented as mean of the number of colonies ± SD ($n = 3$ biological replicates). *$P \leq 0.05$, **$P \leq 0.01$ (Student's *t*-test). In panel (F), statistical analysis was performed using a Mann–Whitney *U* test (*$P \leq 0.05$, ***$P \leq 0.001$). In panel H box plots are defined as follows: for *TCF7* expression in pcDNA3 group; min.: 0.6935, max.: 1.8315, median: 1.0069, mean: 1.1439, 1st quartile: 0.7834, 3rd quartile: 1.4541. For *TCF7* expression in pcLINC group; min.: 0.2343, max.: 2.2716, median: 0.7044, mean: 0.9431, 1st quartile: 0.4198, 3rd quartile: 1.2401. For *SULF2* expression in pcDNA3 group; min.: 0.3881, max.: 1.8193, median: 1.1946, mean: 1.1826, 1st quartile: 0.7388, 3rd quartile: 1.6992. For *SULF2* expression in pcLINC group; min.: 1.183, max.: 3.918, median: 2.407, mean: 2.432, 1st quartile: 1.918, 3rd quartile: 2.791. For *AXIN2* expression in pcDNA3 group; min.: 0.1191, max.: 1.5515, median: 0.9306, mean: 0.8080, 1st quartile: 0.3538, 3rd quartile: 1.1736. For *AXIN2* expression in pcLINC group; min.: 0.8743, max.: 2.1907, median: 1.2524, mean: 1.4254, 1st quartile: 1.0016, 3rd quartile:1.8615. Source data are available online for this figure.

interact with the lncRNA uc.291 to regulate epidermal differentiation (Panatta et al, 2020).

Several lncRNAs establish interactions with SWI/SNF subunits. For example, SWINGN lncRNA binds to SMARCB1 and facilitates transcription of target genes (Grossi et al, 2020). In HCC cells, lncTCF7 interacts with BRG1, BAF170 and SNF5 subunits and recruits SWI/SNF to *TCF7* promoter, thereby activating *TCF7* transcription in cis (Wang et al, 2015). Our mechanistic evidence proposes a model, whereby *LINC00313* acts *in trans* to regulate *TCF7* and *SULF2* transcription, through binding ACTL6A and BRG1, possibly via increased loading of SWI/SNF to *TCF7* and *SULF2* loci, setting an additional layer of regulation of TCF/LEF signalling. Undoubtedly, additional modes of action for *LINC00313* cannot be excluded. For example, nuclear lncRNAs frequently form triplex formation with DNA at chromatin regions, through direct base pairing (Zapparoli et al, 2020). It is worth to investigate whether *LINC00313* participates in these loops, in order to alter chromatin accessibility.

In addition, a molecular role for *LINC00313* as a competing endogenous RNA (ceRNA) has been previously proposed (He and Lin, 2023). Since some *LINC00313* transcripts reside in the cytosol in the two CCA cell lines examined, a bioinformatics search identified common miRNAs predicted to bind both *LINC00313* lncRNA and TCF7 mRNA, using miRcode (Appendix Fig. S9). Fifty non-redundant individual miRNAs or miRNA clusters targeting TCF7, 23 miRNAs targeting *LINC00313* and 16 common miRNAs for the two transcripts were found. By searching the current literature for established associations between *LINC00313* and miRNA, with a special interest on the 16 aforementioned predicted miRNAs, we observed that *LINC00313* may induce cell migration and invasion, possibly by inducing β-catenin and EMT by sequestering miR-138-5p, miR-150-5p, miR-204-5p and miR-205-5p, which target the EMT-associated *VIM* and *ZEB1* mRNAs (Liu et al, 2022a). In addition, *TCF7* could be transcriptionally regulated by β-catenin and TCF7L2 transcription factors (Zhu et al, 2015). Interestingly, β-catenin was identified in our mass-spec analysis, suggesting that *LINC00313* may somehow interact with β-catenin and modulate the activity of target genes, including *TCF7*. Nevertheless, our study highlights an important and new role for *LINC00313* as a modulator of Wnt/TCF signalling, that affects tumour progression and has clinical implications in CCA.

## Methods

### Cell culture

Primary cultures of normal human cholangiocytes (NHC) were established, characterized and cultured as previously described (Urribarri et al, 2014). HuCCT1 and Huh28 CCA cell lines were purchased from RIKEN BioResource Center (Tsukuba-shi, Japan) and CCLP1, Egi-1, SG231, TFK1, Sk-ChA-1 and Mz-ChA-1 were provided by Laura Fouassier (Paris, France). HepG2/C3A and HEK293T were purchased from ATCC (www.lgcstandards-atcc.org). Serum-starved cells were stimulated with the indicated concentrations of human recombinant TGFβ1 (R&D Systems) and for the indicated time periods as shown in the figures and figure legends.

### Treatments with inhibitors

The TβRI inhibitor LY2157299 (Interchim, LSK040) was added to the cells at a final concentration of 10 μM. The SMAD3 inhibitor (E)-SIS3 (MedChemExpress, HY-13013) was used at a final concentration of 10 μM. Cells were treated with 0.5 μM PD184352 (Sigma Aldrich, PZ0181), a MEK inhibitor, 10 μM SP600125 (Sigma Aldrich, S5567), a JNK inhibitor, 10 μM SB203580 (Sigma Aldrich, S8307), a p38 inhibitor and 0.1 μM wortmannin (Sigma Aldrich, W1628), a PI3K inhibitor. All treatments with the inhibitors were performed 1 h prior the addition of TGFβ1. The GSK3 inhibitor CHIR-99021 (TargetMol, T2310) and the tankyrase inhibitor XAV-939 (TargetMol, T1878) were used at 10 μM final concentration. The SMARCA2/4 bromodomain inhibitor PFI-3 (MedChemExpress, HY-12409) and the allosteric dual SMARCA2 and brahma-related gene 1 (BRG1)/SMARCA4 ATPase activity inhibitor BRM/BRG1 ATP Inhibitor-1 (MedChemExpress, HY-119374) were administrated to cells at 10 and 5 μM final concentrations, respectively. Dimethyl sulfoxide (DMSO) was used as a vehicle control treatment.

### Plasmid transfections

Human *LINC00313* (NR_026863.1) was synthesized and cloned into the pcDNA3.1/Hygro (+) expression vector (Invitrogen), by Eurofins (Eurofins Genomics, Germany). The sequence of the

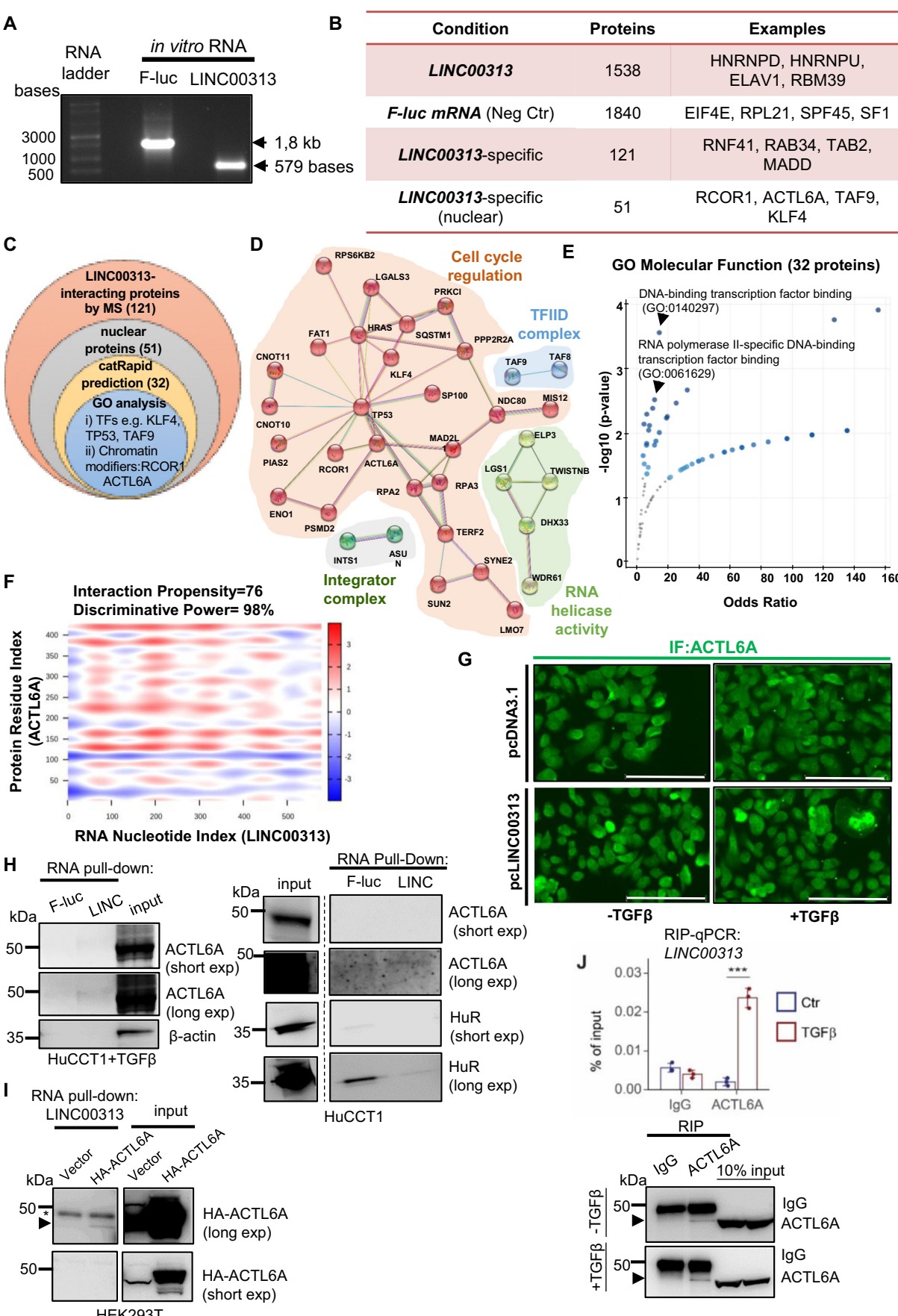

◀ **Figure 6. ACTL6A interacts with *LINC00313* transcripts.**

(A) In vitro transcribed F-luc and *LINC00313* RNAs. (B) Overview of interacting proteins identified by mass spectrometry. (C) Venn diagram representing the strategy to narrow down *LINC0313*-binding proteins. (D) Protein network analysis depicting physical and functional interactions between the 51 nuclear interactors, using STRING. (E) GO analysis for molecular function of 32 nuclear interactors predicted by CatRAPID. (F) Heatmap depicting the probability of interaction between *LINC00313* and ACTL6A. (G) Immunofluorescence of ACTL6A in control or *LINC00313* over-expressing HuCCT1 cells, treated or not with TGFβ1. Scale bars: 150 µm. (H) RNA pull-down assays in HuCCT1 cells, stimulated with TGFβ1 or not, using in vitro synthesized F-luc mRNA or *LINC00313* lncRNA, followed by immunoblotting for ACTL6A, HuR and β-actin. (I) RNA pull-down assay in HEK293T cells, over-expressing an empty vector or HA-tagged ACTL6A using in vitro synthesized *LINC00313* lncRNA. An arrow depicts the specific HA-ACTL6A protein band, while an asterisk marks unspecific protein bands. (J) RIP assay for endogenous ACTL6A followed by qPCR for *LINC00313* in HuCCT1 cells treated with TGFβ or BSA/HCl (vehicle control) for 16 h ($n = 3$). An immunoblotting to verify the efficiency of ACTL6A immunoprecipitation is also shown. The arrows depict the specific endogenous ACTL6A protein bands. Data information: In panel J RIP-qPCR data are presented as mean ± SD ($n = 3$ technical replicates). Source data are available online for this figure.

plasmid was verified by double-strand DNA sequencing. For the generation of stable *LINC00313* over-expressing HuCCT1 clones, cells were transfected with pcDNA3.1-LINC00313 for 48 h. Then, transfected cells were grown, for 2 weeks, in fresh selection medium, consisting of 10% FBS/RPMI, in the presence of 650 µg/ml hygromycin B Gold (Invivogen, ant-hg-1). By using limiting dilution assay, individual clones from the stable LINC00313 over-expressing pool were seeded in 96-well plates and grown in selection medium. The same protocol was followed, in order to establish stable HuCCT1 clones over-expressing an empty vector (pcDNA3.1), which served as control for gain-of-function experiments. Human HA-tagged ACTL6A plasmid (VB211129-1064fwh) was constructed by Vectorbuilder. M50 Super 8x TOPFlash (Addgene plasmid #12456) and M51 Super 8x FOPFlash (TOP-Flash mutant) (Addgene plasmid #12457) were a gift from Randall Moon. HuCCT1 or HEK293T cells were transfected with plasmid DNA, using Lipofectamine 2000 reagent (ThermoFisher Scientific, 11668019), according to the instructions by the manufacturer.

## Lentiviral transduction

The transduction of HuCCT1 cells with pGreenFire 2.0 TCF/LEF reporter virus (pGF2-TCF/LEF-rFluc-T2A-GFP-mPGK-Puro) (SBI System Biosciences, TR413VA-P) was performed using SureEN-TRY Transduction Reagent. Cells were transduced at a final multiplicity of infection (MOI) of 10. The next day medium was replaced with fresh complete medium and cells were incubated for another 24 h. Two days after transduction, cells were incubated with 1 µg/ml puromycin for two weeks for selection of transduced cells.

## siRNA transfections

HuCCT1 cells were seeded at 70% confluency and transfected with siRNAs at a final concentration of 50 nM. For single transfections with two different siRNAs, each siRNA was used a concentration of 25 nM. Similarly, for transfections with three different siRNAs simultaneously, each siRNA was used a concentration of 25 nM, so that the final siRNA concentration reached to 75 nM. Transient siRNA transfections were performed using Lipofectamine RNAi-MAX transfection reagent (ThermoFisher Scientific, 13778075), according to the manufacturer's instructions. Two custom-made siRNAs (Dharmacon, CTM-484556 and CTM-632606) were designed to target *LINC00313* (NR_026863.1). ON-TARGETplus human SMARTpool siRNAs were used for silencing *SMAD2*, *SMAD3*, *SMAD4*, *MAPK11*, *MAPK12*, *MAPK13*, *MAPK14* and

*ACTL6A*. For negative control transfections (siNC), an ON-TARGETplus Non-targeting Pool was used. A complete list of the siRNAs used in this study is provided in Appendix Table S1.

## ATAC sequencing

The ATAC-seq experiment was performed using the ATAC-seq service from Active Motif. The experiment was performed in biological simplicate per condition ($n = 1$). Briefly, 100,000 HuCCT1 pcDNA3.1 (empty vector) and pcLINC00313 over-expressing cells were used per ATAC reaction. Cells were harvested and frozen in culture media containing FBS and 5% DMSO. Cryopreserved cells were sent to Active Motif to perform the ATAC-seq assay. The cells were then thawed in a 37 °C water bath, pelleted, washed with cold PBS, and tagmented as previously described. Briefly, cell pellets were resuspended in lysis buffer, pelleted, and tagmented using the enzyme and buffer provided in the Nextera Library Prep Kit (Illumina). Tagmented DNA was then purified using the MinElute PCR purification kit (Qiagen), amplified with 10 cycles of PCR, and purified using Agencourt AMPure SPRI beads (Beckman Coulter). Resulting material was quantified using the KAPA Library Quantification Kit for Illumina platforms (KAPA Biosystems), and sequenced with PE42 sequencing on the NovaSeq 6000 sequencer (Illumina). Analysis of ATAC-seq data was performed as follows. Reads were aligned using the BWA algorithm (mem mode; default settings). Duplicate reads were removed, only reads mapping as matched pairs and only uniquely mapped reads (mapping quality >= 1) were used for further analysis. Alignments were extended in silico at their 3'-ends to a length of 200 bp and assigned to 32-nt bins along the genome. The resulting histograms (genomic "signal maps") were stored in bigWig files. Peaks were identified using the MACS 2.1.0 algorithm at a cutoff of $p$-value 1e−7, without control file, and with the –nomodel option. Peaks that were on the ENCODE blacklist of known false ChIP-Seq peaks were removed. Signal maps and peak locations were used as input data to Active Motifs proprietary analysis program, which creates Excel tables containing detailed information on sample comparison, peak metrics, peak locations and gene annotations. For differential analysis, reads were counted in all merged peak regions (using Subread), and the replicates for each condition were compared using DESeq2.

## RNA sequencing

Total RNA from HuCCT1 pcDNA3.1 (empty vector) and pcLINC00313 over-expressing clones was isolated using the

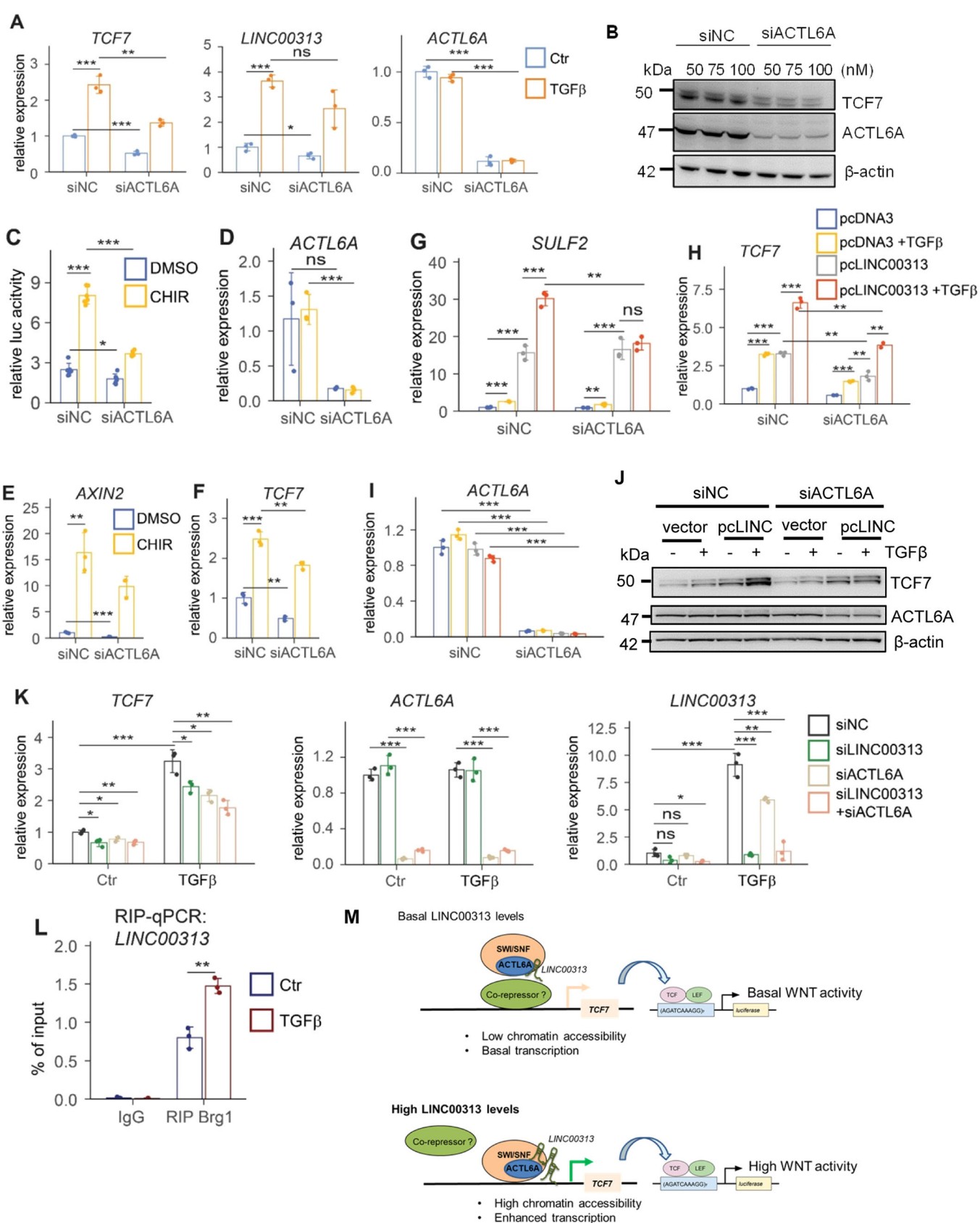

**Figure 7. ACTL6A silencing reduces TCF/LEF-mediated gene expression.**

(A) *TCF7*, *LINC00313* and *ACTL6A* mRNA levels in HuCCT1 transfected with siACTL6A or siNC and in the presence or not of TGFβ. (B) Immunoblotting for TCF7, ACTL6A and β-actin, upon silencing ACTL6A, using the indicated siRNA concentrations. (C) TCF/LEF luciferase reporter assay in siACTL6A or siNC HuCCT1 cells treated with CHIR99021 or DMSO for 24 h. (D–F) qPCR analysis of *ACTL6A* (D), *AXIN2* (E) and *TCF7* (F) mRNAs in the same conditions as these of panel (C). (G–I) qPCR analysis of *SULF2* (G), *TCF7* (H) and *ACTL6A* (I) mRNA levels in pcDNA3 or *LINC00313* over-expressing HuCCT1 transiently transfected with siACTL6A or siNC with or without TGFβ. (J) Immunoblotting for TCF7, ACTL6A and β-actin in the same conditions as these of panels G-I. (K) qPCR analysis of *TCF7*, *ACTL6A* and *LINC00313* RNA levels in HuCCT1 cells transiently transfected with siRNAs against *LINC00313* or *ACTL6A* individually or simultaneously with both siRNAs and treated or not with TGFβ for 16 h. (L) RIP assay for endogenous Brg1 followed by qPCR for *LINC00313* in HuCCT1 cells treated with TGFβ or BSA/HCl (vehicle control) for 16 h. (M) Proposed model for the molecular mechanism of *LINC00313*. Data information: In panels (A), (D–I) and (K) qPCR data are presented as mean ± SD (*n* = 3 biological replicates). In panel (C), luciferase assay data are presented as mean ± SD (*n* = 6 biological replicates). In panel (L), RIP-qPCR data are presented as mean ± SD (*n* = 3 technical replicates). *$P \leq 0.05$, **$P \leq 0.01$, ***$P \leq 0.001$ (Student's *t*-test). Source data are available online for this figure.

miRNeasy kit (Qiagen, 217004). RNA-seq was performed by Active Motif. The experiment was performed in biological triplicates per condition (*n* = 3). For each sample, 2 µg of total RNA was then used in Illumina's TruSeq Stranded mRNA Library kit (Cat# 20020594). Libraries were sequenced on Illumina NextSeq 500 as paired-end 42-nt reads. Sequence reads were analysed with the STAR alignment. The data analysis was performed as follows:

### Read mapping

The paired-end 42 bp sequencing reads (PE42) generated by Illumina sequencing (using NextSeq 500) are mapped to the genome using the STAR algorithm with default settings.

### Fragment assignment

The numbers of fragments overlapping predefined genomic features of interest (e.g. genes) are counted. Only read pairs that have both ends aligned are counted. Read pairs that have their two ends mapping to different chromosomes or mapping to same chromosome but on different strands are discarded. The gene annotations used were obtained from Subread package. These annotations were originally from NCBI RefSeq database and then adapted by merging overlapping exons from the same gene to form a set of disjoint exons for each gene. Genes with the same Entrez gene identifiers were also merged into one gene.

### Differential analysis

After obtaining the gene table containing the fragment counts of genes, we perform differential analyses to identify statistically significant differential genes using DESeq2. The following lists the pre-processing steps before differential calling. a. Data Normalization: DESeq2 expects un-normalized count matrix of sequencing fragments. The DESeq2 model internally corrects for library size using their median-ofratios method. The gene table obtained from Analysis Step 2) is used as input to perform the DESeq2's differential test. b. Filtering before multiple testing adjustment: After a differential test has been applied to each gene except the ones with zero counts, the *p*-value of each gene is calculated and adjusted to control the number of false positives among all discoveries at a proper level. This procedure is known as multiple testing adjustment. During this process, DESeq2 by default filters out statistical tests (i.e. genes) that have low counts by a statistical technique called independent filtering. It uses the average counts of each gene (i.e. baseMean), across all samples, as its filter criterion, and it omits all genes with average normalized counts below a filtering threshold from multiple testing adjustment. This filtering threshold is automatically determined to maximize detection power

(i.e. maximize the number of differential genes detected) at a specified false discovery rate (FDR).

## cDNA synthesis and real-time qPCR

Total RNA was isolated using the miRNeasy kit (Qiagen, 217004). RNA concentration was measured using a NanoDrop One/OneC Microvolume UV-Vis Spectrophotometer (ThermoFisher Scientific, ND-ONE-W). Reverse transcription was performed using SuperScript™ III Reverse Transcriptase (ThermoFisher Scientific, 18080-044) and 0.5 or 1 µg total RNA as a template with both oligo-dT (250 ng) and random hexamers (100 ng), unless stated otherwise. Quantitative-RT-PCR was performed by using a SYBR Green master mix (Applied Biosystems, Carlsbad, CA). Normalization of gene expression was performed by using the ΔΔCt method and statistical analysis by *t* testing. *TATA-box binding protein* (*TBP*) and *glyceraldehyde-3-phosphate dehydrogenase* (*GAPDH*) were used as reference genes for normalization. The list of DNA primers used in this study is provided in Appendix Table S2.

## Nucleo-cytoplasmic fractionation

Nucleo-cytoplasmic fractionation to evaluate subcellular localization of different RNAs was performed using the PARIS kit (Ambion, ThermoFisher Scientific, AM1921) based on the manufacturer's instructions. Briefly, HuCCT1 cells were trypsinized, pelleted and washed in PBS. Then, cells were gently resuspended in Cell Fractionation Buffer and incubated on ice for 10 min. Cell extracts were centrifuged at $500 \times g$ at 4 °C for 5 min and the cytoplasmic fraction was transferred to new tubes. The remaining nuclear pellet was washed once in Cell Fractionation Buffer and lysed in ice-cold Cell Disruption Buffer. RNA was isolated by adding a $2 \times$ Lysis/Binding solution to each fraction, followed by addition of 100% ethanol and capture of the RNA by a filter cartridge. After a series of washes, the RNA was eluted in pre-heated Elution solution (PARIS kit) and stored at −80 °C. For nucleo-cytoplasmic fractionation followed by protein extraction and immunoblotting we utilized the following protocol with some modifications (Gagnon et al, 2014). HuCCT1 pcDNA3 and pcLINC00313 expressing cells were grown in T150 flasks, treated with CHIR or DMSO for 24 h, until they were 80–90% confluent, then washed in ice-cold PBS, scraped gently and centrifuged in 1500 rpm at 4 °C for 3 min. Then, cells were resuspended in PBS and centrifuged again in 800 rcf at 4 °C for 3 min. Cell pellets were resuspended in hypotonic lysis buffer (10 mM Tris, pH 7.4, 10 mM

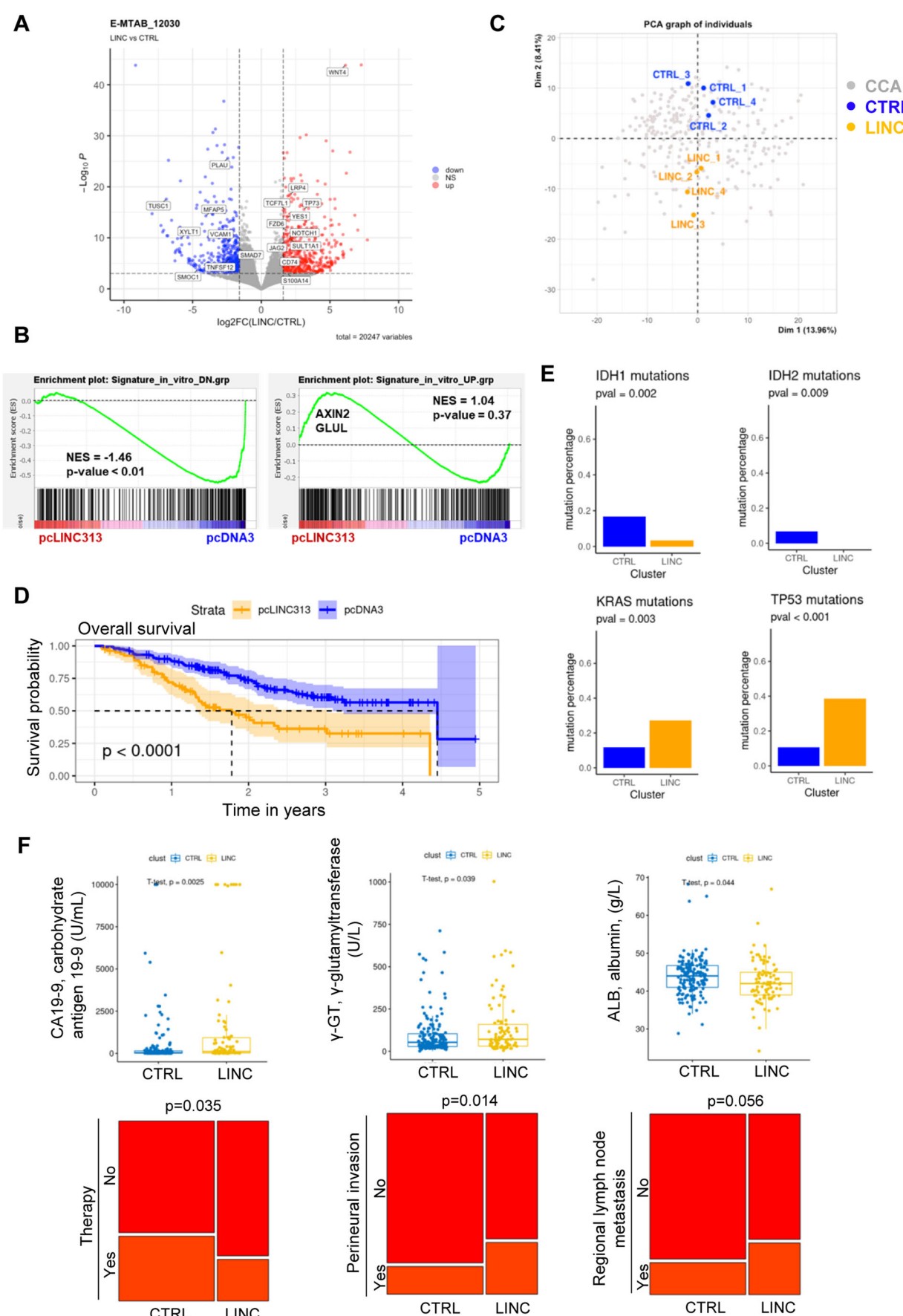

**Figure 8.   In vivo *LINC00313*-associated gene signature predicts outcome in CCA patients.**

(A) Volcano plot showing differentially expressed genes in control ($n = 4$) and LINC00313 ($n = 4$) resected xenograft tumours. Genes significantly differentially expressed between the two conditions (fold change > 3 and FDR < 0.001) are indicated in red (upregulated) and blue (downregulated), respectively. (B) GSEA using gene signatures established in vitro (UP and DN regulated genes) and the gene expression profiles established in vivo (i.e. in tumours from xenografted mice derived from HuCCT1 cells overexpressing pcLINC313 versus pcDNA3 control vector). (C) Principal component analysis (PCA) of 255 iCCA samples from the publicly available from the NODE Project=OEP001105). The four control (CTRL) and LINC00313 (LINC) samples were integrated as supplementary individuals in the PCA analysis (*FactoMineR* R package). (D) Comparison of the overall survival (OS) between CCA patients in CTRL (pcDNA3, $n = 154$) and LINC (pcLINC00313, $n = 101$) clusters based on HCPC. The OS was higher in CTRL patients ($n = 149$) than in LINC patients ($n = 95$) with 4.36 years (95% CI, 4,36 to not reached) versus 1.83 years (95% CI, 1.39 to not reached), respectively; $P = 2e-05$ (E) Mutational analysis performed on 249 CCA samples from the publicly available OEX011131 NODE WES dataset (NODE Project = OEP001105). *IDH1* mutated patients are more frequent in CTRL versus LINC ($n = 26/149$, $n = 4/100$, $p = 1.22e-03$). *IDH2* mutated patients are more frequent in CTRL versus LINC ($n = 11/149$, $n = 0/100$, $p = 3.61e-03$). *KRAS* mutated patient are less frequent in CTRL versus LINC ($n = 14/149$, $n = 29/100$, $p = 1.22e-04$). *TP53* mutated patient are less frequent in CTRL versus LINC ($n = 16/149$, $n = 35/100$, $p = 5.05e-06$). (F) Clinical data analysis between CTRL ($n = 154$) and LINC ($n = 101$) clusters from OEP001105 NODE dataset ($n = 255$). Box plots are defined as follows: For CA19-9 in CTRL group; min: 0.6, lower whisker: 0.6, 25th percentile: 12.6, median: 32.6, mean: 349.9, 75th percentile: 109.9, upper whisker: 229.8, max: 10,000. For CA19-9 in LINC group; min: 0.6, lower whisker: 0.6, 25th percentile: 26.2, median: 125.8, mean: 1437.4, 75th percentile: 928.2, upper whisker: 2278, max: 10,000. For γ-GT in CTRL group; min: 1.6, lower whisker: 1.6, 25th percentile: 26.0, median: 49.0, mean: 91.1, 75th percentile: 97.0, upper whisker: 195.0, max: 712.0. For γ-GT in LINC group; min: 5.2, lower whisker: 5.2, 25th percentile: 35.0, median: 80.0, mean: 140.7, 75th percentile: 167.0, upper whisker: 364.0, max: 1003.0. For ALB in CTRL group; min: 29, lower whisker: 34, 25th percentile: 41, median: 44, mean: 43.9, 75th percentile: 46, upper whisker: 51, max: 68. For ALB in LINC group; min: 24, lower whisker: 33, 25th percentile: 40, median: 42, mean: 42.6, 75th percentile: 45, upper whisker: 52, max: 67. Data information: In panel (A) the volcano plot was created based on a differential analysis (LINC vs CTRL) performed on 20,247 genes with DESeq2 R package. Normalisation of the expression data was performed by variance stabilizing transformation (VST). For calculating the false discovery rate (FDR) the *p*-values adjustment method was BH (Benjamini and Hochberg, 1995). In panel (B) the Kolmogorov–Smirnov test was used for statistical analysis. In panels (C–F) two groups of CCA samples were defined by HCPC (Hierarchical Clustering on Principal Components) performed with the FactoMineR R package on the PCA analysis results. Groups were labelled CTRL and LINC in accordance with the projected supplementary individuals. In panel (D) the Kaplan–Meier method was used to estimate the survival distributions. Log-rank tests were used to test the difference between survival groups. Analyses were carried out with the survival and survivalROC R packages. 11 observations were deleted due to missingness. In panel (E) comparison of the mutation percentages between CTRL and LINC patients was assessed with Fisher's exact tests. In panel (F) for quantitative variables, the difference in means was assessed with two sample *t*-tests, whereas for qualitative variables, the independence was assessed with Fisher's exact tests. Source data are available online for this figure.

NaCl, 3 mM MgCl$_2$, 0.3% NP-40, 10% glycerol) and incubated on ice for 15 min. After a mild centrifugation the supernatant consisting of the cytoplasmic fraction was collected. The pellet, representing the nuclei was washed three times in hypotonic buffer with subsequent centrifugation and nuclear lysis buffer (20 mM Tris, pH 7.4, 150 mM KCl, 3 mM MgCl$_2$, 0.3% NP-40, 10% glycerol) was added to disrupt the nuclei. Nuclear extracts were sonicated and then both nuclear and cytoplasmic extracts were cleared by centrifugation at maximum speed and protein fractions were kept frozen until used for SDS-PAGE.

### Gene expression profiling

Total RNA was purified with an miRNeasy kit (Qiagen, 217004). Genome-wide expression profiling was performed using the low-input QuickAmp labelling kit and human SurePrint G3 8×60K pangenomic microarrays (Agilent Technologies, Santa Clara, USA) as previously described (Merdrignac et al, 2018). Differentially expressed genes were identified by a 2-sample univariate *t* test and a random variance model as previously described. Differentially regulated genes between the different experimental conditions were selected based on stringent criteria, e.g., *P* value ($P < 0.001$) and fold change (FC > 2).

### In vitro RNA transcription

In vitro transcription of *LINC00313* or firefly luciferase (F-luc) mRNA was performed using the HiScribe™ T7 High Yield RNA Synthesis kit (New England Biolabs, E2040S), according to the instructions by the manufacturer. Briefly, the pcDNA3.1-LINC00313 plasmid vector was linearized with single digestion, downstream of the insert site and the purified linearized plasmid

was used as a template for in vitro transcription at 37 °C for 2 h. Residual DNA was degraded using RNase-free DNase I (New England Biolabs, M0303) and the in vitro synthesized RNA was purified using the RNA clean-up protocol from RNeasy Mini kit (Qiagen, 74104). The size and the integrity of the in vitro transcribed RNAs were verified using agarose gel electrophoresis.

### RNA labelling and RNA pull-down

The in vitro transcribed RNAs were labelled at the 3′ terminus with a desthiobiotinylated cytidine biphosphate nucleotide, using the Pierce RNA 3′ End Desthiobiotinylation kit (Pierce/Thermo Fisher Scientific, 20163), and were purified using the RNA clean-up protocol (Qiagen). RNA pull-down assays were performed following the Pierce Magnetic RNA-Protein Pull-Down kit protocol (Pierce/Thermo Fisher Scientific, 20164). Biotinylated RNAs were pre-coupled to Nucleic Acid Compatible Streptavidin Magnetic Beads in RNA Capture Buffer (Pierce/Thermo Fisher Scientific) for 3 h at 4 °C. The RNA-beads complex was incubated with protein extracts from HuCCT1 cells, in the presence of Halt protease inhibitor cocktail (100X) (Thermo Fisher Scientific, 87786) and RNase inhibitor (Superase In, Ambion, Thermo Fisher Scientific, AM2694) for 1 h at 4 °C, with rotation. Next, the protein-RNA-beads complexes were washed and proteins were eluted in Elution Buffer (Pierce/Thermo Fisher Scientific, Stockholm, Sweden), boiled at 95 °C for 5 min and subjected to SDS-PAGE, followed by immunoblotting for the detection of specific protein interactors. For the identification of *LINC00313*-interacting proteins the same protocol was applied followed by an unbiased proteomic analysis. In the latter case, the ribonucleoprotein complexes bound to beads were washed several times before being dissociated, using on-beads digestion and prepared for mass spectrometry analysis.

## Mass spectrometry

The analysis was performed by the Clinical Proteomics Mass Spectrometry facility (Karolinska Institutet, Karolinska University Hospital, Science for Life Laboratory, Stockholm, Sweden), as previously described (Papoutsoglou et al, 2019b, 2). The following detailed protocol was used:

### IP-MS

The bead-bound proteins were reduced in 1 mM DTT at room temperature for 30 min. Next, samples were alkylated by incubation in the dark at room temperature in 5 mM Iodoacetamide for 20 min. After incubation the remaining iodoacetamide was quenched by the addition of 4 mM DTT. Digestion was carried out by the addition of 1 µg Trypsin (sequencing grade modified, Pierce) and over-night incubation at 37 °C. The next day the supernatant was collected and cleaned by a modified sp3 protocol (Moggridge et al, 2018). Briefly, 20 µl Sera-Mag SP3 bead mix (10 µg/µl) was added to the sample. Next, 100% acetonitrile was added to achieve a final concentration of >95%. Samples were pipette-mixed and incubated for 8 min at room temperature and then placed on a magnetic rack. The supernatant was aspirated and discarded and the beads were washed in 180 µl of acetonitrile. Samples were removed from the magnetic rack and beads were reconstituted in 20 µl of (3% Acetonitrile, 0,1% formic acid) solution, followed by 1 min of sonication. Then the beads were placed on a magnetic rack again and the supernatant was recovered and transferred to an MS-vial.

### LC-ESI-MS/MS

Q-Exactive Online LC-MS was performed using a Dionex Ultimate™ 3000 RSLCnano System coupled to a Q-Exactive mass spectrometer (Thermo Scientific). 5 µL was injected from each sample. Samples were trapped on a C18 guard desalting column (Acclaim PepMap 100, 75 µm × 2 cm, nanoViper, C18, 5 µm, 100 Å), and separated on a 50 cm long C18 column (Easy spray PepMap RSLC, C18, 2 µm, 100 Å, 75 µm × 50 cm). The nano capillary solvent A was 95% water, 5%DMSO, 0.1% formic acid; and solvent B was 5% water, 5% DMSO, 95% acetonitrile, 0.1% formic acid. At a constant flow of 0.25 µl min$^{-1}$, the curved gradient went from 6%B up to 43%B in 180 min, followed by a steep increase to 100%B in 5 min.

FTMS master scans with 60,000 resolution (and mass range 300–1500 *m/z*) were followed by data-dependent MS/MS (30,000 resolution) on the top 5 ions using higher energy collision dissociation (HCD) at 30% normalized collision energy. Precursors were isolated with a 2 *m/z* window. Automatic gain control (AGC) targets were 1e6 for MS1 and 1e5 for MS2. Maximum injection times were 100 ms for MS1 and MS2. The entire duty cycle lasted ~2.5 s. Dynamic exclusion was used with 60 s duration. Precursors with unassigned charge state or charge state 1 were excluded. An underfill ratio of 1% was used.

### Peptide and protein identification

The MS raw files were searched using Sequest-Percolator or Target Decoy PSM Validator under the software platform Proteome Discoverer 1.4 (Thermo Scientific) against Homo sapiens database (Uniprot) and filtered to a 1% FDR cut off. We used a precursor ion mass tolerance of 10 ppm, and product ion mass tolerances of 0.02 Da for HCD-FTMS. The algorithm considered tryptic peptides with maximum 2 missed cleavage; carbamidomethylation (C) as fixed modifications and oxidation (M), as variable modifications.

## RNA immunoprecipitation (RIP)

RIP was performed according to the Magna-RIP™ RNA-binding protein immunoprecipitation kit (Millipore/Merck, 17-700) as previously described (Papoutsoglou et al, 2019b), with minor modifications. Briefly, cells were scraped in PBS, centrifuged and re-suspended in RIP lysis buffer. After short incubation on ice, the lysates were stored at −80 °C. Magnetic beads carrying protein A/G were incubated with anti-ACTL6A (Cell Signaling technologies, #76682) at 1:50 dilution or anti-Brg1 (Cell Signaling technologies, #49360) at 1:100 dilution or a negative control normal rabbit IgG (Millipore/Merck, # PP64B) and lysates in RIP immunoprecipitation buffer at 4 °C, overnight, without previous pre-coupling between antibody and beads. A part (10%) of the lysate from each sample was used as an input for normalization. The next day, the RNA-protein-bead complexes were washed in RIP Wash Buffer (Millipore/Merck) and split into two fractions: from the major fraction, bound RNA, together with total RNA from input samples was purified using the RNA cleanup protocol from the RNeasy kit (Qiagen, 74104); from the minor fraction, washed lysate was loaded on polyacrylamide gels for immunoblotting using the same antibody with which the RNA immunoprecipitation was performed and described above. Using RNA purified from the major fraction, cDNA was generated from the isolated RNA and subjected to real-time qPCR.

## Immunoblotting

Total proteins were extracted using RIPA buffer (ThermoFisher Scientific, 89901) supplemented with protease and phosphatase inhibitor cocktails (ThermoFisher Scientific, 13393126). Cell lysates were briefly sonicated and cleared by centrifugation (10,000 × *g*, 15 min). Protein concentration was determined, using the Pierce bicinchoninic acid (BCA) Protein Assay kit (ThermoFisher Scientific, 23227). Then, 4X NuPAGE LDS sample buffer (ThermoFisher Scientific, NP0007), supplemented with 10X NuPAGE reducing agent (ThermoFisher Scientific, NP0009) was added to the lysates, which were then boiled at 70 °C for 10 min. Protein samples (20 or 40 µg) were loaded in NuPAGE™ Novex™ 4–12% Bis-Tris Protein Gels, 1.0 mm, 10-well (ThermoFisher Scientific, NP0321) and the resolved proteins were transferred to a nitrocellulose filter using the iBlot Dry Blotting System (Invitrogen, IB1001). Then, the filters were blocked in 5% BSA/Tris-buffered saline (TBS) containing 0.1% Tween-20 or 5% non-fat dry milk (Cell Signaling Technology, 9999S) in TBS-Tween-20 and incubated with primary antibody solutions (overnight, 4 °C). Anti-rabbit (Cell Signaling Technology, #7074) or anti-mouse (Cell Signaling Technology, #7076) horseradish peroxidase-conjugated secondary antibodies were incubated with membranes for 1 h at room temperature and blots were developed, using enhanced chemiluminescence (ECL) assays (Cytiva,). Images were taken using an ImageQuant LAS 4000 imager (GE Healthcare). A list of the primary antibodies used in this study is provided in Appendix Table S3.

## Immunofluorescence

Empty vector or LINC00313 overexpressing HuCCCT1 cells were seeded in 96-well plates and fixed in 4% PFA for 15 min at room temperature. After three washes in PBS, cells were blocked in 5% FBS/PBS, supplemented with 0.3% Triton X-100 for 1 h at room temperature. Then, samples were incubated with primary antibodies (anti-β-catenin, anti-ACTL6A) in 1% BSA/PBS solution overnight at 4 °C. The next day, samples were washed three times in PBS and Dylight 488-conjugated goat anti-rabbit secondary antibody (Insight Biotechnology, 5230-0385) in 1% BSA/PBS solution was added to the cells together with Hoechst stain for nuclear staining. Images were acquired with an EVOS M5000 Imaging System (ThermoFisher Scientific, AMF5000).

## Single molecule inexpensive FISH (smiFISH)

smiFISH was performed according to a previously established protocol (Tsanov et al, 2016). Briefly, 80,000 cells were seeded in coverslips and the day after were starved and treated with TGFβ for 16 h. Then, cells were washed twice in PBS and fixed in 4% PFA for 20 min. After that, cells were washed twice in PBS and permealized and stored in ice-cold 70% ethanol at 4 °C. The next day in situ hybridization was performed (37 °C, overnight), using a specific LINC00313 probe set, consisting of 14 probes (listed in Appendix Table S2) and designed using Oligostan. Finally, the next day cells were washed twice in formamide-containing buffer and mounted in vectashield mounting medium, containing DAPI. Images were obtained using a Zeiss Axioimager Z1/Apotome microscope.

## Luciferase assays

For TCF/LEF-luciferase reporter assays, HuCCT1 cells stably expressing pGreenFire 2.0 TCF/LEF reporter virus (pGF2-TCF/LEF-rFluc-T2A-GFP-mPGK-Puro) (System Biosciences, TR413VA-P) were silenced, over-expressed or treated with inhibitors or TGFβ as described in the figure legends. Luciferase activity was normalized to the number of viable cells measured by a PrestoBlue cell viability reagent (ThermoFisher Scientific, A13261). When using TOPFlash or FOPFlash for TCF/LEF-luciferase reporter assays, cells were transiently transfected with each one of TOPFlash or FOPFlash together with pGL4.74 [hRluc/TK] Vector, expressing Renilla luciferase. Firefly luciferase activity was normalized to Renilla luciferase activity using the Dual-Luciferase Reporter Assay System (Promega, E1910). Luciferase activity measurements were performed using a TECAN Spark multimode microplate reader.

## Functional tests

Cell viability was measured using a PrestoBlue reagent (Thermo-Fisher Scientific, A13261) according to the manufacturer's instructions. Wound healing assays were performed at different time points (0, 8, 24 h) using silicone inserts with a defined cell-free gap (Ibidi, 80366). After the removal of the insert, cells were grown in 0.1% FBS/RPMI, supplemented with 10 μg/ml mitomycin C (Sigma-Aldrich, M4287), in order to stop their proliferation. Wound closure was evaluated using the ImageJ software. For colony formation assays, 800 HuCCT1 cells were seeded in six-well

plates and cultured for 11 days. Colonies were washed in PBS twice and fixed in 4% paraformaldehyde (PFA) for 15 min. Then, colonies were stained with 0.1% crystal violet and counted manually. Each experiment was performed in biological triplicates. For proliferation assays, 5000 cells were seeded per well (in a 96-well plate). Then the plate was placed in the Incucyte and pictures of the wells were taken every 4 h during five days. Analysis of the results was performed using the Incucyte software, with the tool "IA confluence" that calculates the confluence in each well. Migration assays were performed according to the protocol of Sartorius. 30,000 cells were seeded per well (treated or not with TGFβ) in a 96-well plate provided by Sartorius (Incucyte ImageLock Plate BA-04855). Pictures were taken every 2 h during 48 h. Data analysis was performed with the Incucyte software using the "Scratch wound" tool (which quantify the dynamics of migration or invasion into a cell free zone). For invasion assays, 30,000 cells were seeded per well in a 96-well plate provided by Sartorius (Incucyte ImageLock Plate BA-04855). Matrigel was prepared on ice and polymerized directly in an incubator at 37 °C. Pictures were taken every two hours during 96 h. Data analysis was also done with the Incucyte software using the "Scratch wound" tool (which quantify the dynamics of migration or invasion into a cell free zone).

## CUT&RUN

The CUT&RUN experiment was performed following the kit protocol from Cell Signaling Technology (#86652) in simplicates. Briefly, 250,000 cells were seeded in 6-well plate and when they reached 80–90% confluency, they were trypsinized, pelleted and washed in 1x Wash buffer: Then, they were subjected to light fixation, using 0.1% formaldehyde for 2 min at room temperature and fixed cell pellets were stored in −80 °C. The next day, concavalin A beads were prepared and together with Brg1 antibody (used at 1:50 dilution) were added in cell suspension and incubated at 4 °C, overnight. The next day the antibody/beads/cell suspension was washed in digitonin buffer and pAG-MNase enzyme was added and incubated at 4 °C for 1 h. After two washes in digitonin buffer, pAG-MNase was activated using calcium chloride to digest DNA. Input DNA fragmentation was also performed in parallel, using MNase. DNA fragments were purified using spin columns (Cell Signaling Technology, #14209) and isolated DNA was stored in −20 °C until used for qPCR.

## In vivo mouse xenograft experiment

HuCCT1 cells over-expressing empty vector (pcDNA3.1) or LINC00313 (pcLINC00313) were infected with GL261-Luc (CMV-Firefly luciferase lentivirus (Neo), PLV-10064-50, Cellomics Technology, USA). 4,000,000 cells (100 μL in PBS + 100 μL matrigel) were implanted on the flanks of immunodeficient NSG (*NOD Cg-Prkdcscid Il2rgtm1Wjl/SzJ*) 8-week-old male mice (Charles River, USA). All animal procedures met the European Community Directive guidelines (Agreement B35-238-40, Biosit Rennes, France; DIR #7163) and were approved by the local ethics committee and ensuring the breeding and the daily monitoring of the animals in the best conditions of well-being according to the law and the 3R rule (Reduce-Refine-Replace). Tumour growth was evaluated by a direct measurement of tumour size using caliper. The presence of lung and liver metastases was evaluated using

bioluminescence. Mice were sacrificed 77 days after implantation and tumour, liver and lung tissues were isolated and subjected to molecular analysis.

## RNA sequencing and immunoblotting from mouse xenograft tumours

Xenograft tumours were cut into smaller pieces in a Petri dish placed on ice and transferred into 2 mL tubes containing 2.8 mm ceramic beads (P000911-LYSK0-A, Bertin Corp) filled with 1 ml Qiazol or 1 ml RIPA lysis buffer, supplemented with protease and phosphatase inhibitors. Homogenization of the tumours was performed using the Precellys 24 tissue homogenizer (P000669-PR240-A, Bertin Instruments) at 5500-rpm speed, 3 cycles of 20 s each with a pause of 20 s, between the cycles. Then, the homogenates were transferred to new tubes and subjected to RNA extraction using the RNeasy Lipid Tissue Mini Kit (74804, Qiagen), according to the instructions by the manufacturer or protein extraction. RNA-seq was performed by BGI Genomics (Hong Kong), using the DNBSEQ Eukaryotic Strand-specific mRNA library. The library construction method and sequencing process were carried out according to the following steps:

1. Total RNA samples were enriched in mRNA with poly A tail using oligo dT beads.
2. mRNA molecules were fragmented into small pieces.
3. The fragmented mRNA was synthesized into first strand cDNA using random primers.
4. The second strand cDNA was synthesized with dUTP instead of dTTP.
5. The synthesized cDNA was subjected to end-repair and 3' adenylated. Adaptors were ligated to the ends of these 3' adenylated cDNA fragments.
6. Digest the U-labelled second-strand template with Uracil-DNA-Glycosylase (UDG) and perform PCR amplification.
7. Library quality control.
8. Library circularization.
9. The library was amplified to make DNA nanoball (DNB).
10. Sequencing on DNBSEQ (DNBSEQ Technology) platform.

Initial bioinformatics analysis and parameters for data filtering:
Raw data with adapter sequences or low-quality sequences was filtered. We first went through a series of data processing to remove contamination and obtain valid data. This step was completed by SOAPnuke software (SOAPnuke software filter parameters: " -n 0.001 -l 20 -q 0.4 --adaMR 0.25 --ada_trim") developed by BGI.
Steps of filtering:

1. Filter adapter: if the sequencing read matches 25.0% or more of the adapter sequence (maximum 2 base mismatches are allowed), cut the adapter;
2. Filter low-quality data: if the bases with a quality value of less than 20 in the sequencing read account for 40.0% or more of the entire read, discard the entire read;
3. Remove N: if the N content in the sequencing read accounts for 0.1% or more of the entire read, discard the entire read;
4. Obtain Clean reads: the output read quality value system is set to Phred+33.

## Integrative analysis

We used intrahepatic cholangiocarcinoma RNA-Seq data Data Ref: (Dong et al, 2022) publicly available in the NODE database under the accession number OEP001105. Raw counts were normalised with the DESeq2 and scaled. We then performed a Principal Component Analysis (PCA) on the OEP001105 normalised data with samples from our study as supplementary individuals (datasets were merged based on the gene symbols of the signature). We used an agglomerative hierarchical clustering on the PCA components (Hierarchical Clustering on Principle Components) to divide the samples from the OEP001105 dataset into two groups: CTRL (pcDNA3) or LINC (pcLINC00313). Overall survival (OS) was estimated using the Kaplan–Meier method. Comparison between survival groups was performed using a log-rank test for binary variables. All the analyses were conducted with the following R packages: DESeq2, FactoMineR and survival.

## TCGA analysis

Data used for TCGA analysis were obtained from the Xena Functional Genomics Explorer (https://xenabrowser.net/) (for the data presented in Fig. EV5A), from GEPIA2 (http://gepia2.cancer-pku.cn/#index) (for the data presented in Fig. EV5B) and from cbioportal for Cancer Genomics (https://www.cbioportal.org/) (for the data presented in Fig. EV1C). The GCD TCGA Bile Duct Cancer (CHOL) cohort was used to generate the Kaplan Meier plot based on LINC00313 expression (high vs low). For comparing LINC00313, ACTL6A, TCF7, WNT5A, AXIN2 and SULF2 expression between cholangiocarcinoma patients and normal tissues the CHOL dataset from GEPIA2 was used. For the analysis of LINC00313 expression in multiple cancers, TCGA PanCancer Atlas studies were selected representing 29 different cancer types.

## Statistical analysis

Comparisons were performed using a two-tailed paired Student's $t$ test or a Mann–Whitney $U$ test (*$P < 0.05$, **$P < 0.01$, ***$P < 0.001$). The results are shown as mean ± standard deviation from three biological experiments, unless stated otherwise in the figure legends.

# Data availability

The datasets produced in this study are available in the following databases:

RNA-seq data: ArrayExpress E-MTAB-12030 RNA-seq data: https://www.ebi.ac.uk/biostudies/arrayexpress/studies/E-MTAB-12030?key=6905955f-420a-4002-ad36-5a5655b0cb65

ATAC-seq data: https://www.ebi.ac.uk/biostudies/arrayexpress/studies/E-MTAB-12879?key=7fc6a860-07ac-4c45-a060-19de0d84f1d9

Visualization of ATAC-seq peaks: UCSC Genome Browser (https://genome.ucsc.edu/s/ppapouts/ATAC%2Dseq%20HuCCT1%20LINC00313%20overexpression)

## Peer review information

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

inflammatory factors for lung diseases. J Transl Med 13:273

## Acknowledgements
We thank Dr Yutaro Tsubakihara and Prof. Aristidis Moustakas (Uppsala
University, Sweden) for providing the pcDNA3.1/Hygro (+) and the pGL4.74
TK-Renilla luciferase reporter expression vectors. We thank Dr Laura Fouassier
for providing CCA cell lines and for helpful discussion. We also thank Dr
Clotilde Théry's team (Institut Curie) for the antibody against 14-3-3 protein.
Stéphanie Lhomond, Gaëlle Angenard, Léa Chailloux, Matthis Desoteux and
Giorgia Ianese Regin are acknowledged for their technical help. The authors are
supported by Inserm, Université de Rennes, Région Bretagne, Ministère de
l'Enseignement Supérieur de la Recherche et de l'Innovation, Ligue Nationale
Contre le Cancer, Ligue Contre le Cancer (CD22, CD29, CD35, CD44, CD85,
R22026NN, R21011NN), Fondation ARC (R21043NN), INCa (EU
TRANSCAN23-002-2023-129, INCa_18688) and ITMO Cancer of AVIESAN
within the framework of the 2021-2030 Cancer Control Strategy, on funds
administered by Inserm (Equipment and Non-coding RNA in cancerology:
fundamental to translational) (C18007NS, C20013NS, C20014NS). This work
was also supported by a grant from the French Ministry of Health and the
French National Cancer Institute, PRT-K20-136, CHU Rennes, CLCC Eugène
Marquis, Rennes. JMB is supported by Spanish Carlos III Health Institute
(ISCIII) [(FIS PI18/01075, PI21/00922, and Miguel Servet Programme CPII19/
00008) cofinanced by "European Union"] and CIBERehd (ISCIII); La Caixa
Scientific Foundation (HR17-00601); AMMF-The Cholangiocarcinoma Charity
(EU/2019/AMMFt/001); PSC Partners US and PSC Supports UK (06119JB);
European Union's Horizon 2020 Research and Innovation Program [grant
number 825510, ESCALON]. We thank the Onassis Scholars Association for
covering the publication cost of this article through a special grant awarded to
PP, via the Academic Article Publication Support Program.

## Author contributions
**Panagiotis Papoutsoglou**: Conceptualization; Data curation; Formal analysis;
Validation; Investigation; Visualization; Methodology; Writing—original draft;
Project administration; Writing—review and editing. **Raphael Pineau**:
Resources; Formal analysis; Investigation; Methodology. **Raffaële Leroux**:
Investigation; Writing—review and editing. **Corentin Louis**: Formal analysis;
Investigation. **Anaïs L'Haridon**: Data curation; Software. **Dominika Foretek**:
Investigation; Methodology; Writing—review and editing. **Antonin Morillon**:
Resources. **Jesús M Banales**: Resources; Funding acquisition; Writing—review
and editing. **David Gilot**: Funding acquisition; Writing—review and editing.
**Marc Aubry**: Data curation; Software; Formal analysis; Investigation;
Visualization; Writing—review and editing. **Cedric Coulouarn**:
Conceptualization; Resources; Data curation; Supervision; Funding acquisition;
Validation; Visualization; Methodology; Writing—original draft; Project
administration; Writing—review and editing.

## Disclosure and competing interests statement
The authors declare no competing interests.

# Expanded View Figures

**Figure EV1.  Characterization of LINC00313.**

(**A**) Schematic representation of LINC00313 genomic locus on chromosome 21. (**B**) LINC00313 expression profile among 27 different human tissues from 95 individuals, as determined by RNA-seq (accession: PRJEB4337). (**C**) LINC00313 expression levels among 29 cancer types from the PanCancer Atlas of TCGA, sorted from the highest (top) to the lowest (bottom) expression. (**D**) ORF analysis of LINC00313 lncRNA sequence, using the Open Reading Frame Finder tool from NCBI. (**E**) Snapshot of the LINC00313 genomic locus from the Ensembl database showing the different splice variants. Data information: In panel (**B**) LINC00313 expression is presented in Reads Per Kilobase per Million mapped reads. In panel (**C**) LINC00313 expression is quantified with RSEM and displayed in logarithmic scale.

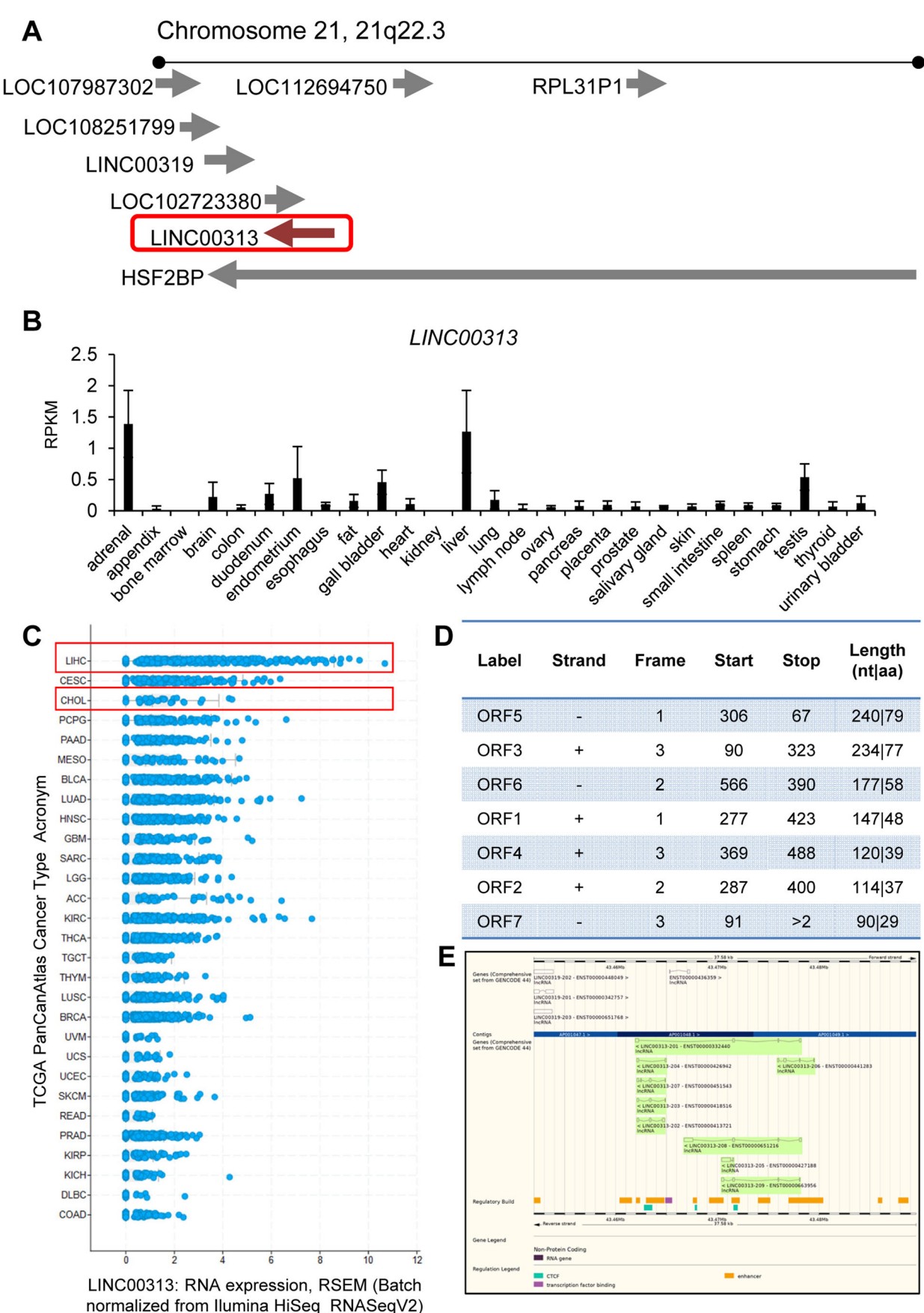

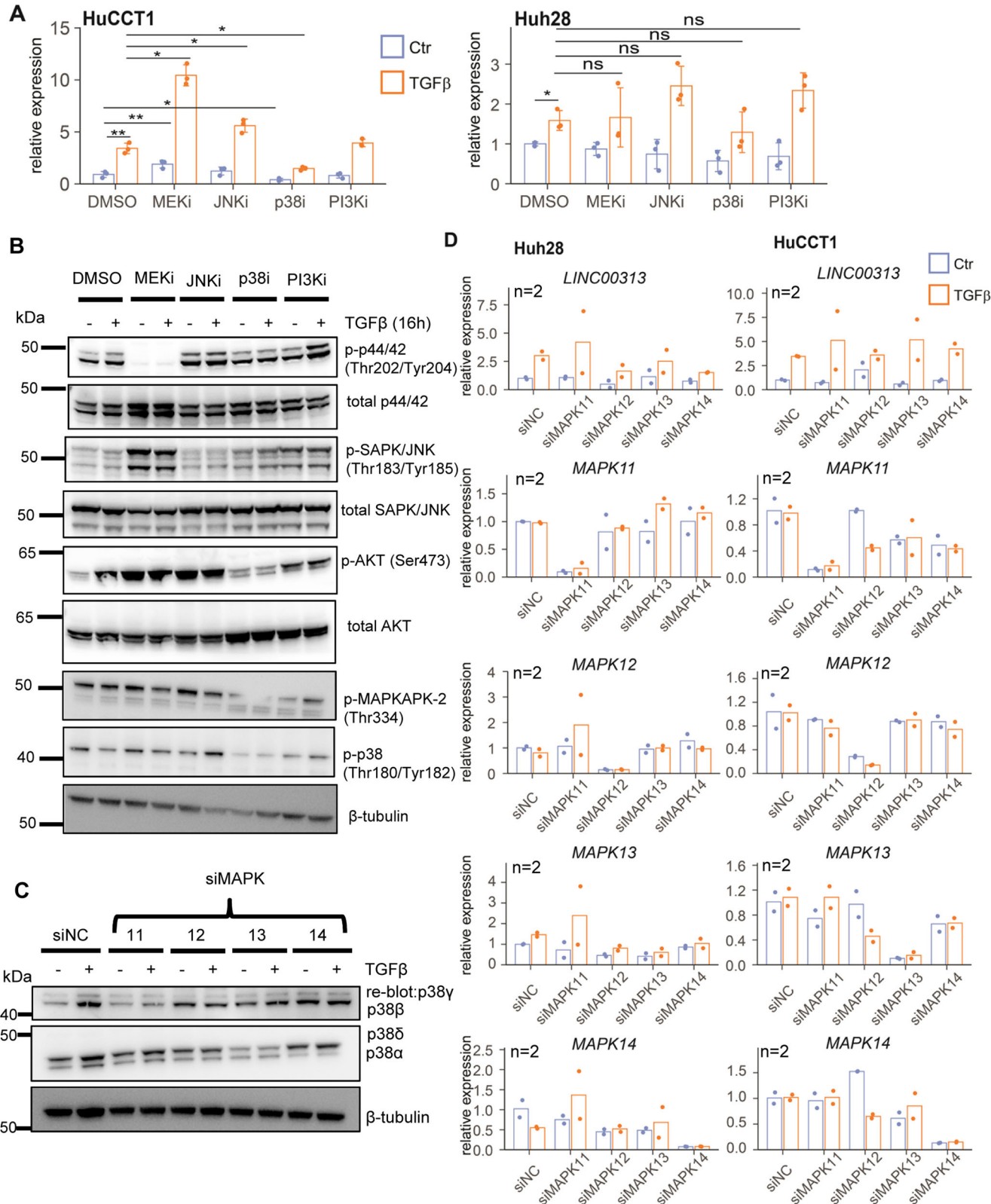

**Figure EV2.  Effect of the non-canonical TGFβ pathway on LINC00313 expression.**

(**A**) *LINC00313* levels in HuCCT1 or Huh28 treated with MEK, p38, JNK or PI3K inhibitors, with or without TGFβ1 stimulation for 16 h ($n = 3$ biological replicates).
(**B**) Immunoblotting for detection of phosphorylated p44/42 (Thr202/Tyr204), phosphorylated p-SAPK/JNK (Thr183/Tyr185), phosphorylated AKT (Ser473), phosphorylated p38 (Thr180/Tyr182), phosphorylated MAPKAPK-2 (Thr334), total p44/42, SAPK/JNK and AKT protein levels and β-tubulin (used as a loading control) in HuCCT1 cells treated with MEK, p38, JNK or PI3K inhibitors, with or without TGFβ1 stimulation for 16 h. DMSO was used as a vehicle treatment. (**C**) Immunoblotting for detection of the p38 isoforms p38α, p38β, p38γ and p38δ in HuCCT1 cells transiently silenced for *MAPK11* or *MAPK12* or *MAPK13* or *MAPK14* and treated or not with TGFβ1 for 16 h. β-tubulin is used as a loading control. (**D**) Real-time qPCR analysis of *LINC00313, MAPK11, MAPK12, MAPK13* and *MAPK14* expression in HuCCT1 and Huh28 cell lines transiently transfected with siRNAs targeting *MAPK11* or *MAPK12* or *MAPK13* or *MAPK14* and treated or not with TGFβ1 for 16 h. Data information: In Panel (**A**) data are presented as mean ± SD ($n = 3$ biological replicates). In panel (**D**) data are presented as single data points ($n = 2$ biological replicates). $*P \leq 0.05$, $**P \leq 0.01$, n.s.: not significant (Student's *t*-test). Source data are available online for this figure.

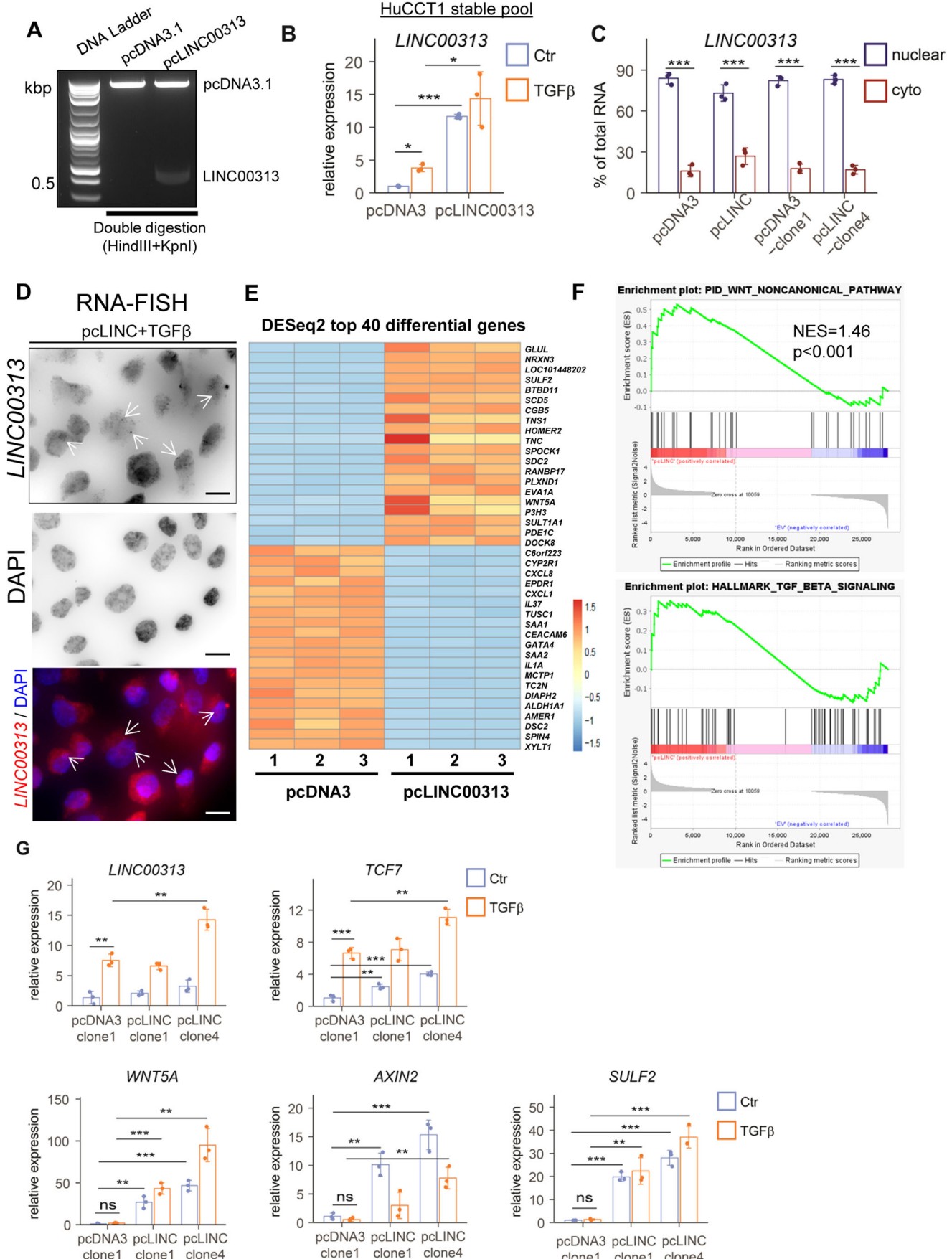

◀ **Figure EV3.** *LINC00313* **gain of function up-regulates Wnt-related genes.**

(**A**) Efficient ligation of LINC00313 insert to pcDNA3.1 plasmid vector was verified by double digestion with restriction enzymes, followed by agarose gel electrophoresis. Molecular size (in kbp) marker ladder is shown in the first lane. (**B**) Real-time qPCR analysis of *LINC00313* expression in HuCCT1 cells stably over-expressing pcDNA3.1 or pcLINC00313 expression vectors stimulated or not with TGFβ1 for 16 h. (**C**) Subcellular fractionation followed by RT-qPCR analysis of LINC00313 RNA levels in nuclear and cytoplasmic fractions of HuCCT1 cells transiently over-expressing an empty vector (pcDNA3.1) or LINC00313 and HuCCT1 clones, stably over-expressing pcDNA3.1 or LINC00313. (**D**) smiFISH for LINC00313 RNA in HuCCT1 cells stably over-expressing LINC00313 and stimulated with TGFβ for 16 h. DAPI was used to stain nuclei. Scale bars: 15 μm. (**E**) Heatmap presenting the top 40 differentially regulated genes in HuCCT1 pcLINC00313 versus pcDNA3-expressing cells (shrunken log2FC > 1, adjusted *p*-value < 0.1). Biological triplicates (1-3) were used per condition. (**F**) GSEA analysis of differentially expressed genes in pcLINC00313 versus pcDNA3 HuCCT1 cells. (**G**) Real-time qPCR analysis of *LINC00313*, *TCF7*, *WNT5A*, *AXIN2*, and *SULF2* expression in a monoclonal HuCCT1 cell line stably expressing pcDNA3.1 and in two monoclonal HuCCT1 cell lines stably expressing pcLINC00313, in the presence or not of TGFβ1 for 16 h. Data information: Data in panels (**B**), (**C**) and (**G**) are presented as mean ± SD (*n* = 3 biological replicates). *$P \le 0.05$, **$P \le 0.01$, ***$P \le 0.001$ (Student's *t*-test). In panel (**F**) the Kolmogorov-Smirnov test was used for statistical analysis of GSEA. Source data are available online for this figure.

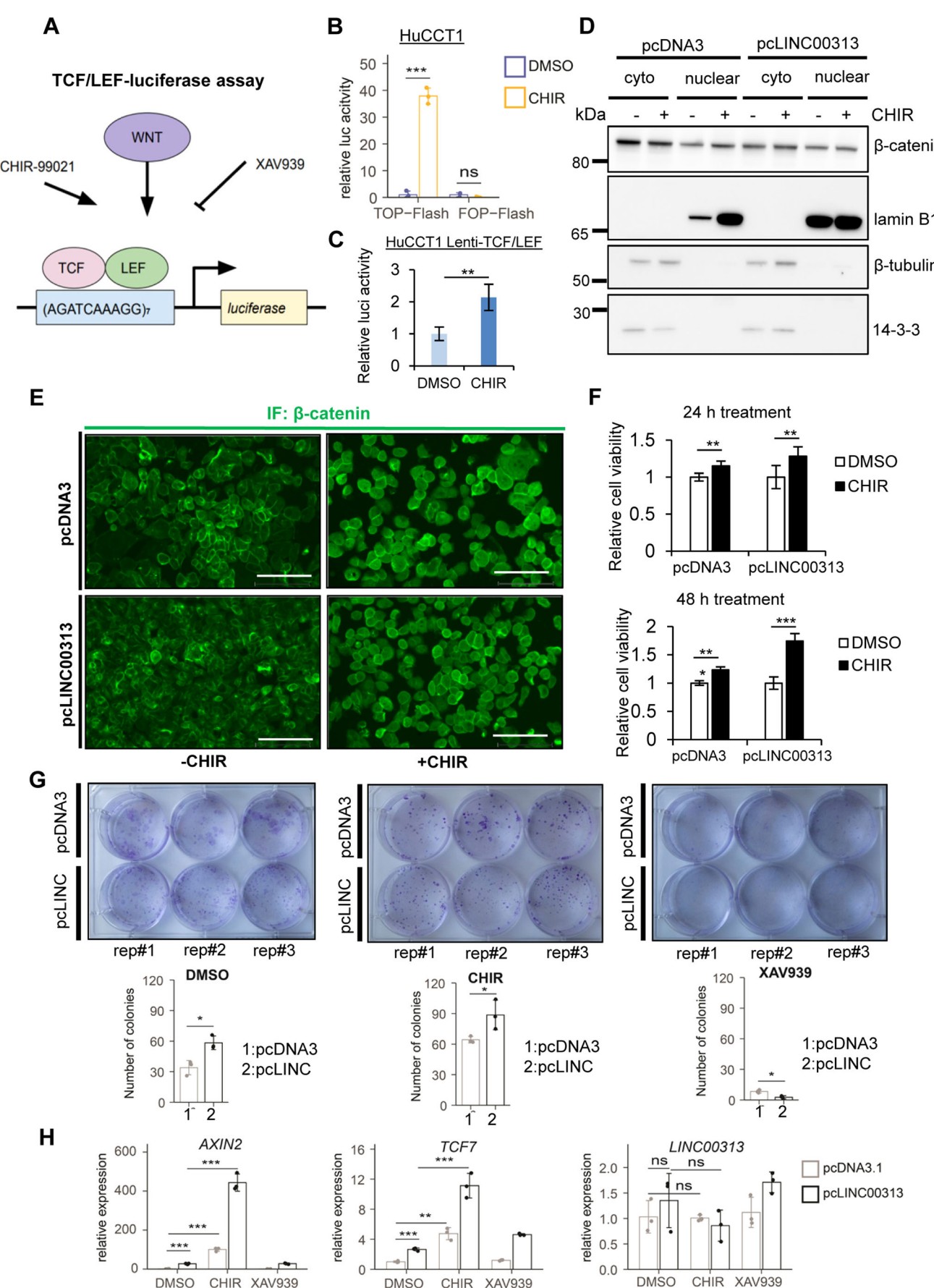

◀  **Figure EV4.  *LINC00313* modulates chemically-induced Wnt/β-catenin signalling and impacts physiological processes in CCA.**

(A) Schematic representation of the TCF/LEF-luciferase reporter assay. Activators (Wnt ligands, CHIR) and inhibitors (XAV939) of TCF/LEF-dependent responses are also shown. (B) TCF/LEF-luciferase reporter assay in HuCCT1 cells transiently transfected with TOP-Flash or FOP-Flash luciferase expression vectors and treated with CHIR99021 or DMSO for 24 h. Cells were co-transfected with a Renilla luciferase expression vector for normalization of the firefly luciferase activity. (C) TCF/LEF-luciferase reporter assay in HuCCT1 cells stably expressing the pGreenFire 2.0 TCF/LEF reporter construct and treated with CHIR99021 or DMSO for 24 h. TCF/LEF-luciferase reporter assay in HuCCT1 cells stably expressing the pGreenFire 2.0 TCF/LEF reporter construct and treated with the indicated concentrations of TGFβ1 or BSA/HCl (Ctr) for 16 h. (D) Subcellular fractionation followed by immunoblotting for β-catenin, lamin B1 (nuclear marker), β-tubulin and 14-3-3 (cytoplasmic markers) using nuclear and cytoplasmic fractions of HuCCT1 cells stably over-expressing an empty vector (pcDNA3.1) or LINC00313 and treated with CHIR99021 or DMSO for 24 h. (E) Immunofluorescence to detect β-catenin subcellular localization in control or LINC00313 over-expressing HuCCT1 cells, treated or not with CHIR99021 or DMSO for 24 h. Scale bars: 150 μm. (F) Cell viability assays in control or LINC00313 over-expressing HuCCT1 cells treated with CHIR99021 or DMSO for 24 h. (G) Colony formation assay in control or LINC00313 over-expressing HuCCT1 cells, treated with CHIR99021 or XAV939 or DMSO for 24 h. Quantification of the number of colonies for each condition is also shown. (H) Real-time qPCR analysis of *AXIN2*, *TCF7* and *LINC00313* expression in control or LINC00313 overexpressing HuCCT1 cells, treated with CHIR99021, XAV939, or DMSO for 24 h. Data information: In panel (B) data are presented as mean ± SD ($n = 3$ biological replicates). In panel (F) data are presented as mean ± SD ($n = 6$ biological replicates). **$P \leq 0.01$, ***$P \leq 0.001$ (Student's *t*-test). In panels (G) and (H) data are presented as mean ± SD ($n = 3$ biological replicates). *$P \leq 0.05$, **$P \leq 0.01$, ***$P \leq 0.001$ (Student's *t*-test). Source data are available online for this figure.

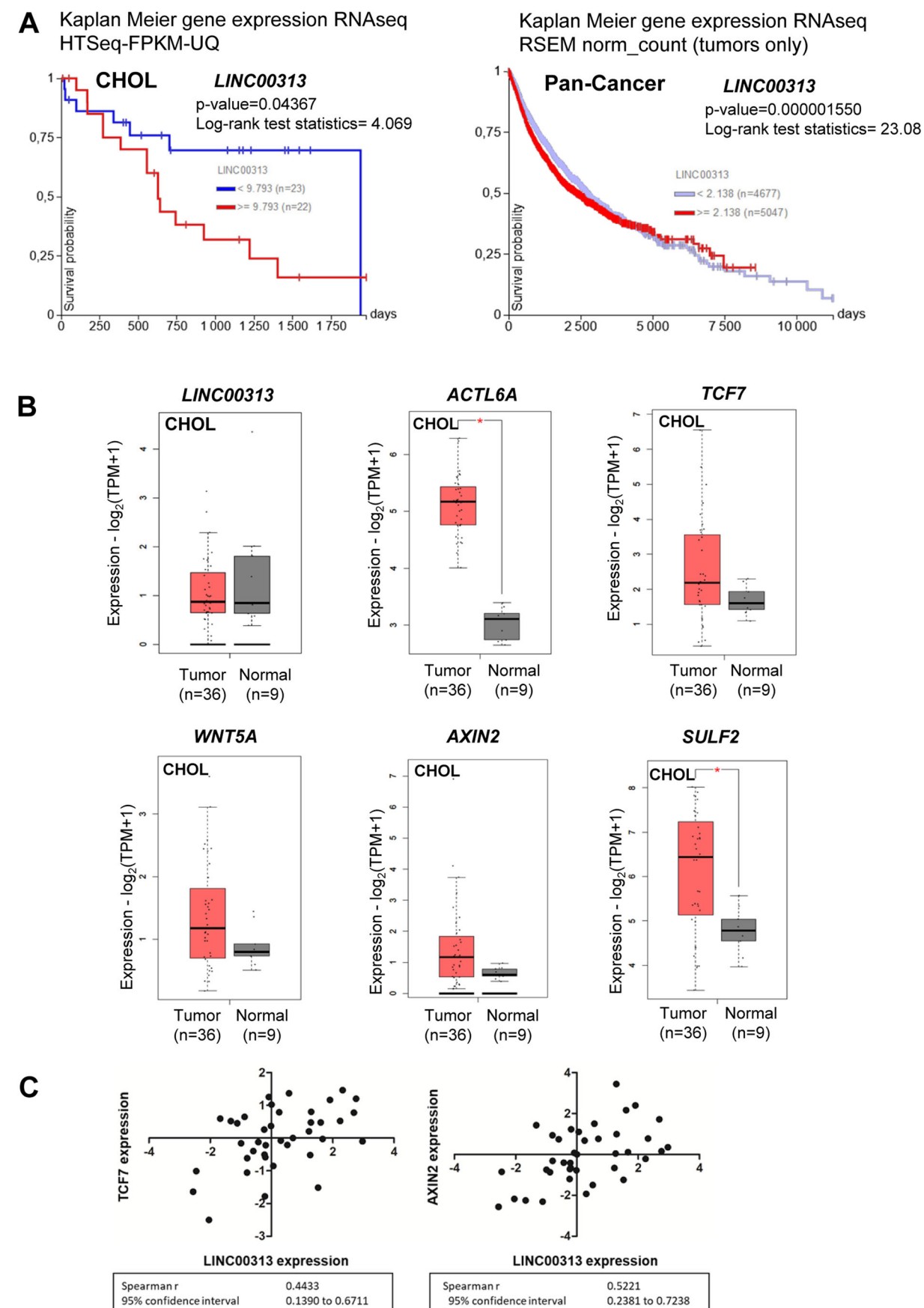

◀   **Figure EV5.   Clinical relevance of LINC00313 in CCA.**

(**A**) Kaplan Meier curves depicting overall survival in CHOL patients and Pan-Cancer patients grouped based on *LINC00313* median expression, using the Xena Functional Genomics Explorer from the UCSC. (**B**) Boxplots of LINC00313, ACTL6A, TCF7, WNT5A, AXIN2 and SULF2 transcripts per million (TPMs) in cholangiocarcinoma (CHOL) tumours ($n = 36$) and their paired normal tissues ($n = 9$), using GEPIA2. *$p < 0.01$. (**C**) Expression correlation of *LINC00313* with *TCF7* and *AXIN2* in a cohort of 39 iCCA patients $p < 0.05$. Data information: In panel (**A**), log-rank test was used to compare the different Kaplan–Meier curves. In panel (**B**), transcript expression is shown in logarithmic scale and the method for differential analysis is one-way ANOVA, using disease state (tumour vs normal) as variable for calculating differential expression. The $\log_2$FC cut-off equals 1 and the $p$-value cut-off equals 0.01. In panel (**C**) a two-tailed Student's $t$-test was used to calculate $p$-values.

                                          