## [Peer Review File · EMBO Reports]

TGF β -induced long non-coding RNA LINC00313 activates Wnt signaling and promotes cholangiocarcinoma

Panagiotis Papoutsoglou, Raphael Pineau, Raffaële Leroux, Corentin Louis, Anaïs L'Haridon, Dominika Foretek, Antonin Morillon, Jesús Banales, David Gilot, Marc Aubry, and Cedric Coulouarn

Corresponding author(s): Cedric Coulouarn (cedric.coulouarn@inserm.fr)

Review Timeline:

Submission Date:	26th Sep 22
Editorial Decision:	25th Oct 22
Revision Received:	8th Nov 23
Editorial Decision:	4th Dec 23
Revision Received:	13th Jan 24
Accepted:	18th Jan 24

Editor: Achim Breiling

Transaction Report:

Dear Dr. Coulouarn,

Thank you for the submission of your research manuscript to EMBO reports. I have now received the reports from the three referees that were asked to evaluate your study, which can be found at the end of this email.

As you will see, the referees think that these findings are of interest. However, they have several comments, concerns, and suggestions, indicating that a major revision of the manuscript is necessary to allow publication of the study in EMBO reports. As the reports are below, and all the referee concerns need to be addressed, I will not detail them here.

Given the constructive referee comments, we would like to invite you to revise your manuscript with the understanding that all referee concerns must be addressed in the revised manuscript and in a detailed point-by-point response. Acceptance of your manuscript will depend on a positive outcome of a second round of review. It is EMBO reports policy to allow a single round of revision only and acceptance of the manuscript will therefore depend on the completeness of your responses included in the next, final version of the manuscript.

- 1) a .docx formatted version of the final manuscript text (including legends for main figures, EV figures and tables), but without the figures included. Figure legends should be compiled at the end of the manuscript text.
- 2) individual production quality figure files as .eps, .tif, .jpg (one file per figure), of main figures (up to 8) and EV figures. Please upload these as separate, individual files upon re-submission.

- 3) a complete author checklist, which you can download from our author guidelines (<https://www.embopress.org/page/journal/14693178/authorguide>). Please insert page numbers in the checklist to indicate where the requested information can be found in the manuscript. The completed author checklist will also be part of the RPF.

- 4) that primary datasets produced in this study (e.g. RNA-seq, ChIP-seq, structural and array data) are deposited in an appropriate public database. If no primary datasets have been deposited, please also state this in a dedicated section (e.g. 'No primary datasets have been generated and deposited'), see below.

The accession numbers and database should be listed in a formal "Data Availability" section (placed after Materials & Methods) that follows the model below. This is now mandatory (like the COI statement). Please note that the Data Availability Section is restricted to new primary data that are part of this study. This section is mandatory. As indicated above, if no primary datasets have been deposited, please state this in this section

Data availability

8) Regarding data quantification and statistics, please make sure that the number "n" for how many independent experiments were performed, their nature (biological versus technical replicates), the bars and error bars (e.g. SEM, SD) and the test used to calculate p-values is indicated in the respective figure legends (also for potential EV figures and all those in the final Appendix). Please also check that all the p-values are explained in the legend, and that these fit to those shown in the figure. Please provide statistical testing where applicable. Please avoid the phrase 'independent experiment', but clearly state if these were biological or technical replicates. Please also indicate (e.g. with n.s.) if testing was performed, but the differences are not significant. In case n=2, please show the data as separate datapoints without error bars and statistics. See also: <http://www.embopress.org/page/journal/14693178/authorguide#statisticalanalysis>

9) Please also note our reference format:

10) Please add scale bars of similar style and thickness to all the microscopic images, using clearly visible black or white bars (depending on the background). Please place these in the lower right corner of the images themselves. Please do not write on or near the bars in the image but define the size in the respective figure legend.

12) We now use CRediT to specify the contributions of each author in the journal submission system. CRediT replaces the author contribution section. Please use the free text box to provide more detailed descriptions. See also guide to authors: <https://www.embopress.org/page/journal/14693178/authorguide#authorshipguidelines>

I look forward to seeing a revised version of your manuscript when it is ready. Please let me know if you have questions or comments regarding the revision.

Yours sincerely,

Achim Breiling

Referee #1:

The manuscript by Papoutsoglou et al., explores the functional role and potential mechanism of action of LINC00313 lncRNA in Cholangiocarcinoma (CCA). The authors identified LINC00313 as a novel TGF β target that acts as a transcriptional regulator of Wnt signaling by interacting with the SWI/SNF subunit ACTL6A, with implications in CCA tumor growth.

The manuscript presents valid data that convincingly demonstrate TGF β -mediated regulation, while orthogonal approaches, such as lncRNA over-expression, RNAi and the use of different inhibitors, support the involvement of this lncRNA in both TGF β and Wnt signaling.

However, most of the genomic data and some of the functional assays suffer of a concerning lack of method description, which hinders reviewer's ability to judge the quality of this work. Furthermore, the proposed mechanism is not supported by robust data.

Therefore, while an interesting study, several important shortcomings must be addressed, as follow:

Major points:

1. Method description, including essential information regarding number of samples/replicates, is insufficient almost everywhere across the manuscript. This includes but is not limited to ATAC-seq analysis, lacking both sample description (how many replicates were used? sequencing layout/depth/coverage?) as well as details about downstream data analysis (what quality testing, peak filtering, differential binding analyses were performed?). Similarly, mass spectrometry processing and data filtering were omitted with only limited reference to published literature, a regrettable practice noticed in multiple instances in the paper. These and similar issues must be addressed to properly assess the quality of this study.

2. More conceptually, the relationship between LINC00313 and TGF β and Wnt pathways is unclear. Since TGF β activates LINC00313 expression and in turn LINC00313 controls the Wnt pathway, it would be expected that TGF β could regulate Wnt signaling to some extent. Instead, the only gene that appears to be consistently regulated by TGF β is TCF7 (Figure 4G-S6). Also, although the authors present some pathway analysis of gene profiling with and without TGF β (S5), LINC00313 significance in this context remains unclear, in particular given that LINC00313 KD+TGF β treatment results in fewer differentially expressed genes (p-values must be included in Table S3). In figure S9A, the authors suggest a possible antagonistic role for LINC00313 in counteracting TGF β -mediated growth arrest (this assay should be shown as a growth curve). Is this finding supported by gene expression data? Further data mining could be used to propose a hypothesis presented in the discussion.

3. The proposed molecular mechanism involving LINC00313 mediated regulation of ACTL6A activity is not convincing for several reasons:

- ACTL6A was selected as candidate based on cherry picking of protein interactors revealed by MS analysis. Since ACTL6A is a SWI/SNF subunit, additional components of the complex would be expected to interact with LINC00313 in this type of experiment.
- The validation of the supposed interaction by RNA pulldown and RIP assays is not convincing, given the barely detectable level of ACTL6A in pulldown extracts (6H) and the low LINC00313 enrichment in RIP-qPCR (6J - performed without appropriate positive and negative control RNAs). Since this might be due to the low efficiency observed for ACTL6A IP (6J), authors could try to improve their assay by using a different antibody for ACTL6A, the tagged version or by IP-ing a different subunit of the SWI/SNF complex (e.g., BRG1).
- The results obtained with ATPase inhibitors are highly misleading (7E-F): as SWI/SNF complexes regulate transcription of a significant fraction of the genome, a reduction in gene expression is expected (Iurlaro et al., Nat Genet 2021). Moreover, the authors show that SULF2 and TCF7 basal levels are already reduced in control cells, confirming that LINC00313 role is neglectable.
- ACTL6A KD is very variable (7A&D) at protein level, weakening the author's conclusions.

In conclusion, ACTL6A role in LINC00313-mediated regulation of the Wnt pathway does not appear convincing. However, given the strong correlation between TCF7 and LINC00313 levels (5G) and the robust set of data connecting these two genes, the authors should explore whether other mechanisms could regulate this gene, since as a transcriptional activator of the Wnt pathway, it could serve as mediator of LINC00313 function.

Minor points:

1. [92-93] The low number of DE genes in the initial gene expression array +/- TGF β (1A) is surprising, given the different results in S5. Is this due to high variability between cell lines? If so, this should be shown and justified. Also, Table S1 should report separate fold change for the two conditions and adj. p-values)

2. [99-100] Assuming these primers were used to investigate LINC00313 different isoforms (based on hg38-T2T genome browsers), a schematic should be included in supplementary figures (possibly including siRNA binding sites).
3. [113-116] WB showing SMADs KD would be beneficial, as different protein levels could explain the differences in the role of distinct SMADs in HuCCT1 vs Huh28. As general comment, this paragraph is rather confusing and not very insightful. While the authors demonstrate that SMAD-dependent signaling could regulate LINC00313, the role of the SMAD-independent pathway is unclear, in particular since the authors show by luciferase assay that MEK pathway signals through the SMAD response. Whereas feedback mechanisms connecting the two pathways could be present, investigating the SMAD-independent pathway seems irrelevant. Instead, it would be interesting to analyze whether LINC00313 transcription is directly controlled by SMAD proteins by scanning LINC00313 promoter for SMAD-binding elements or performing ChIP.
4. [210-213] The discrepancy in TCF7 correlation/levels is very surprising. What is LINC00313 level in tumors from xenografted mice?
5. [Figure 8A] This figure is not informative, as it doesn't show whether the genes up and down regulated in xenografted mice reflect in vitro analyses. Substituting the nameless heatmap with more insightful GSEA would be useful.
6. Please, add statistics in figures where needed.

Referee #2:

In this work, Papoutsoglou and cols have explored the role of the Long non-coding RNA LINC00313 in cholangiocarcinoma (CCA). They identify it as a novel target of the TGF-beta pathway in CCA cells. Through elegant gene expression analysis and genome-wide chromatin accessibility profiling, authors demonstrate that LINC00313 in the nucleus transcriptionally regulates genes involved in Wnt signaling. In vivo experiments, analysis with human tissues and mechanistic studies led to the authors to propose a model whereby TGF-beta induces LINC00313 to regulate expression of hallmark Wnt genes in cooperation with SWI/SNF, which would boost cholangiocarcinogenesis.

Work has been well designed and performed. The manuscript is clear, well-presented figures and excellent text. Overall, the results open new perspectives on the molecular mechanisms responsible for cholangiocarcinogenesis.

Nevertheless, I have some comments/recommendations to improve the manuscript.

MAJOR POINTS

1. Results strongly support a role for TBRI/SMAD pathway in the induction of LINC00313 expression. However, when other non-canonical pathways are analysed, results are not so clear. Authors indicate a strong role for p38, whereas other signals, such as ERK or JNK, could be inhibiting the TGF-beta induction of LINC00313, due to its potential to counteract SMAD signaling. However, these results are only observed in the HuCCT1, not so clear in the Huh28 cells. Furthermore, the p38 inhibitor is the only one that is not analysed in the Figure S3 (probably because it is necessary to analyse p38 activity). And, relevant, p38 inhibitor strongly inhibits AKT phosphorylation, even at higher levels than the PI3Ki. To firmly demonstrate that p38 is mediating the regulation of LINC00313 by TGF-beta, it is necessary to do specific experiments downregulating p38 with siRNA, as performed with SMADs. And it would be necessary to check this point also in the Huh28 cells, or other alternative CCA cells.
2. The functional analysis for the role of LINC00313 on proliferation and migration is not conclusive. Effects on cell viability are very modest. The analysis on cell migration, only through a wound healing assay, is insufficient. And these experiments are essential to support the results later presented in the in vivo model. I suggest doing specific experiments of progression in cell cycle, such as BrdU or analysis of cell cycle by flow cytometry. And for migration, it is necessary to work in a two-chamber plate, where authors may visualize cell migration or invasion, modulating the matrix composition between the two chambers.
3. A point that is difficult to understand and that could compromise the model proposed by the authors is that in the in vivo experiments (Fig. 4) the tumors from xenografted pcLINC00313 cells do not show increase in TCF7 expression, although these cells presented higher mRNA levels in vitro (Fig. 3F) and a perfect correlation between TCF7 and LINC00313 is observed in vivo (Fig. 4G). Authors show that protein is increased, but it is only clear in 2-3 of the 6 tumors analysed (densitometric analysis and statistics is required). How LINC00313 could regulate the levels of TCF7 protein? Authors must discuss this point.
4. In parallel, in the TCGA dataset, LINC00313 was neither overexpressed in CCA tissues nor correlated with overall or disease-free survival (Fig. S11A). Authors then focus on the analysis of LINC00313 "activity". Design of the experiment is correct, however, the in vivo gene signature, which correlates with a differential overall survival, is a gene signature extremely related to other signaling pathways. How could authors demonstrate, or at least discuss, that the "activity" of LINC00313 could be modulated in some CCA patients?

MINOR POINTS

5. Page 5, line 104. Here authors cite SNORD48 gene, whereas in the figure another alias, RNU48, is included. I suggest homogenising the names.

6. Figure S8E: it is difficult to extrapolate conclusions from this experiment. I would recommend complementing it with analysis of Western blots after isolating nuclei. It is very important to clarify this point.

7. Fig. 7C-F: I would recommend including statistical analysis.

8. Considering the strong effects of BRM/BRG1 ATPi on TCF7 expression in TGF-beta-treated cells (Fig. 7F), why do authors analyse the effect of this inhibitor on CHIR-induced TCF/LEF luciferase reporter activity, instead on TGF-beta treated cells (Fig. 7G)?

Referee #3:

Here the authors investigated the role of the linc00131 and TGFbeta in cholangiocarcinoma (CCA). LINC00313 lncRNA is a target of TGFβ signalling in CCA cells. LINC00313 is nuclear and transcriptionally regulated genes involved in Wnt signalling (TCF7). LINC00313 expression enhanced TCF/LEF-dependent target genes, associated to tumor formation. And decrease inpatient's overall survival. The authors found that ACTL6A interacts with LINC00131 controlling TCF7 and SULF2 transcription. They propose a working model in which TGFbeta induces LINC00313 in to regulate expression of Wnt pathway genes, in co-operation with SWI/SNF complex.

The study is potentially interesting, and the conclusions are supported by the experiment shown. The authors should address the following points:

1) Explain why the authors choose to study LINC00313. At basal level (Fig.1B), LINC00313 is expressed at different levels in iCCA/eCCA cell lines, indicating that its role is not necessary for the biology of the tumors, how the authors explain this point? Also, not all iCCA LINC00313 expression is TGFbeta dependent. The authors should also indicate the RTqPCRs DeltaCt to give information on the abundance of lincRNA. Is LINC00131 undergoing maturation? A characterization of this lincRNA should also be shown.

2) Fig 2: controls by western blot of silencing should be included.

3) Nuclear localization should also be confirmed by FISH experiments in cell lines +/-TGFbeta.

4) siRNA used for LINC00313 silencing should be better characterized to demonstrate that no off target occurred.

5) The authors should perform FISH experiments to detect LINC00131 in CCA tissue specimens.

6) Correlation in datasets of LINC00131-associated genes expression in CCA patients.

Rebuttal letter

Manuscript number: EMBOR-2022-56179V1

Papoutsoglou et al. TGF β -induced long non-coding RNA LINC00313 activates Wnt signalling and promotes cholangiocarcinoma.

Rennes, October 26th, 2023

Dear Dr. Breiling, Senior Editor *EMBO Reports*

We thank you for your editorial assessment and reviews from external referees. I would like here to submit the revised version of our paper entitled “TGF β -induced long non-coding RNA LINC00313 activates Wnt signalling and promotes cholangiocarcinoma” and authored by P. Papoutsoglou, R. Pineau, T. Ferlier, R. Leroux, C. Louis, A. L’Haridon, D. Foretek, A. Morillon, JM. Banales, D. Gilot, M. Aubry and C. Coulouarn. Please note that four new authors, whose names are underlined, are included in this version.

In the revised version, we enhanced the quality of the manuscript, including a set of new experimental data, changes in the text and interpretation of some results, adding extensive information on methods description, and re-formatting the paper according to the *EMBO Reports* style, following the constructive comments by the reviewers and editorial suggestions. As explained below, we have responded to all reviewers’ comments, except for one (Reviewer #3, point #6, about performing FISH experiments to detect LINC00131 in human CCA tissue specimens), which was not feasible due to technical constraints (no access to freshly frozen human CCA tissues). Hereby, you will find a point-by-point response to your comments as well as the comments of the three referees.

Oncogenesis Stress Signaling Laboratory

INSERM U1242 Université Rennes 1 -Centre de Lutte Contre le Cancer Eugène Marquis
Rue de la Bataille Flandres Dunkerque - CS 44229 35042 RENNES CEDEX

Point-by-point response to the Editor's comments

Thank you for the submission of your research manuscript to *EMBO reports*. I have now received the reports from the three referees that were asked to evaluate your study, which can be found at the end of this email. As you will see, the referees think that these findings are of interest. However, they have several comments, concerns, and suggestions, indicating that a major revision of the manuscript is necessary to allow publication of the study in *EMBO reports*.

✓ Authors' response

We thank you for giving us the opportunity to re-submit a revised version of our manuscript. In the revised manuscript we answered the points raised by the reviewers by performing new experiments and by enhancing the quality of the presented data.

We now request the publication of original source data with the aim of making primary data more accessible and transparent to the reader. Our source data coordinator will contact you to discuss which figure panels we would need source data for and will also provide you with helpful tips on how to upload and organize the files.

✓ Authors' response

In response to editor's request we have now provided in the revised manuscript all the source data required for the indicated figure panels, as discussed with the *EMBO Reports* source data coordinator.

Our journal encourages inclusion of *data citations in the reference list* to directly cite datasets that were re-used and obtained from public databases. Data citations in the article text are distinct from normal bibliographical citations and should directly link to the database records from which the data can be accessed. In the main text, data citations are formatted as follows: "Data ref: Smith et al, 2001" or "Data ref: NCBI Sequence Read Archive PRJNA342805, 2017". In the Reference list, data citations must be labeled with "[DATASET]". A data reference must provide the database name, accession number/identifiers and a resolvable link to the landing page from which the data can be accessed at the end of the reference. Further instructions are available at: <http://www.embopress.org/page/journal/14693178/authorguide#referencesformat>

✓ Authors' response

Oncogenesis Stress Signaling Laboratory

INSERM U1242 Université Rennes 1 -Centre de Lutte Contre le Cancer Eugène Marquis
Rue de la Bataille Flandres Dunkerque - CS 44229 35042 RENNES CEDEX

We have modified accordingly our citation for using intrahepatic cholangiocarcinoma RNA-Seq data (Dong et al, 2022) for our integrative analysis, which are publicly available in the NODE database.

Regarding data quantification and statistics, please make sure that the number "n" for how many independent experiments were performed, their nature (biological versus technical replicates), the bars and error bars (e.g. SEM, SD) and the test used to calculate p-values is indicated in the respective figure legends (also for potential EV figures and all those in the final Appendix). Please also check that all the p-values are explained in the legend, and that these fit to those shown in the figure. Please provide statistical testing where applicable. Please avoid the phrase 'independent experiment', but clearly state if these were biological or technical replicates. Please also indicate (e.g. with n.s.) if testing was performed, but the differences are not significant. In case n=2, please show the data as separate datapoints without error bars and statistics. See also: <http://www.embopress.org/page/journal/14693178/authorguide#statisticalanalysis>

✓ **Authors' response**

In response to editor's request, we have provided adequate information about sample size, data quantification and statistics in every figure legend for main, EV and Appendix figures.

Please also note our reference format:

✓ **Authors' response**

We have updated our reference format according to the latest *EMBO Reports* style.

Please add scale bars of similar style and thickness to all the microscopic images, using clearly visible black or white bars (depending on the background). Please place these in the lower right corner of the images themselves. Please do not write on or near the bars in the image but define the size in the respective figure legend.

✓ **Authors' response**

As instructed by the Editor, we have modified our images (main Figure 6 and Figure EV4) derived from immunofluorescence experiments. We have also edited accordingly the microscopy images from the RNA FISH experiment shown in Figure EV.

Oncogenesis Stress Signaling Laboratory

INSERM U1242 Université Rennes 1 -Centre de Lutte Contre le Cancer Eugène Marquis
Rue de la Bataille Flandres Dunkerque - CS 44229 35042 RENNES CEDEX

We updated our journal's competing interests policy in January 2022 and request authors to consider both actual and perceived competing interests. Please review the policy <https://www.embopress.org/competing-interests> and update your competing interests if necessary. Please name this section 'Disclosure and Competing Interests Statement' and put it after the Acknowledgements section.

✓ **Authors' response**

We have now added this section as requested by the Editor.

We now use CRediT to specify the contributions of each author in the journal submission system. CRediT replaces the author contribution section. Please use the free text box to provide more detailed descriptions. See also guide to authors: <https://www.embopress.org/page/journal/14693178/authorguide#authorshipguidelines>

✓ **Authors' response**

We have now used the CRediT to specify authors' contributions and removed the author contribution section, initially provided in the first manuscript.

Oncogenesis Stress Signaling Laboratory

INSERM U1242 Université Rennes 1 -Centre de Lutte Contre le Cancer Eugène Marquis
Rue de la Bataille Flandres Dunkerque - CS 44229 35042 RENNES CEDEX

Point-by-point response to the Referee #1's comments

The manuscript by Papoutsoglou et al., explores the functional role and potential mechanism of action of LINC00313 lncRNA in Cholangiocarcinoma (CCA). The authors identified LINC00313 as a novel TGF β target that acts as a transcriptional regulator of Wnt signaling by interacting with the SWI/SNF subunit ACTL6A, with implications in CCA tumor growth. The manuscript presents valid data that convincingly demonstrate TGF β -mediated regulation, while orthogonal approaches, such as lncRNA over-expression, RNAi and the use of different inhibitors, support the involvement of this lncRNA in both TGF β and Wnt signaling.

However, most of the genomic data and some of the functional assays suffer of a concerning lack of method description, which hinders reviewer's ability to judge the quality of this work. Furthermore, the proposed mechanism is not supported by robust data. Therefore, while an interesting study, several important shortcomings must be addressed, as follow:

✓ **Authors' response**

We thank Referee #1 for her/his encouraging comments and critical review of our study. We have taken into account your comments, as described below, and modified the manuscript accordingly. Notably, in the revised manuscript we have made serious efforts to respond to your comments by significantly enhancing the quality of the methods description and by performing new experiments to strengthen the mechanistic aspects of LINC00313 molecular function, but also its biological roles, as described below.

Major points:

1. Method description, including essential information regarding number of samples/replicates, is insufficient almost everywhere across the manuscript. This includes but is not limited to ATAC-seq analysis, lacking both sample description (how many replicates were used? sequencing layout/depth/coverage?) as well as details about downstream data analysis (what quality testing, peak filtering, differential binding analyses were performed?). Similarly, mass spectrometry processing and data filtering were omitted with only limited reference to published literature, a regrettable practice noticed in multiple instances in the paper. These and similar issues must be addressed to properly assess the quality of this study.

✓ **Authors' response**

We thank the reviewer for her/his comment and we apologize for not including this essential information. We have updated the Material and Methods section by adding

Oncogenesis Stress Signaling Laboratory

INSERM U1242 Université Rennes 1 -Centre de Lutte Contre le Cancer Eugène Marquis
Rue de la Bataille Flandres Dunkerque - CS 44229 35042 RENNES CEDEX

extensive information about the experimental procedures and data analysis of ATAC-seq, mass-spec and RNA immunoprecipitation (RIP) experiments. In addition, we have added detailed information about the number of samples or replicates in the main, extended view and supplementary figure legends. We have also deposited the raw data from the microarray experiments, RNA-seq and ATAC-seq analysis to Gene Expression Omnibus (accession number: GSE102109) and ArrayExpress (please see below for the specific links). In the deposited data, reviewers can get information about the methodological approach used to acquire the gene expression profiling (RNA-seq and microarray) and chromatin accessibility (ATAC-seq) data. The reviewers can access them using the following login details:

The microarray data (TGF-beta signatures in HuCCT1 and Huh28 cholangiocarcinoma cell lines, Fig. 1A and Table S1) are publicly available on the following link:

<https://www.ncbi.nlm.nih.gov/geo/query/acc.cgi?acc=GSE102109>

RNA-seq and ATAC-seq data are currently private but the following links are available for the Reviewers:

- RNA-seq data:

<https://www.ebi.ac.uk/biostudies/arrayexpress/studies/E-MTAB-12030?key=6905955f-420a-4002-ad36-5a5655b0cb65>

- ATAC-seq data:

<https://www.ebi.ac.uk/biostudies/arrayexpress/studies/E-MTAB-12879?key=7fc6a860-07ac-4c45-a060-19de0d84f1d9>

2. More conceptually, the relationship between LINC00313 and TGFb and Wnt pathways is unclear. Since TGFb activates LINC00313 expression and in turn LINC00313 controls the Wnt pathway, it would be expected that TGFb could regulate Wnt signaling to some extent. Instead, the only gene that appears to be consistently regulated by TGFb is TCF7 (Figure 4G-S6). Also, although the authors present some pathway analysis of gene profiling with and without TGFb (S5), LINC00313 significance in this context remains unclear, in particular given that LINC00313 KD+TGFb treatment results in fewer differentially expressed genes (p-values must be included in Table S3).

✓ **Authors' response**

We fully understand the Reviewer's comment. In fact, it has been previously reported that TGF β is able to enhance Wnt pathway activity in the Huh7 hepatocellular carcinoma cell line (Hoshida et al., *Cancer Res.* 2009, PMID: 19723656). Thus, to better clarify the

Oncogenesis Stress Signaling Laboratory

INSERM U1242 Université Rennes 1 - Centre de Lutte Contre le Cancer Eugène Marquis
Rue de la Bataille Flandres Dunkerque - CS 44229 35042 RENNES CEDEX

relationship between TGF β stimulation and Wnt pathway activation we have performed a gene set enrichment analysis (GSEA) using 4 well-characterized Wnt signaling pathway signatures (from MSigDB, UC San Diego and Broad Institute, www.gsea-msigdb.org) and the gene expression profiles of HuCCT1 cells treated with TGF β versus control. Supporting our initial hypothesis that TGF β could regulate Wnt signaling to some extent, GSEA demonstrated that all 4 tested Wnt signaling pathway signatures are positively and significantly enriched ($P < 0.01$) in the gene expression profiles of HuCCT1 cells treated with TGF β (please see the 4 GSEA plots below, included in the revised Appendix Figure S3). In addition, several master genes from the Wnt signaling pathway (e.g. TCF7, WNT11, WNT5B, WNT7A) are present in the core enrichment when performing GSEA, as exemplified below for the GOBP_CANONICAL_WNT_SIGNALING_PATHWAY.

Figure. Supervised GSEA using gene expression profiles of HuCCT1 cells exposed to TGF β and Wnt signaling pathway signatures. A) GSEA of curated Wnt signaling pathway signatures (from MSigDB) in the gene expression profiles of HuCCT1 cells treated with TGF β versus control (NES, normalized enrichment score; $P < 0.01$). GSEA demonstrated an enrichment of the 4 Wnt signatures in the gene expression profiles of HuCCT1 cells exposed to TGF β versus control. B) Core enrichment genes for the GOBP_CANONICAL_WNT_SIGNALING_PATHWAY signature.

In response to the referee's comment, Appendix Table S3 has been updated and now includes for each group comparison the following information: gene ID, p-value, FDR, geometric mean in each experimental group and fold-change ($n=3$ for each experimental group).

Oncogenesis Stress Signaling Laboratory

INSERM U1242 Université Rennes 1 - Centre de Lutte Contre le Cancer Eugène Marquis
Rue de la Bataille Flandres Dunkerque - CS 44229 35042 RENNES CEDEX

In figure S9A, the authors suggest a possible antagonistic role for LINC00313 in counteracting TGF β -mediated growth arrest (this assay should be shown as a growth curve). Is this finding supported by gene expression data? Further data mining could be used to propose a hypothesis presented in the discussion.

✓ Authors' response

In response to reviewer's comment, we have performed a proliferation assay using a live cell imaging Incucyte device. In addition, we also performed cell migration and invasion assays. The results demonstrated that LINC00313 overexpression resulted in increased cell viability, migration and invasion ($P < 0.001$), as shown below. These new results have been included in the revised Appendix Figure S6.

Figure. Functional proliferation, migration and invasion assays in control (pcDNA3.1) or LINC00313 over-expressing (pcLINC313) HuCCT1 cells (***, $P < 0.001$).

In addition, as suggested by the referee, data mining of the RNA-seq data from LINC00313 over-expressing HuCCT1 cells (Appendix Table S2) identified some known genes that are usually regulated by TGF β during cell cycle arrest. Indeed, we observed that the gene *CDKN1A*, which encodes the cyclin-dependent kinase (CDK) inhibitor p21, is repressed upon *LINC00313* over-expression. *CDKN1A* is one of the main genes, together with *CDKN2B* (p15) and *CDKN1B* (p27) induced by TGF β to promote cell cycle arrest (Heldin et al., 2009). In addition, we found two genes designated as inhibitor of DNA binding 1 (*ID1*) and inhibitor of DNA binding 3 (*ID3*), normally repressed by TGF β , to be up-regulated in LINC00313 over-expressing cells. These two genes encode for transcription factors that inhibit cell differentiation, while promoting cell proliferation and their repression by TGF β is important for eliciting its inhibitory effects on cell proliferation (Zhang et al., 2017). Accordingly, we have performed RT-Q-PCR experiments to validate gene expression profiling data. These new data demonstrate that LINC00313 overexpression results in a decreased expression of *CDKN1A* and an increased expression of *ID1* and *ID3*, as shown below. Therefore, these new data suggest that LINC00313 may counteract TGF β -mediated growth arrest by down-regulating p21 and

Oncogenesis Stress Signaling Laboratory

INSERM U1242 Université Rennes 1 -Centre de Lutte Contre le Cancer Eugène Marquis
Rue de la Bataille Flandres Dunkerque - CS 44229 35042 RENNES CEDEX

up-regulating ID1 and ID3 in HuCCT1 cells. The revised manuscript has been modified accordingly (please see the revised Appendix Figure S6B).

Figure. Real-time q-RT-PCR analysis of *CDKN1A*, *ID1* and *ID3* expression in control (pcDNA3.1) or LINC00313 over-expressing (pcLINC313) HuCCT1 cells (***, $P < 0.001$).

3. The proposed molecular mechanism involving LINC00313 mediated regulation of ACTL6A activity is not convincing for several reasons:

- ACTL6A was selected as candidate based on cherry picking of protein interactors revealed by MS analysis. Since ACTL6A is a SWI/SNF subunit, additional components of the complex would be expected to interact with LINC00313 in this type of experiment.

✓ **Authors' response**

We understand the Reviewer's concern. We agree that the proposed mechanism may represent only a part of the reality and that the molecular mechanisms by which LINC00313 regulates gene expression may be overall more complex. This is particularly true in the case of lncRNAs located and acting in both nucleus and cytosol. However, we do not share the same opinion with the Reviewer concerning the cherry picking approach to finally select ACTL6A. As described in Figure 6C (the full data are presented in Appendix Table S5 [datasheets 1 to 7], modified for clarity in the revised version of the manuscript), we followed a clear strategy to identify potential interesting protein interactors, based on the observation that LINC00313 lncRNA is predominantly nuclear in HuCCT1 cells. Thus, we first focused on nuclear interactors, then we utilized the catRapid algorithm to also predict the observed interaction and finally the proteins that passed these two filters were subjected to gene ontology analysis to focus on epigenetic regulators and/or transcription factors. This allowed us to focus on proteins such as ACTL6A, KLF4, TAF8 and TAF9 (Figure 6D). We then did a bibliography search to check whether any of the above proteins is associated with transcription regulation of the main LINC00313-target genes, such as TCF7, SULF2, AXIN2 or WNT5A. At the same time we went through the ATAC-seq data to check whether any of the identified

Oncogenesis Stress Signaling Laboratory

INSERM U1242 Université Rennes 1 - Centre de Lutte Contre le Cancer Eugène Marquis
Rue de la Bataille Flandres Dunkerque - CS 44229 35042 RENNES CEDEX

DNA binding motifs, at the open or closed chromatin regions (Appendix Figure S5), corresponds to a LINC00313 interactor protein. Indeed, two such proteins were the transcription factors KLF4 and KLF5. Finally, additional components of the SWI/SNF complex, such as SMARCA1 and SMARCA5, were identified in the MS analysis. However, they were also pulled-down (although less strong) by the unrelated F-luc RNA and therefore we did not choose to focus on them. We, however, validated an interaction between LINC00313 and endogenous BRG1 in HuCCT1 cells, using RIP-qPCR, as suggested by the reviewer below. We additionally evaluated the binding of BRG1 in the chromatin regions of *TCF7* and *SULF2*, particularly at the loci that were found to be in less compacted (more “open”) state upon LINC00313 over-expression from the ATAC-seq experiment. Notably, BRG1 binding at these loci was enhanced in LINC00313 over-expressing cells. We believe that these findings strengthen the hypothesis of a cooperative function of LINC00313 and the SWI/SNF complex towards the transcriptional regulation of specific genes.

- The validation of the supposed interaction by RNA pulldown and RIP assays is not convincing, given the barely detectable level of ACTL6A in pulldown extracts (6H) and the low LINC00313 enrichment in RIP-qPCR (6J – performed without appropriate positive and negative control RNAs). Since this might be due to the low efficiency observed for ACTL6A IP (6J), authors could try to improve their assay by using a different antibody for ACTL6A, the tagged version or by IP-ing a different subunit of the SWI/SNF complex (e.g., BRG1).

✓ **Authors' response**

We understand the Reviewer's concern about the robustness of the interaction between ACTL6A and LINC00313 lncRNA. In the experiment of Fig. 6H we tried to validate the data obtained after RNA pull-down followed by mass-spec, by immunoblotting for endogenous ACTL6A, using the protein preps as the ones used for MS. Thus, it is expected that the visualization of this interaction by western blot is weak, considering that ACTL6A was not among the top protein interactors found by mass-spec. Nevertheless, we were able to observe a weak but detectable interaction, especially in the presence of TGF β stimulation. For that reason, we repeated the experiment in HEK293T cells over-expressing HA-tagged ACTL6A (Fig. 6I). Using this over-expression system we were able to detect a stronger interaction between ACTL6A and *in vitro* synthesized LINC00313. The specific band that corresponds to ACTL6A and is observed only in HA-ACTL6A condition is the lower one and we have marked with asterisk the unspecific band that runs just above. In addition, following the Reviewer's suggestion, we also evaluated whether the catalytic subunit of the SWI/SNF complex

Oncogenesis Stress Signaling Laboratory

(BRG1) binds LINC00313 in HuCCT1 cells, even if we did not identify it as a hit in the initial MS analysis. The RIP-qPCR results suggested that LINC00313 interacts with BRG1 in HuCCT1 cells and TGF β treatment further enhances this interaction (Fig. 7L).

- The results obtained with ATPase inhibitors are highly misleading (7E-F): as SWI/SNF complexes regulate transcription of a significant fraction of the genome, a reduction in gene expression is expected (Iurlaro et al., Nat Genet 2021). Moreover, the authors show that SULF2 and TCF7 basal levels are already reduced in control cells, confirming that LINC00313 role is neglectable.

✓ **Authors' response**

We agree with the Reviewer's comment. Probably the inhibition of the SWI/SNF catalytic activity, using the ATPase inhibitor is too potent, leading to a strong down-regulation of *SULF2* and *TCF7* even in the control condition. For this reason we have moved these panels to the Appendix Figure S8. We kindly ask to keep these data in the Appendix section, as the results show the dependency of these genes to SWI/SNF-mediated regulation and also that the ATPase inhibitor is much more efficient than the bromodomain inhibitor, at least in this cell line.

- ACTL6A KD is very variable (7A&D) at protein level, weakening the author's conclusions.

✓ **Authors' response**

We agree with the Reviewer's point. Although we achieved a really efficient silencing of ACTL6A at the mRNA level (Fig. 7A, 7B, 7D), ACTL6A protein levels seem to be more stable, although to some extent reduced compared to cells transfected with a negative control siRNA. In the revised version we have also added a new graph with the ACTL6A mRNA levels after ACTL6A silencing, in the same conditions as the ones of the immunoblotting shown in Fig. 7F, which was missing.

In conclusion, ACTL6A role in LINC00313-mediated regulation of the Wnt pathway does not appear convincing. However, given the strong correlation between TCF7 and LINC00313 levels (5G) and the robust set of data connecting these two genes, the authors should explore whether other mechanisms could regulate this gene, since as a transcriptional activator of the Wnt pathway, it could serve as mediator of LINC00313 function.

✓ **Authors' response**

Oncogenesis Stress Signaling Laboratory

INSERM U1242 Université Rennes 1 - Centre de Lutte Contre le Cancer Eugène Marquis
Rue de la Bataille Flandres Dunkerque - CS 44229 35042 RENNES CEDEX

We thank the Reviewer for this suggestion. Although we think that the detected interaction between LINC00313 and ACTL6A has a functional role towards gene expression regulation of specific targets, such as *TCF7* and *SULF2*, following the Reviewer's advice we searched for alternative mechanisms that may explain the up-regulation of *TCF7* by LINC00313. Considering that 20% of LINC00313 transcripts show cytosolic localization in the cell lines investigated, and based on previous literature suggesting miRNA sponging functions for LINC00313 (please see below), we searched for miRNAs predicted or experimentally validated to bind to LINC00313 that also to target *TCF7* mRNA. For this reason, miRcode (<http://mircode.org/>) was used to identify miRNAs able to bind both LINC00313 lncRNA and *TCF7* mRNA. Thus, we identified 23 and 50 non-redundant miRNAs or miRNA clusters/families for LINC00313 and *TCF7*, respectively. From these, 16 miRNAs (individual or clusters) were found to target both LINC00313 and *TCF7* (see Figure below).

By searching the current literature for established associations between LINC00313 and miRNAs, with a special interest on the 16 aforementioned miRNAs, we saw that LINC00313 may induce cell migration and invasion, possibly by inducing β -catenin and by mediating EMT via sponging miR-138-5p, miR-150-5p, miR-204-5p and miR-205-5p, which target EMT-associated VIM and ZEB1 (Liu et al. *Bioengineered*, 2022). In addition, the literature suggests that *TCF7* could be transcriptionally regulated by β -catenin and TCF7L2 transcription factors (e.g. Roose et al. *Science*, 1999; Zhu et al. *Journal of Translational Medicine*, 2015). Interestingly, β -catenin was identified in 1 out of 3 pull-down experiments (FC=2.06 versus control) in the mass-spec analysis, suggesting that LINC00313 may somehow interact with β -catenin and modulate the activity of target genes, including *TCF7*. These points are now discussed in the revised version of the manuscript.

Oncogenesis Stress Signaling Laboratory

INSERM U1242 Université Rennes 1 - Centre de Lutte Contre le Cancer Eugène Marquis
Rue de la Bataille Flandres Dunkerque - CS 44229 35042 RENNES CEDEX

Minor points:

1. [92-93] The low number of DE genes in the initial gene expression array +/- TGF β (1A) is surprising, given the different results in S5. Is this due to high variability between cell lines? If so, this should be shown and justified. Also, Table S1 should report separate fold change for the two conditions and adj. p-values)

✓ **Authors' response**

We thank the Reviewer for her/his comment and we regret the misunderstanding. Actually, Fig. 1A (Appendix Table S1) refers to a specific set of genes commonly deregulated in both HuCCT1 and Huh28 cell lines upon TGF β treatment, while Fig. S5 (Appendix Table S3) refers to genes deregulated only in HuCCT1 cell line. Indeed, the 2 cell lines show a different phenotype, as we previously reported (Merdrignac et al. *Hepatol Commun.* 2018), explaining the lower number of DE genes when merging the profiles of the 2 cell lines. According to the Referee's comment, Appendix Table S1 has been updated in the revised manuscript and now includes 2 additional datasheets with statistical analyses for both HuCCT1 and Huh28 cell upon TGF β treatment.

2. [99-100] Assuming these primers were used to investigate LINC00313 different isoforms (based on hg38-T2T genome browsers), a schematic should be included in supplementary figures (possibly including siRNA binding sites).

✓ **Authors' response**

We thank the Reviewer for the comment. We have now added the requested scheme in the updated Appendix Figure S1B, where we show the precise regions amplified using each primer pair, as well as the siRNA binding sites for the individual siRNAs used later on in the study. In addition, we provide expression data of another isoform designated as LINC00313-205 in Ensembl database (ENST00000427188.1), which we found to be induced by TGF β in HuCCT1 cells, but not in Huh28 cell line, suggesting expression of different transcript variants, depending on the cellular context.

3. [113-116] WB showing SMADs KD would be beneficial, as different protein levels could explain the differences in the role of distinct SMADs in HuCCT1 vs Huh28. As general comment, this paragraph is rather confusing and not very insightful. While the authors demonstrate that SMAD-dependent signaling could regulate LINC00313, the role of the SMAD-independent pathway is unclear, in particular since the authors show by luciferase assay that MEK pathway signals through the SMAD response. Whereas feedback mechanisms connecting the two pathways could be present, investigating the

Oncogenesis Stress Signaling Laboratory

SMAD-independent pathway seems irrelevant. Instead, it would be interesting to analyze whether LINC00313 transcription is directly controlled by SMAD proteins by scanning LINC00313 promoter for SMAD-binding elements or performing ChIP.

✓ **Authors' response**

We thank the Reviewer for the constructive suggestion. We now provide the protein levels of SMAD2, SMAD3 and SMAD4, after their silencing in HuCCT1 cell line, in the new Appendix Figure S2, thereby confirming their efficient silencing not only at the mRNA, but also at the protein level. We also agree with the Reviewer concerning the role of SMAD-independent pathways in the regulation of LINC00313. We have therefore reconstructed this paragraph and deleted the data of the SBE-reporter luciferase assay of the panel B of the old Figure S3, which generates confusion. We still believe that the investigation of the non-canonical TGF β pathway in regulating *LINC00313* is relevant, because normally TGF β signals through both SMAD-dependent and SMAD-independent pathways. We thus added some extra data concerning the regulation of *LINC00313* specifically by p38 kinase, downstream of TGF β , as asked by the referee #2. However, we have reconstructed the relevant paragraph in the main text, as well as the legend of Figure 2, emphasizing only on the role of the SMAD-dependent pathway in regulating *LINC00313* expression. Additionally, we searched for SMAD-binding elements (SBEs) on the *LINC00313* upstream region, using the JASPAR Transcription Factor Binding Site track of the UCSC genome browser, a functionality that shows genome-wide predicted binding sites for transcription factors from the JASPAR CORE Collection. As shown in the new main Figure 2E, there are distinct SBEs spanning the proximal upstream region, the transcription start site (TSS) as well as the downstream region of *LINC00313* gene. Another interesting feature is the enhanced H3K27ac mark around *LINC00313* TSS, indicating that the chromatin in that region is “open” and can be actively transcribed.

4. [210-213] The discrepancy in TCF7 correlation/levels is very surprising. What is LINC00313 level in tumors from xenografted mice?

✓ **Authors' response**

We thank the reviewer for her/his comment. We have evaluated the LINC00313 levels and it seems that some of the xenografts, to some extent, lost their high LINC00313 levels, over the course of the 80 days, that the experiment was performed. We present here the relevant qPCR data for LINC00313 and TCF7.

Oncogenesis Stress Signaling Laboratory

INSERM U1242 Université Rennes 1 -Centre de Lutte Contre le Cancer Eugène Marquis
Rue de la Bataille Flandres Dunkerque - CS 44229 35042 RENNES CEDEX

Figure. *LINC00313* and *TCF7* expression levels, normalized to *GAPDH*, using RNA samples from tumors developed in pcDNA3 and pcLINC00313 xenografted mice. The pcDNA3 samples are indicated with blue color, while the pcLINC00313 samples with red color.

5. [Figure 8A] This figure is not informative, as it doesn't show whether the genes up and down regulated in xenografted mice reflect *in vitro* analyses. Substituting the nameless heatmap with more insightful GSEA would be useful.

✓ **Authors' response**

We thank the Reviewer for her/his comment and we totally agree with it. In the revised Figure 8 we have substituted the heatmap by a Volcano Plot that now indicates some of the key genes identified that reflect the *in vitro* analysis, as shown below.

Figure. Volcano Plot of genes from experiments in xenografted mice. Genes up- and down-regulated in tumors derived from HuCCT1 cell overexpressing pcLINC313 versus pcDNA3 control vector are indicated in red and blue, respectively.

Oncogenesis Stress Signaling Laboratory

INSERM U1242 Université Rennes 1 - Centre de Lutte Contre le Cancer Eugène Marquis
Rue de la Bataille Flandres Dunkerque - CS 44229 35042 RENNES CEDEX

In addition, we have performed a GSEA using the gene signatures established *in vitro* (i.e. genes up and down regulated in HuCCT1 cells overexpressing pcLINC313 versus pcDNA3 control, as described in Figure 3B) and the gene expression profiles established *in vivo* (i.e. in tumors from xenografted mice). GSEA demonstrated that the differentially regulated genes identified in the *in vitro* experiments were significantly enriched in the gene expression profiles of the *in vivo* experiments, as expected.

Figure. GSEA using gene signatures established *in vitro* (UP and DN regulated genes) and the gene expression profiles established *in vivo* (i.e. in tumors from xenografted mice derived from HuCCT1 cells overexpressing pcLINC313 versus pcDNA3 control vector).

6. Please, add statistics in figures where needed.

✓ **Authors' response**

We thank the Reviewer for her/his comment. Statistical analysis has now been performed for the missing panels and statistical significance is shown with asterisks, based on p-values calculated using Student's t-test. Information about the number of biological replicates, error bars and the test to calculate p-values is added in the figure legends for the respective panels.

Again, we thank Referee #1 for her/his comments and critical review of our study.

Oncogenesis Stress Signaling Laboratory

INSERM U1242 Université Rennes 1 -Centre de Lutte Contre le Cancer Eugène Marquis
Rue de la Bataille Flandres Dunkerque - CS 44229 35042 RENNES CEDEX

Point-by-point response to the Referee #2's comments

In this work, Papoutsoglou and cols have explored the role of the Long non-coding RNA LINC00313 in cholangiocarcinoma (CCA). They identify it as a novel target of the TGF-beta pathway in CCA cells. Through elegant gene expression analysis and genome-wide chromatin accessibility profiling, authors demonstrate that LINC00313 in the nucleus transcriptionally regulates genes involved in Wnt signaling. In vivo experiments, analysis with human tissues and mechanistic studies led to the authors to propose a model whereby TGF-beta induces LINC00313 to regulate expression of hallmark Wnt genes in cooperation with SWI/SNF, which would boost cholangiocarcinogenesis. Work has been well designed and performed. The manuscript is clear, well-presented figures and excellent text. Overall, the results open new perspectives on the molecular mechanisms responsible for cholangiocarcinogenesis. Nevertheless, I have some comments/recommendations to improve the manuscript.

✓ **Authors' response**

We thank the reviewer for her/his positive opinion towards our manuscript and for sharing the opinion that our study could open new perspectives on the molecular mechanisms that drive cholangiocarcinogenesis.

MAJOR POINTS

1. Results strongly support a role for TBRI/SMAD pathway in the induction of LINC00313 expression. However, when other non-canonical pathways are analysed, results are not so clear. Authors indicate a strong role for p38, whereas other signals, such as ERK or JNK, could be inhibiting the TGF-beta induction of LINC00313, due to its potential to counteract SMAD signaling. However, these results are only observed in the HuCCT1, not so clear in the Huh28 cells. Furthermore, the p38 inhibitor is the only one that is not analysed in the Figure S3 (probably because it is necessary to analyse p38 activity). And, relevant, p38 inhibitor strongly inhibits AKT phosphorylation, even at higher levels than the PI3Ki. To firmly demonstrate that p38 is mediating the regulation of LINC00313 by TGF-beta, it is necessary to do specific experiments downregulating p38 with siRNA, as performed with SMADs. And it would be necessary to check this point also in the Huh28 cells, or other alternative CCA cells.

✓ **Authors' response**

We thank the Reviewer for her/his comment. We totally understand and agree with the Reviewer's suggestion. Since there are four isoforms of the p38 kinase i.e. p38 α (encoded by *MAPK14* gene), p38 β (encoded by *MAPK11* gene), p38 γ (encoded by

Oncogenesis Stress Signaling Laboratory

INSERM U1242 Université Rennes 1 - Centre de Lutte Contre le Cancer Eugène Marquis
Rue de la Bataille Flandres Dunkerque - CS 44229 35042 RENNES CEDEX

MAPK12 gene) and p38 δ (encoded by *MAPK13* gene), we have now silenced each one of the four p38 isoforms with specific siRNAs, as shown in the updated Figure EV2C. We confirmed the silencing of these genes both at the mRNA and at the protein levels (EV2C, EV2D). Interestingly, *LINC00313* expression was not significantly affected by silencing individual p38 isoforms in any of the two cell lines tested (HuCCT1 and Huh28). A possible explanation for this observation is that the rest of the active p38 isoforms compensate for the loss of a single p38 isoform in the knockdown experiment. That would explain why we observed an effect when using the p38 inhibitor, which can target any p38 isoform expressed in the cells. Nonetheless, these results prompted us to reconsider the importance of the p38 kinase in regulating *LINC00313* downstream of TGF β . Therefore, we have re-phrased the relevant paragraph in the main text, as well as in the legend of Figure 2, emphasizing on the role of the SMAD-dependent pathway in regulating *LINC00313* expression.

We also apologize for not including the appropriate control for the confirmation of the inhibition of p38 kinase activity. Upon p38 activation, p38 is phosphorylated and, in turn, it phosphorylates target substrates, such as the MAP kinase-activated protein kinase 2 (MAPKAPK-2). In the revised version of the manuscript, we have performed an immunoblotting to evaluate the phosphorylated levels of p38 on Thr180/Tyr182 as well as the phosphorylated levels of the direct target of p38 kinase, MAPKAPK-2 (on Thr334), after treatment with p38 inhibitor. Both of them seem to be reduced after p38 inhibition only, but they are not affected by inhibiting the rest of the kinases, confirming that SB203580 efficiently inhibited p38 activity.

2. The functional analysis for the role of *LINC00313* on proliferation and migration is not conclusive. Effects on cell viability are very modest. The analysis on cell migration, only through a wound healing assay, is insufficient. And these experiments are essential to support the results later presented in the *in vivo* model. I suggest doing specific experiments of progression in cell cycle, such as BrdU or analysis of cell cycle by flow cytometry. And for migration, it is necessary to work in a two-chamber plate, where authors may visualize cell migration or invasion, modulating the matrix composition between the two chambers.

✓ **Authors' response**

We thank the Reviewer for this comment. As indicated to Reviewer #1, we have strengthened the functional analysis by performing cell proliferation, migration and invasion assays using a live cell imaging Incucyte device. The results demonstrated that *LINC00313* overexpression resulted in increased cell viability, migration and invasion ($P < 0.001$), as shown below.

Oncogenesis Stress Signaling Laboratory

INSERM U1242 Université Rennes 1 - Centre de Lutte Contre le Cancer Eugène Marquis
Rue de la Bataille Flandres Dunkerque - CS 44229 35042 RENNES CEDEX

Figure. Functional proliferation, migration and invasion assays in control (pcDNA3.1) or LINC00313 over-expressing (pcLINC313) HuCCT1 cells (***, $P < 0.001$).

3. A point that is difficult to understand and that could compromise the model proposed by the authors is that in the *in vivo* experiments (Fig. 4) the tumors from xenografted pcLINC00313 cells do not show increase in TCF7 expression, although these cells presented higher mRNA levels *in vitro* (Fig. 3F) and a perfect correlation between TCF7 and LINC00313 is observed *in vivo* (Fig. 4G). Authors show that protein is increased, but it is only clear in 2-3 of the 6 tumors analysed (densitometric analysis and statistics is required). How LINC00313 could regulate the levels of TCF7 protein? Authors must discuss this point.

We thank the Reviewer for the comment. As the Reviewer #1 also raised a similar point, we believe that a possible explanation is the compromised LINC00313 overexpression in particular xenografted mice at the end of the experiment. It seems that some of the xenografts, to some extent, lost their high LINC00313 levels, over the course of the 80 days, during the development of the tumors and until the sacrifice of the mice. Since we included all 6 over-expressing samples and the 7 control samples for the qPCR and WB analyses, this could explain the lack of TCF7 mRNA increase and to some extent the compromised TCF7 protein increase. As far as the *in vivo* TCF7 immunoblotting concerns we have now performed densitometric analysis to show more clearly the differences in TCF7 protein expression between the samples. We apologize for not including statistics, but the immunoblotting was performed once, therefore statistics cannot be applied. It is also worth to note that *SULF2* and *AXIN2* mRNAs remained higher in the pcLINC xenografts compared to control, but this could be due to the different degradation rates of these mRNAs. In addition, the total tumor volume was bigger in the pcLINC xenografts, although that was apparent towards the final days of the *in vivo* experiment and not from the beginning. Overall, to increase the strength of the *in vivo* findings a higher number of animals would be needed with perhaps a different strategy for the generation of stable LINC00313 overexpressing xenografts, with the use of lentiviral constructs.

Oncogenesis Stress Signaling Laboratory

INSERM U1242 Université Rennes 1 - Centre de Lutte Contre le Cancer Eugène Marquis
Rue de la Bataille Flandres Dunkerque - CS 44229 35042 RENNES CEDEX

4. In parallel, in the TCGA dataset, LINC00313 was neither overexpressed in CCA tissues nor correlated with overall or disease-free survival (Fig. S11A). Authors then focus on the analysis of LINC00313 "activity". Design of the experiment is correct, however, the *in vivo* gene signature, which correlates with a differential overall survival, is a gene signature extremely related to other signaling pathways. How could authors demonstrate, or at least discuss, that the "activity" of LINC00313 could be modulated in some CCA patients?

We thank the Reviewer for this comment. Indeed, querying the GEPIA2 database, we could not find a statistically significant correlation between LINC00313 levels and patient survival in CCA, although this is not the case in different cancer types. This could be attributed to the low patient number of this cohort. However, using the Xena Functional Genomics Explorer from the UCSC (<https://xenabrowser.net/heatmap/>) that contains higher number of patients we observed a significant inverse correlation between LINC00313 expression levels and survival probability in CCA patients. Moreover, survival analysis at the PanCancer level (all tumors, from every cancer type included) showed that high LINC00313 expression correlates with lower overall survival for cancer patients (please see the Figure below and the new Figure EV5A).

Figure. Kaplan Meier plot analysis of overall survival datasets of patients with cholangiocarcinoma (left panel) and or Pan-Cancer dataset (right panel) grouped based on LINC00313 median expression (RNAseq data).

MINOR POINTS

5. Page 5, line 104. Here authors cite SNORD48 gene, whereas in the figure another alias, RNU48, is included. I suggest homogenising the names.

Oncogenesis Stress Signaling Laboratory

INSERM U1242 Université Rennes 1 -Centre de Lutte Contre le Cancer Eugène Marquis
Rue de la Bataille Flandres Dunkerque - CS 44229 35042 RENNES CEDEX

✓ **Authors' response**

We thank the Reviewer for spotting this mistake. We have corrected the name of the gene in the text, replacing SNORD48 with RNU48.

6. Figure S8E: it is difficult to extrapolate conclusions from this experiment. I would recommend complementing it with analysis of Western blots after isolating nuclei. It is very important to clarify this point.

✓ **Authors' response**

We agree with the Reviewer's suggestion. We have, thus, performed a subcellular fractionation, followed by protein isolation from nuclear and cytosolic fractions to evaluate the localization of β -catenin with or without CHIR treatment for 24 hours in empty vector (control) and LINC00313 over-expressing HuCCT1 cells (Figure EV4H). In control cells, treatment with CHIR modestly increased the nuclear presence of β -catenin. The absence of a strong nuclear translocation of β -catenin could be due to the chosen time period for CHIR treatment. Optimization would be needed to evaluate the best time period for CHIR to induce maximum nuclear β -catenin levels. Over-expression of LINC00313 did not have any effect on β -catenin translocation, as observed in the immunofluorescence experiment, suggesting that LINC00313 fine-tunes the Wnt pathway downstream, at the transcriptional or epigenetic levels and not at the initial stages of Wnt pathway activation. Nuclear and cytosolic markers are also shown to confirm the clarity of the fractions.

7. Fig. 7C-F: I would recommend including statistical analysis.

✓ **Authors' response**

We thank the Reviewer for her/his comment. Statistical analysis has now been performed for the indicated graphs and statistical significance is shown with asterisks, based on p-values calculated using Student's t-test.

8. Considering the strong effects of BRM/BRG1 ATPi on TCF7 expression in TGF-beta-treated cells (Fig. 7F), why do authors analyse the effect of this inhibitor on CHIR-induced TCF/LEF luciferase reporter activity, instead on TGF-beta treated cells (Fig. 7G)?

✓ **Authors' response**

We thank the Reviewer for her/his comment. The reason for analyzing the effect of BRM/BRG1 ATPase inhibitor on TCF/LEF luciferase activity in CHIR-treated cells is that we wanted to induce the TCF/LEF luciferase activity, which at the basal levels is low in

Oncogenesis Stress Signaling Laboratory

INSERM U1242 Université Rennes 1 -Centre de Lutte Contre le Cancer Eugène Marquis
Rue de la Bataille Flandres Dunkerque - CS 44229 35042 RENNES CEDEX

HuCCT1 cell line. Therefore, by inducing TCF/LEF luciferase activity we were able to observe the inhibitory effect of BRM/BRG1 ATPase inhibitor on the TCF/LEF luciferase activity. Although stimulation of cells with 1 ng/ml TGF β results in induction of several Wnt pathway genes, this dose may not be sufficient to cause an augmentation of TCF/LEF luciferase activity and, thus, we chose to do this experiment in CHIR-treated cells.

Oncogenesis Stress Signaling Laboratory

INSERM U1242 Université Rennes 1 -Centre de Lutte Contre le Cancer Eugène Marquis
Rue de la Bataille Flandres Dunkerque - CS 44229 35042 RENNES CEDEX

Point-by-point response to the Referee #3's comments

Here the authors investigated the role of the linc00131 and TGFbeta in cholangiocarcinoma (CCA). LINC00313 lncRNA is a target of TGFβ signalling in CCA cells. LINC00313 is nuclear and transcriptionally regulated genes involved in Wnt signalling (TCF7). LINC00313 expression enhanced TCF/LEF-dependent target genes, associated to tumor formation. And decrease inpatient's overall survival. The authors found that ACTL6A interacts with LINC00131 controlling TCF7 and SULF2 transcription. They propose a working model in which TGFbeta induces LINC00313 in to regulate expression of Wnt pathway genes, in co-operation with SWI/SNF complex. The study is potentially interesting, and the conclusions are supported by the experiment shown.

✓ **Authors' response**

We thank the Reviewer for finding our study potentially interesting and for appreciating that there is a direct agreement between our conclusions and the experimental data that support them.

The authors should address the following points:

1) Explain why the authors choose to study LINC00313. At basal level (Fig.1B), LINC00313 is expressed at different levels in iCCA/eCCA cell lines, indicating that its role is not necessary for the biology of the tumors, how the authors explain this point? Also, not all iCCA LINC00313 expression is TGFbeta dependent. The authors should also indicate the RTqPCRs DeltaCt to give information on the abundance of lincRNA. Is LINC00131 undergoing maturation? A characterization of this lincRNA should also be shown.

✓ **Authors' response**

We thank the reviewer for her/his comment. The main reason to choose LINC00313 lncRNA for further studies was its induction by TGFβ in cholangiocarcinoma cell lines, as well as the current lack of understanding about its role in cholangiocarcinoma.

As shown in Table S1, there were only 3 LINC-RNAs commonly induced by TGFβ in HuCCT1 and Huh28: LINC00312, LINC00313 and LINC00340. We already characterized the role of LINC00340 in CCA (Merdrignac et al, *Hepatol Commun.* 2018) and we are currently investigating the role of LINC00312. In the present report, we are presenting the role of LINC00313. Our study is the first to describe LINC00313 as a novel target of TGFβ signaling in CCA. In addition, to date, there are very few descriptive reports investigating the role of LINC00313 in different cancer types, but without any deep investigation on its molecular mechanisms of action.

Oncogenesis Stress Signaling Laboratory

INSERM U1242 Université Rennes 1 -Centre de Lutte Contre le Cancer Eugène Marquis
Rue de la Bataille Flandres Dunkerque - CS 44229 35042 RENNES CEDEX

Another interesting observation is the selectivity of the TGF β -mediated regulation of LINC00313 in different CCA cell lines. A possible explanation is the lack of responsiveness to TGF β in some of these cell lines. For example, preliminary results (not yet published) indicate that in CCLP1 cells the canonical TGF β pathway is disrupted because of the lack of SMAD3 expression. On the other hand the cell line SG231 is characterized by low SMAD3 and high SMAD7 (a negative regulator of the signaling) expression, resulting possibly to the observed lack of TGF β response. Mutation of TGFBR1 is also reported for Egi-1 cell line. A deep characterization of TGF β signaling activation in multiple CCA cells lines has not yet been done and it is absolutely necessary to understand the effects of TGF β in CCA.

The basal LINC00313 expression levels are moderate to high in the CCA cell lines used in this study, and they are clearly elevated in response to TGF β stimulation, in the responsive cell lines, as shown in the DeltaCt values, which are available with the source data files (see excel files, Figure 1B). For example, in HuCCT1 untreated cells LINC00313 DeltaCt (Ct_{LINC00313}-Ct_{TBP})=6, while in TGF β -stimulated cells DeltaCt=3.5. In the rest of the cell lines the values are indicated below:

Huh28 control: DeltaCt=6.5
Huh28+TGF β : DeltaCt=5
CCLP1 control: DeltaCt=9.5
CCLP1+TGF β : DeltaCt=12
Egi-1 control: DeltaCt=7
Egi-1+TGF β : DeltaCt=6.5
SG231 control: DeltaCt=6.5
SG231+TGF β : DeltaCt=6.5
SkChA1 control: DeltaCt=5
SkChA1+TGF β : DeltaCt=4.3
MzChA1 control: DeltaCt=2.8
MzChA1+TGF β : DeltaCt=2.8
TFK1 control: DeltaCt=4.2
TFK1+ TGF β : DeltaCt=4
NHC control: DeltaCt=7
NHC+TGF β : DeltaCt=5

Concerning a general characterization of *LINC00313* gene, we provide information about its genomic localization, as well as its expression in normal human tissues (from NCBI) and in different cancer types (from TCGA) in Figure EV1. Interestingly,

Oncogenesis Stress Signaling Laboratory

INSERM U1242 Université Rennes 1 -Centre de Lutte Contre le Cancer Eugène Marquis
Rue de la Bataille Flandres Dunkerque - CS 44229 35042 RENNES CEDEX

LINC00313, like several other lncRNAs, shows tissue-specific expression, with highest expression notably in the liver (Figure EV1B). In addition, among 29 different cancer types from the PanCancer Atlas, hepatocellular carcinoma (LIHC) scores at the top and cholangiocarcinoma (CHOL) the third from the top concerning LINC00313 expression. We also provide information about the presence of possible small ORFs within LINC00313 sequence, possibly leading to peptide generation, utilizing the ORF finder software. Moreover, we show in Appendix Figure S1B structural details of the LINC00313 Refseq transcript variant, consisting of 4 exons and 3 introns. In addition, in Ensembl database different LINC00313 splice variants are annotated, indicating that this lncRNA undergoes alternative splicing. Our RNA-seq data also suggest that LINC00313 possibly undergoes maturation in HuCCT1 cells.

2) Fig 2: controls by western blot of silencing should be included.

✓ **Authors' response**

In response to Reviewer's request, we have now added the requested controls, shown in the new Appendix Figure S2B.

3) Nuclear localization should also be confirmed by FISH experiments in cell lines +/- TGFβ.

✓ **Authors' response**

We thank the Reviewer for her/his suggestion. We now provide experimental data in the new Figure EV3 concerning the subcellular localization of LINC00313 transcripts, using smiFISH, after designing specific probes for LINC00313 lncRNA detection. We performed this experiment in the LINC00313 over-expressing system, as it was easier to visualize the specific RNA FISH signal, especially in the condition where cells are stimulated with TGFβ. We detected several nuclear spots but also a few cytoplasmic spots, thereby confirming the nucleo-cytoplasmic fractionation experiments.

4) siRNA used for LINC00313 silencing should be better characterized to demonstrate that no off target occurred.

✓ **Authors' response**

We thank the Reviewer for her/his important comment to clarify the specificity of our siRNA against LINC00313. In order to exclude off-target effects of the siRNA used to silence *LINC00313*, we have performed nucleotide BLAST (BLASTn) to identify potentially non-specific (off target) sequences for the siRNA targeting LINC00313 in the human genome. We did not find any of the main genes, suggested in this study as

Oncogenesis Stress Signaling Laboratory

INSERM U1242 Université Rennes 1 -Centre de Lutte Contre le Cancer Eugène Marquis
Rue de la Bataille Flandres Dunkerque - CS 44229 35042 RENNES CEDEX

LINC00313-targets (TCF7, SULF2, AXIN2 etc.) in the list of the potential off targets (Appendix Figure S4C). However, we found that a large part of LINC00313 transcript shares high degree of similarity to another lncRNA, designated as long intergenic non-protein coding RNA 1669 (LINC01669) or alternatively LOC102724354. Our siRNA targets exon 3 of LINC00313 (nucleotides 266 to 284) and, therefore, also LINC01669. However, according to NCBI this annotation is considered as a false duplication and thus likely redundant with LINC00313. In addition, LINC01669 is annotated as ENSG00000280191 in the Ensembl genome browser (genomic location: chr21: 6,060,340-6,076,305), but again is noted as artifact duplication of ENSG00000185186 (LINC00313). Another possible hit that could be targeted by our siRNA against LINC00313 is the heat shock transcription factor 2 binding protein (HSF2BP), a gene that also resides in the LINC00313 locus. However, our gene expression analysis did not reveal any differential expression of HSF2BP upon *LINC00313* silencing (see Appendix Table S3). Moreover, in the *in vitro* transcriptomic analysis after *LINC00313* gain of function, HSF2BP was slightly induced, but not with statistical significance (see Appendix Table S2).

Nevertheless, in order to minimize possible doubts we designed a second individual siRNA, which we designated as siLINC00313#2, that targets *LINC00313* in a region that does not overlap with LINC01669 (exon 1, nucleotides 33 to 51, see Appendix Figure S4D). Silencing *LINC00313* with each one of these two siRNAs yielded similar effects on target gene expression in Huh28 and TFK-1 cell lines, as shown in Appendix Fig S4A, S4B, S4E and S4F.

5) The authors should perform FISH experiments to detect LINC00131 in CCA tissue specimens.

We thank the Reviewer for her/his comment. Unfortunately, we have only access to FFPE human CCA tissues that are not optimal to perform RNA FISH, as the overall RNA integrity is compromised. We provide the QC of the RNA samples from the different tissues to show the low RNA integrity (Figure below). Instead FISH was performed on CCA cell lines and gene expression profiling was used to characterize the expression of LINC00313 in human CCA tissues (as shown in Figure EV5).

Oncogenesis Stress Signaling Laboratory

INSERM U1242 Université Rennes 1 -Centre de Lutte Contre le Cancer Eugène Marquis
Rue de la Bataille Flandres Dunkerque - CS 44229 35042 RENNES CEDEX

Figure. QC (gel electrophoresis) of total RNA extracted from human FFPE CCA samples. M: size marker; FFPE: RNA extracted from 3 independent FFPE human CCA showing a smear (degraded RNAs); C: positive control (RNA extracted from HuCCT1 cell line) showing intact bands for ribosomal RNAs.

6) Correlation in datasets of LINC00131-associated genes expression in CCA patients.

✓ Authors' response

We thank the Reviewer for this comment. Accordingly, we report now a significant correlation ($p < 0.05$) between the expression of *LINC00313* and *TCF7* and *AXIN2* in a set of 39 human iCCA. These results are incorporated in Figure EV5.

Spearman r	0.4433
95% confidence interval	0.1390 to 0.6711
P value (two-tailed)	0.0047

Spearman r	0.5221
95% confidence interval	0.2381 to 0.7238
P value (two-tailed)	0.0007

Oncogenesis Stress Signaling Laboratory

INSERM U1242 Université Rennes 1 - Centre de Lutte Contre le Cancer Eugène Marquis
Rue de la Bataille Flandres Dunkerque - CS 44229 35042 RENNES CEDEX

Dear Dr. Coulouarn,

Thank you for the submission of your revised manuscript to our editorial offices. I have now received the reports from two of the three referees that I asked to re-evaluate your study, you will find below. Referee #3 was completely unresponsive to my invitations to re-assess the study. However, going through your p-b-p-response, I consider her/his points as adequately addressed. As you will see, the other two referees now support the publication of the study in EMBO reports. Referee #1 has a remaining point (regarding the description of the TCGA analysis), I ask you to address in a final revised manuscript.

Moreover, I have these editorial requests I ask you to address:

- Please provide the abstract written in present tense throughout.
- Please reduce the number of keywords to 5.
- Please make sure that the number "n" for how many independent experiments were performed, their nature (biological versus technical replicates), the bars and error bars (e.g. SEM, SD) and the test used to calculate p-values is indicated in the respective figure legends (for main, EV and Appendix figures) of the final revised manuscript. Please also check that all the p-values are explained in the legend, and that these fit to those shown in the figure. Please provide statistical testing where applicable. Please avoid the phrase 'independent experiment', but clearly state if these were biological or technical replicates. Please also indicate (e.g. with n.s.) if testing was performed, but the differences are not significant. In case n=2, please show the data as separate datapoints without error bars and statistics. See also:
<http://www.embopress.org/page/journal/14693178/authorguide#statisticalanalysis>

If n<5, please show single datapoints for diagrams. Single data points are mostly missing in the bar diagrams. Also the statistics seems incomplete (see e.g. Figs. 1E, 7L, EV1B, EV3C and EV4H). Moreover:

- Please note that the box plots need to be defined in terms of minima, maxima, centre, bounds of box and whiskers, and percentile in the legends of figures 5h; 8f; EV5b.
- Please note that information related to n is missing in the legends of figures 6j; 8f; EV2a; EV5b.
- Please note that the error bars are not defined in the legends of figures 6j; EV2a.
- Please note that n=2 in figure EV2d.
- Please indicate the statistical test used for data analysis in the legends of figures 3b-d; EV2a, d; EV5a, b.
- Please define the annotated p values ***/**/* in the legends of figures EV2a, d as appropriate.
- Please add to each legend a 'Data Information' section explaining the statistics used or providing information regarding replicates and scales. See:

- Appendix Tables S1-S5 are datasets. Please upload the original excel files as dataset, with a legend on the first TAB and name the file Dataset EVx. Finally, please update the callouts ('Dataset EVx') for these in the main manuscript text file and remove their legends from the Appendix.
- Accordingly, please rename Appendix Tables S6-S8 to Appendix Tables S1-S3 and change their callouts in the manuscript text file.
- Please remove the sections 'Supporting Information' and 'Expanded view' (before the references) from the manuscript text file.
- Please make sure that all the funding information is also entered into the online submission system and that it is complete and similar to the one in the acknowledgement section of the manuscript text file. Presently, R22026NN, R21011NN, R21043NN, C18007NS and C20013NS seem missing in the acknowledgements.
- During our standard image analysis, we detected potential aberrations in the figure set, and we would like to clarify these issues: Please provide source data (uncropped blots) for the Western blots shown in Appendix Fig. S2B.

In addition, I would need from you:

- a short, two-sentence summary of the manuscript (not more than 35 words).
- two to four short (!) bullet points highlighting the key findings of your study (two lines each).
- a schematic summary figure that provides a sketch of the major findings (not a data image) in jpeg or tiff format (with the exact width of 550 pixels and a height of not more than 400 pixels) that can be used as a visual synopsis on our website.

I look forward to seeing the final revised version of your manuscript when it is ready. Please let me know if you have questions

regarding the revision.

All the best,

Referee #1:

The revised manuscript by Papoutsoglou et al. greatly improved compared to the initial version. I have appreciated the author's effort to expand the method section and provide technical and statistical details. Despite some minor points remain (e.g. TCGA analysis not described), overall the manuscript provides multiple evidence characterizing LINC00313 in CCA and is suitable for publication in this journal.

Referee #2:

Authors have correctly addressed the concerns and modified the manuscript properly, including a lot of new experiments. From my part, paper is ready for publication.

Manuscript number: EMBOR-2022-56179V2

Papoutsoglou et al. TGF β -induced long non-coding RNA LINC00313 activates Wnt signalling and promotes cholangiocarcinoma.

Point-by-point response to the Editor's comments

Dear Dr. Coulouarn,

Thank you for the submission of your revised manuscript to our editorial offices. I have now received the reports from two of the three referees that I asked to re-evaluate your study, you will find below. Referee #3 was completely unresponsive to my invitations to re-assess the study. However, going through your p-b-p-response, I consider her/his points as adequately addressed. As you will see, the other two referees now support the publication of the study in EMBO reports. Referee #1 has a remaining point (regarding the description of the TCGA analysis), I ask you to address in a final revised manuscript.

✓ **Authors' response**

We thank the Editor for giving us the opportunity to re-submit a final revised version of our manuscript. In the revised manuscript we have addressed all the editorial requests, including the only remaining comment of Referee #1 and we have performed a final experiment regarding the data shown in Appendix Figure S2.

Moreover, I have these editorial requests I ask you to address:

- Please provide the abstract written in present tense throughout.

✓ **Authors' response**

In response to editor's request we have now corrected and written the abstract in the present tense.

- Please reduce the number of keywords to 5.

✓ **Authors' response**

In response to editor's request we have reduced the number of keywords to 5.

- Please make sure that the number "n" for how many independent experiments were performed, their nature (biological versus technical replicates), the bars and error bars (e.g. SEM, SD) and the test used to calculate p-values is indicated in the respective figure legends (for main, EV and Appendix figures) of the final revised manuscript. Please also check that all the p-values are explained in the legend, and that these fit to those shown in the figure. Please provide statistical testing where applicable. Please avoid the phrase 'independent experiment', but clearly state if these were biological or technical

replicates. Please also indicate (e.g. with n.s.) if testing was performed, but the differences are not significant. In case n=2, please show the data as separate datapoints without error bars and statistics. See also:

<http://www.embopress.org/page/journal/14693178/authorguide#statisticalanalysis>

If n<5, please show single datapoints for diagrams. Single data points are mostly missing in the bar diagrams. Also the statistics seems incomplete (see e.g. Figs. 1E, 7L, EV1B, EV3C and EV4H).

✓ **Authors' response**

In response to editor's request, we have now re-graphed all the bar diagrams for which n<5 (mainly RT-qPCR experiments and luciferase and colony formation assays) for main and EV figures. In addition, we regret to have forgotten the statistics of the indicated panels, which are now performed and completed, with the exception of panel EV1B, which derives from RNA-seq data publicly available at NCBI, therefore it is not possible to add error bars and perform statistics.

Moreover:

- Please note that the box plots need to be defined in terms of minima, maxima, centre, bounds of box and whiskers, and percentile in the legends of figures 5h; 8f; EV5b.

✓ **Authors' response**

We have now defined the box plots as suggested for the figures 5h and 8f, however this was not possible for the panel EV5b, as the graphs derive from RNA-seq data from NCBI and are generated using GEPIA2. Therefore we did not have access to the raw values, in order to calculate the requested values.

- Please note that information related to n is missing in the legends of figures 6j; 8f; EV2a; EV5b.

✓ **Authors' response**

We have updated the information related to the number of replicates in the requested figure legends.

- Please note that the error bars are not defined in the legends of figures 6j; EV2a.

✓ **Authors' response**

We have now defined the error bars in the legends of these panels.

- Please note that n=2 in figure EV2d.

✓ **Authors' response**

We have now added the number of biological replicates and updated the graphs according to the journal guidelines, without error bars and statistics.

- Please indicate the statistical test used for data analysis in the legends of figures 3b-d; EV2a, d; EV5a, b.

✓ **Authors' response**

We now mention the statistical test used for the requested figure legends.

- Please define the annotated p values ***/**/* in the legends of figures EV2a, d as appropriate.

✓ **Authors' response**

The p-values are now defined in the indicated figure legends.

- Please add to each legend a 'Data Information' section explaining the statistics used or providing information regarding replicates and scales. See:

✓ **Authors' response**

We have now added a 'Data Information' section at the end of each figure legend, providing information about the statistics used for each analysis.

- Appendix Tables S1-S5 are datasets. Please upload the original excel files as dataset, with a legend on the first TAB and name the file Dataset EVx. Finally, please update the callouts ('Dataset EVx') for these in the main manuscript text file and remove their legends from the Appendix.

✓ **Authors' response**

In response to editor's request we have now corrected these datasets and renamed their callouts in the main manuscript text as instructed.

- Accordingly, please rename Appendix Tables S6-S8 to Appendix Tables S1-S3 and change their callouts in the manuscript text file.

✓ **Authors' response**

In response to editor's request we have now renamed the Appendix Tables and changed their callouts in the manuscript text file accordingly.

- Please remove the sections 'Supporting Information' and 'Expanded view' (before the references) from the manuscript text file.

✓ **Authors' response**

We have now removed these sections from the manuscript text file.

- Please make sure that all the funding information is also entered into the online submission system and that it is complete and similar to the one in the acknowledgement section of the manuscript text file. Presently, R22026NN,

R21011NN, R21043NN, C18007NS and C20013NS seem missing in the acknowledgements.

✓ **Authors' response**

We thank the editor for pointing out this issue. We have now updated all the funding information in the manuscript text file, under the acknowledgement section.

- During our standard image analysis, we detected potential aberrations in the figure set, and we would like to clarify these issues: Please provide source data (uncropped blots) for the Western blots shown in Appendix Fig. S2B.

✓ **Authors' response**

This experiment was performed twice and we provide the uncropped blots from these experiments, together with the rest of the source data files. Below, we present the results from one of the two biological experiments (the results of the second experiment are shown in the new Appendix FigS2B).

Figure next page. Immunoblotting of SMAD2/3 or SMAD4, after silencing SMAD2, SMAD3 or SMAD4, or simultaneously SMAD2/3/4 in HuCCT1 cells treated or not with TGF β for 16h. Beta-actin was used as a loading control and a scrambled siRNA as a control for silencing.

Quantification: band intensity/actin band intensity then Quantification condition of interest normalized by Quantification condition CTL without TGFb.

	SMAD2				
	siCTL	siSMAD2	siSMAD3	siSMAD4	siSMAD2/3/4
- TGFβ	100%	8%	79%	68%	12%
+ TGFβ	26%	4%	70%	45%	4%

	SMAD3				
	siCTL	siSMAD2	siSMAD3	siSMAD4	siSMAD2/3/4
- TGFβ	100%	38%	10%	54%	41%
+ TGFβ	26%	32%	11%	41%	24%

Anti-SMAD4 antibody

Reblot with anti-actin antibody

Quantification: band intensity/actin band intensity then Quantification condition of interest normalized by Quantification condition CTL without TGFb (per blot)

SMAD4	Blot 1				Blot 2	
	siCTL	siSMAD2	siSMAD3	siSMAD4	siCTL	siSMAD2/3/4
- TGFβ	100%	108%	222%	2%	100%	8%
+ TGFβ	20%	79%	69%	3%	41%	18%

In addition, I would need from you:

- a short, two-sentence summary of the manuscript (not more than 35 words).
- two to four short (!) bullet points highlighting the key findings of your study (two lines each).
- a schematic summary figure that provides a sketch of the major findings (not a data image) in jpeg or tiff format (with the exact width of 550 pixels and a height of not more than 400 pixels) that can be used as a visual synopsis on our website.

✓ **Authors' response**

We have now provided the requested texts and summary figure in the last revised version of the manuscript.

Two-sentence summary of the manuscript: "The study identifies LINC00313 as a TGF β -regulated long non-coding RNA in cholangiocarcinoma cells. Mechanistically, by modulating key genes of the Wnt pathway, LINC00313 fine-tunes Wnt/TCF/LEF-dependent transcriptional responses and promotes cholangiocarcinogenesis."

Response to the Referees' comments

Referee #1:

The revised manuscript by Papoutsoglou et al. greatly improved compared to the initial version. I have appreciated the author's effort to expand the method section and provide technical and statistical details. Despite some minor points remain (e.g. TCGA analysis not described), overall the manuscript provides multiple evidence characterizing LINC00313 in CCA and is suitable for publication in this journal.

✓ **Authors' response**

We really thank the reviewer for appreciating our efforts during the revision and for finding this version of our manuscript suitable for publication. We have also added a paragraph in the Materials and Methods section, describing the TCGA analysis performed in this study.

Referee #2:

Authors have correctly addressed the concerns and modified the manuscript properly, including a lot of new experiments. From my part, paper is ready for publication.

✓ **Authors' response**

We appreciate the positive evaluation of our manuscript by the reviewer and her/his favorable suggestion towards publication.

Dr. Cedric Coulouarn
INSERM, France
Inserm U1242 - Oncogenesis, Stress, Signaling (OSS)
Rue de la Bataille Flandres Dunkerque, Bat D, 1er étage
Rennes 35042
France

Dear Dr. Coulouarn,

I am very pleased to accept your manuscript for publication in the next available issue of EMBO reports. Thank you for your contribution to our journal.

Yours sincerely,
